# BIGBIO: A Framework for Data-Centric Biomedical Natural Language Processing

**Jason Alan Fries**[1*]    **Leon Weber**[2,3*]    **Natasha Seelam**[4*]    **Gabriel Altay**[5*]
**Debajyoti Datta**[6†]    **Ruisi Su**[7†]    **Samuele Garda**[2†]    **Sunny MS Kang**[8†]
**Stella Biderman**[9,10†]    **Matthias Samwald**[11†]    **Stephen H. Bach**[12†]    **Wojciech Kusa**[13†]
**Samuel Cahyawijaya**[14†]    **Fabio Barth**[2†]    **Simon Ott**[11†]    **Mario Sänger**[2†]    **Bo Wang**[15]
**Alison Callahan**[1]    **Daniel León Periñán**[16]    **Théo Gigant**[7]    **Patrick Haller**[2]
**Jenny Chim**[17]    **Jose Posada**[18]    **John Giorgi**[19]    **Karthik Rangasai Sivaraman**[20]
**Marc Pàmies**[21]    **Marianna Nezhurina**[22]    **Robert Martin**[2]    **Moritz Freidank**[23]
**Nathan Dahlberg**[7]    **Shubhanshu Mishra**[24]    **Shamik Bose**[7]    **Nicholas Broad**[25]
**Yanis Labrak**[26]    **Shlok S Deshmukh**[27]    **Sid Kiblawi**[28]    **Ayush Singh**[7]    **Minh Chien Vu**[29]
**Trishala Neeraj**[30]    **Jonas Golde**[2]    **Albert Villanova del Moral**[25]    **Benjamin Beilharz**[31]

[1]Stanford University    [2]Humboldt-Universität zu Berlin
[3]Max Delbrück Center for Molecular Medicine    [4]Sherlock Biosciences    [5]Tempus Labs Inc.
[6]University of Virginia    [7]BigScience    [8]Immuneering    [9]EleutherAI    [10]Booz Allen Hamilton
[11]Institute of Artificial Intelligence, Medical University of Vienna    [12]Brown University
[13]TU Wien    [14]The Hong Kong University of Science and Technology
[15−31]See Appendix B    *Equal Contribution    †Equal Contribution
Corresponding Authors: `jason-fries@stanford.edu`  `leonweber@posteo.de`
`nseelam1@gmail.com`  `gabriel.altay@gmail.com`

## Abstract

Training and evaluating language models increasingly requires the construction of *meta-datasets* – diverse collections of curated data with clear provenance. Natural language prompting has recently lead to improved zero-shot generalization by transforming existing, supervised datasets into a variety of novel instruction tuning tasks, highlighting the benefits of meta-dataset curation. While successful in general-domain text, translating these data-centric approaches to biomedical language modeling remains challenging, as labeled biomedical datasets are significantly underrepresented in popular data hubs. To address this challenge, we introduce BIGBIO a community library of 126+ biomedical NLP datasets, currently covering 13 task categories and 10+ languages. BIGBIO facilitates reproducible meta-dataset curation via programmatic access to datasets and their metadata, and is compatible with current platforms for prompt engineering and end-to-end few/zero shot language model evaluation. We discuss our process for task schema harmonization, data auditing, contribution guidelines, and outline two illustrative use cases: zero-shot evaluation of biomedical prompts and large-scale, multi-task learning. BIGBIO is an ongoing community effort and is available at this URL.

## 1   Introduction

In large-scale language modeling, creating *meta-datasets* – diverse collections of curated data with clear provenance – is a foundational component of self-supervised machine learning, The sources of these data have significant impact on downstream model behaviors such as performance on domain-specific tasks [16], recapitulating biases present in data [56, 20], and few-shot generalization [39]. While early language models were pretrained using only unlabeled, web-crawled data [37, 8], recent

36th Conference on Neural Information Processing Systems (NeurIPS 2022) Track on Datasets and Benchmarks.

models such as T0 have demonstrated performance gains in zero-shot classification by also training on task data encoded as natural language prompts for *instruction tuning* [42, 50, 53]. Here, existing labeled datasets are transformed into prompted training examples, which redefine classification tasks as generative, text completion tasks [36]. Related findings from massive, multi-task learning (MTL) [1] further highlight the benefits of using a pre-finetuning step to incorporate existing labeled task data into language modeling.

The importance of carefully controlling the data a language model is exposed to during training highlights how meta-dataset curation is critical for state-of-the-art language modeling. This process aligns with the principles of *data-centric machine learning*, which focuses on workflows for curating training data to improve model performance. Prompting offers new opportunities for constructing meta-datasets that capture desirable language reasoning skills. In the general NLP domain, data-centric methods have benefited from community efforts such as Hugging Face's datasets hub [26], which provides easy, programmatic access to thousands of datasets and their attributes. However, biomedical datasets are significantly underrepresented in popular dataset hubs [14] creating challenges in reproducibly accessing, curating, and remixing biomedical data for prompting and other use cases.

To help address these challenges, we introduce BIGBIO (*Big*Science *Bio*medical), a community library for programmatically accessing biomedical NLP datasets at scale and encouraging reproducibility when generating meta-datasets. BIGBIO was developed as part of BigScience, a year-long workshop on large language modeling, and codifies many lessons of the biomedical working group as we developed dataset curation strategies. BIGBIO is, to the best of our knowledge, the largest public collection of curated and unit-tested biomedical NLP datasets.

A summary of our contributions:

- Programmatic access to 126+ unit-tested, biomedical datasets, covering 13 tasks, 10+ languages, and providing structured metadata for key attributes on provenance and licensing.
- Support for multiple lightweight schemata, which preserve the dataset as released and provide harmonized access for prompt engineering and cross-dataset integration.
- Community tools and guides for contributing new datasets.
- BIGBIO is built upon Hugging Face's datasets library, integrating with PromptSource [3], a prompt engineering system and repository, and the EleutherAI Language Model Evaluation Harness [17] to support rapidly designing and evaluating prompts on biomedical tasks.

We illustrate the utility of BIGBIO in two representative use cases: (1) zero-shot, prompted biomedical language model evaluation; and (2) large-scale MTL using 107 tasks. In our zero-shot evaluation of 14 language models, ranging from 220M to 176B parameters, we find that only the T0 family consistently outperformed a simple majority class baseline for question answering and entailment. We found GPT-3 performed best for prompted biomedical entity recognition, formulated as as summarization task, in some cases doubling performance over T0 models. Our MTL experiments suggest prompting may be a key component to facilitating transfer learning across tasks common in biomedical NLP. In both use cases, we substantially lower the engineering costs required to construct the meta-datasets commonly utilized for language modeling and other machine learning applications.

## 2 Related Work

BIGBIO is a data-centric approach to natural language processing in the biomedical domain. We briefly overview related work in these two areas.

### 2.1 Data-Centric Machine Learning

*Data-centric machine learning* emphasizes the thoughtful curation of data as centrally important to the development of models. Multiple arguments for this emphasis have been advanced. Paullada et al. [33] survey many aspects, including mitigating biases and annotation artifacts in training data that lead models to rely on spurious correlations that do not generalize to other datasets, and addressing representational harms in which certain people are under, over, or misrepresented. Sambasivan et al. [41] document prevalent "data cascades," situations in AI and machine learning practice in which low-quality data causes downstream problems in high-stakes applications. Biderman and Scheirer [4]

make several recommendations for improved data practices, including auditing and documenting datasets. Rogers [40] outlines issues with models that can be exacerbated by low-quality data. This encompasses for instance: learning spurious patterns, being vulnerable to basic input perturbations, and struggling with rare inputs. BIGBIO is motivated by these same arguments, hence its emphasis on careful metadata curation and harmonized task schemata.

Data quality has a large impact on model performance. Deduplicating data leads to more accurate and more robust models with faster convergence. [9, 25]. For instance, cleaning up the consistency of answer response strings was reported to improve biomedical question answering [55]. Duplication contamination is a serious risk in biomedical datasets, which often iteratively build or extend prior annotations, introducing risk of test leakage in evaluation [13]. As we describe in §3, BIGBIO's centralization of data in a unified format enables systematic data quality checks.

Data governance is also an important issue when curating biomedical language data. Jernite et al. [22] survey many aspects of the governance of language data, and propose a framework for distributed governance of large language corpora. Vayena et al. [46] describe models of data governance that enable biomedical research while respecting patient privacy. Jones et al. [23] propose data governance standards for clinical text data with personally identifiable information. Some of these issues are not directly applicable to BIGBIO, which currently only includes loaders for datasets that are compliant with the United States Health Insurance Portability and Accountability Act (HIPAA) as public research datasets. Further, BIGBIO is not itself a repository of data, but a centralized repository of data loaders and metadata, meaning that future dataset creators can programmatically define how a dataset should be accessed and share this information with the community.

## 2.2 Biomedical Benchmarks

Task-specific benchmark datasets are common in biomedical workshops like BioNLP and BioCreative [24, 21]. However, these datasets typically assess a restricted set of skills learned by a model. Several recent efforts have focused on curating larger collections of datasets and tasks to evaluate the performance of biomedical NLP models. BLUE (Biomedical Language Understanding Evaluation) is a benchmark for 10 datasets representing 5 tasks [34]. BLURB (Biomedical Language Understanding and Reasoning Benchmark) includes 13 datasets and 7 tasks [19]. HunFlair provides harmonized access to 23 NER datasets, but imposes assumptions on preprocessing choices (e.g., tokenization) [49]. Most benchmarks provide no multilingual data. CBLUE is the only non-English benchmark consisting of 8 datasets and tasks for Chinese biomedical language [57].

Multiple biomedical prompt datasets have been released for few and zero-shot classification. NATURAL-INSTRUCTIONS$_{v2}$ provides 1600+ task instructions for a variety of domains, including 30 tasks for medicine and healthcare [48]. BoX provides natural language instructions for 32 datasets and 9 tasks, where instructions consist of an explanation, a prompt, and a collection of example input/outputs [32]. Agrawal et al. [2] released 2 datasets for zero-shot clinical information extraction.

BIGBIO differs from previous efforts by focusing on the infrastructure and curation required to reproducibly generate meta-datasets. Existing benchmarks provide consistent mechanisms for evaluating machine learning performance, however they do not support consistent tooling to access and ingest data into machine learning workflows. This is a serious limitation in practice, especially as novel training and evaluation strategies increasingly require transforming input data. We emphasize direct, easy and programmatic access to datasets with community curation to build open tools for data loading. We have curated detailed metadata about tasks, e.g., languages, licensing and other aspects of dataset provenance. We provide harmonized views of datasets by task schema, enabling easier integration into workflows, while also imposing minimal assumptions on NLP preprocessing decisions like sentence splitting and tokenization. Existing benchmarks typically fix preprocessing choices, creating challenges when comparing end-to-end workflows common in prompting.

## 3 The BIGBIO Framework

This research effort was initiated as part of BigScience, a year-long workshop on the creation of very large language models, comprised of over 1000 researchers from 60 countries and dozens of working groups. The BigScience biomedical working group consisted of machine learning

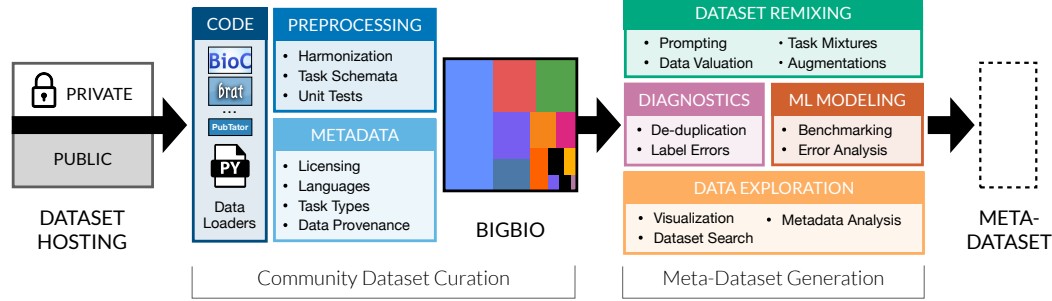

Figure 1: BIGBIO's data-centric workflow. Datasets are hosted by their original owners to preserve data governance. Community curation focuses on shared code for reproducibly loading, preprocessing, and harmonizing data. These standardized datasets can then be integrated into workflows that further transform and curate meta-datasets for downstream use cases, e.g., training large language models.

researchers and other stakeholders interested in the curation of biomedical data for language modeling. BIGBIO reflects the lessons and best practices we learned while developing a framework for more easily and reproducibly generating biomedical NLP meta-datasets.

Fig. 1 outlines BIGBIO's data-centric workflow, which emphasises reproducible preprocessing, meta-data access, and quality assurance checks via community dataset curation. Dataset creators contribute data loaders that define the terms by which the community accesses the data. Datasets themselves remain hosted by their original owners, preserving existing data governance. BIGBIO supports loading public datasets hosted online and private datasets that can only be loaded locally in secure environments. We outline the community curation process in detail below and discuss downstream meta-dataset generation techniques that benefit from data-centric practices.

## 3.1 Community Dataset Curation

**Building the Dataset Catalog** Our initial efforts in the BigScience working group produced a catalog of important biomedical datasets, key metadata, and other provenance [14]. Selection criteria followed several principles: (1) relevance to biomedical research, (2) diversity of domains, tasks, and languages; and (3) researcher accessibility. We used this catalog as the starting point for BIGBIO.

**Task Schema Harmonization** In biomedical NLP there are a proliferation of data formats (e.g., BioC, BRAT) but inconsistent adherence across those formats. Developing common data models for interoperability [7], while beneficial for cross-dataset integration, risks possible information loss when translating or *harmonizing* information across schemata. To develop shared infrastructure for data ingestion and minimize information loss, we designed data loaders to support 2 dataset views: (1) a source schema that preserves the original dataset format as faithfully as possible; and (2) task-specific, harmonized BIGBIO schema. We developed 6 lightweight schema (see §C) supporting common NLP tasks including knowledge base construction (KB), question answering (QA), textual entailment (ENTAIL), text to text (T2T), textual pairs (PAIRS), and text classification (TEXT).

**Unit Tests and Dataset Cleaning** To safeguard correctness of data loader implementations, we developed a testing suite of unit-tests for monitored quality issues. BIGBIO schemata are designed to support key dataset integrity checks, such as enforcing unique IDs across elements, relational consistency, confirming text offsets are correctly aligned within document text, etc. The unit testing suite is runnable as part of the dataset submission process, providing feedback on diagnosing implementation or dataset errors. Where possible, we implemented tools for common data cleaning tasks, such as normalizing PubMed IDs (PMIDs).

**Acceptance Checklist** Submissions require completing a checklist of inclusion criteria before acceptance into the project GitHub repository. The complete checklist is provided in §F and covers metadata annotation, correctness of task choices by dataset, materialization of all data subsets defined by the original dataset creators, and passing all unit tests. All public datasets were manually reviewed and accepted by a BIGBIO admin. Local datasets that required manual downloading were checked if

an admin had appropriate authorization (e.g., several authors have PhysioNet credentials). In absence of dataset access, data loaders were accepted contingent on providing successful unit test logs.

**Iterative Improvement**   Most biomedical datasets involve complex labeling tasks, so even in cases when datasets pass unit tests they may contain subtle bugs or misunderstandings that require revisiting. To identify and improve our data loader implementations, we implemented several representative use cases outlined in §5. Implementing these machine learning workflows enabled identifying non-obvious dataset errors or limitations in our current schema. For example, some datasets do not provide natural language class labels, such as labeling a relation with an internal code (CPR:6) instead of language describing the underlying biological relationship (ANTAGONIST), which creates challenges when writing prompts.

## 3.2   Meta-Dataset Generation

Data-centric methods for generating meta-datasets require adding various complex operations into machine learning workflows. Data cleaning methods [51] handle de-duplicating data, identifying label errors [29], and filtering toxic or biased content [28]. Unlabeled data can be weakly labeled to define new training tasks [38, 15] and existing labeled datasets can be remixed into new forms using prompting. Data valuation measures can be used to filter existing datasets or guide sampling of new data [18]. Implementing these workflows to support reproducible machine learning benefits from modular software components that connect to a standardized API for accessing raw data. For example, developing the "P3: Public Pool of Prompts" dataset, which was used to fine-tune the T0 language models, was accelerated by a standardized dataset API that enabled scaling to 170+ datasets and 2,000+ prompts.

However, replicating the creation of P3 using biomedical data faced several challenges. We found that only 13% of the BIGBIO catalog was available via existing dataset APIs. The lack of programmatic access to biomedical datasets was a prime motivation for developing BIGBIO. To demonstrate the utility of our library for meta-dataset generation, we integrated BIGBIO with several downstream data-centric frameworks. First, we integrated with PromptSource [3], a development environment for prompts, to enable creating prompted representations of BIGBIO datasets. For evaluating language models and prompts, we interface with the EleutherAI Language Model Evaluation Harness [17]. This harness handles the loading, querying, and scoring of language models, with programmatic definitions of how evaluations are carried out. Here BIGBIO's unified task schemata enable standardized evaluation schemes to be automatically applied to a wide collection of datasets.

## 3.3   Biomedical Hackathon

After internally testing the elements outlined in §3.1, we drafted instructional material and code tutorials for external collaborators. We then launched an international call for participation in a biomedical hackathon to implement all 174 datasets in the BIGBIO catalog. Participants were recruited through Twitter. We established formal participation guidelines and corresponding credit, including co-authorship on this manuscript, given implementation of 3 or more data loaders. The hackathon officially ran for 2 weeks with an unofficial 2 week wrap-up period. During the official period, we held daily office hours to help participants, running a Discord server to facilitate rapid communication and up-to-date FAQ. At the hackathon's conclusion, 48 participants had implemented 126 total datasets with an additional 18 dataset still undergoing quality control.

## 4   The BIGBIO Dataset

Table 1: Summary statistics for BIGBIO. Note datasets may contain multiple schema.

|  | KB | TEXT | PAIRS | QA | ENTAIL | T2T | ALL |
|---|---|---|---|---|---|---|---|
| Datasets | 84 | 21 | 12 | 9 | 4 | 7 | 126 |
| Public Datasets | 73 | 12 | 12 | 8 | 1 | 6 | 105 |
| Private Datasets | 11 | 9 | 0 | 1 | 3 | 1 | 21 |
| PubMed Datasets | 64 | 7 | 3 | 5 | 0 | 1 | 77 |
| Languages | 7 | 4 | 1 | 1 | 1 | 4 | 10 |
| Tasks | 5 | 1 | 2 | 1 | 1 | 3 | 13 |

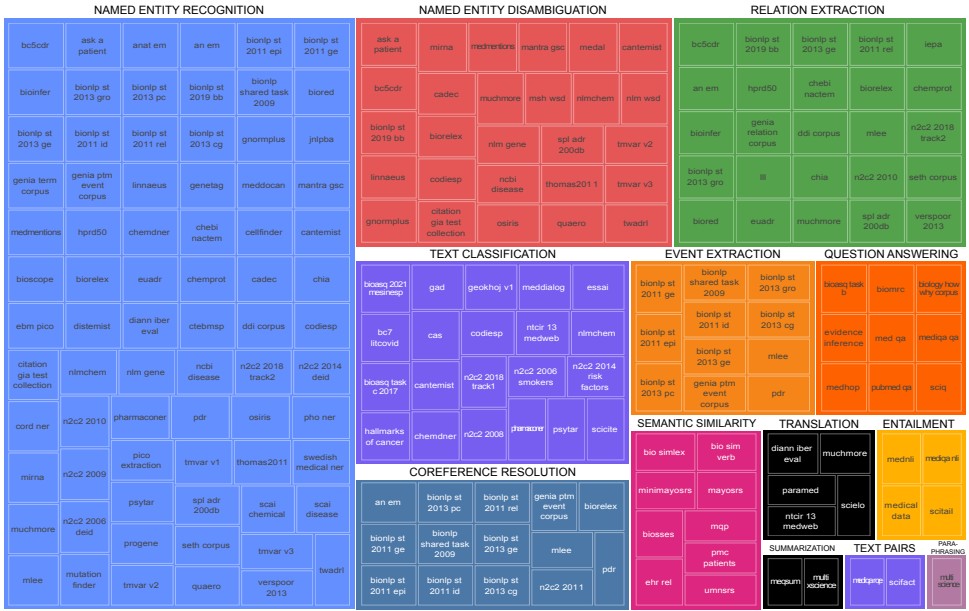

Figure 2: Treemap visualization of BIGBIO's 126 datasets and 13 task categories, denoted by color.

We provide a `bigbio` Python package that supports streamlined loading of 126 biomedical datasets covering 13 tasks grouped into 6 schema types for a total of 24 million examples comprising 18 trillion characters. To the best of our knowledge, BIGBIO is the largest single collection of curated and unit tested biomedical NLP datasets. Figure 2 visualizes the datasets and tasks in BIGBIO and Table 1 provides dataset counts by schema and key attributes. Publicly available datasets (105/126) can be automatically downloaded. The remaining 21 datasets require further access approvals, but can be loaded locally after securely downloading files. This restriction is common in clinical datasets, which require credentialing and training on how to handle protected health information.

**Metadata Summary** Overall 10 languages are represented, with English being the majority (83%) followed by Spanish (6.5%), French (2.9%), Chinese (2.2%), and German (1.4%). Japanese, Dutch, Portuguese, Swedish, and Vietnamese are each present in one dataset. Creative Commons licenses are used more frequently than any other type covering 44 (35%) of datasets with 8 (6.3%) using the non commercial use (NC) option. In 34 (27%) of datasets, the license is unknown, corresponding to cases where dataset authors did not choose or clearly denote a license. The remaining licenses are a mixture of open source and custom data use agreements. See Appendix §D for a complete summary.

## 5 Use Cases

We describe two downstream use cases of BIGBIO to showcase the utility of the library. In the first, we evaluate zero-shot performance of prompted language models on 10 biomedical tasks. In the second, we train a large-scale biomedical multi-task learning (MTL) model using 107 tasks. Both cases were run using an 8x A40 and 4x RTX 3090 compute node. BLOOM and OPT models were evaluated on a 4xA100 80GB node using LLM.int8()[11]. Expanded results, experimental details, and additional use cases are described in Appendices §J zero-shot evaluation, §K MTL, §H de-duplication, and §I data visualization.

### 5.1 Zero-Shot Evaluation of Prompted Language Models

**Datasets and Prompt Engineering** We selected 10 representative datasets from BIGBIO: BIOSSES (semantic textual similarity), BioASQ (yes/no question answering), GAD (relation extraction), SciTail (textual entailment), MedNLI (clinical textual entailment), and 5 NER tasks (BC5-Chemical, BC5-Disease, BC2GM, JNLPBA, NCBI Disease) from the BLURB benchmark. For each dataset, we wrote 5 prompts using PromptSource to reflect the original classification task, using task

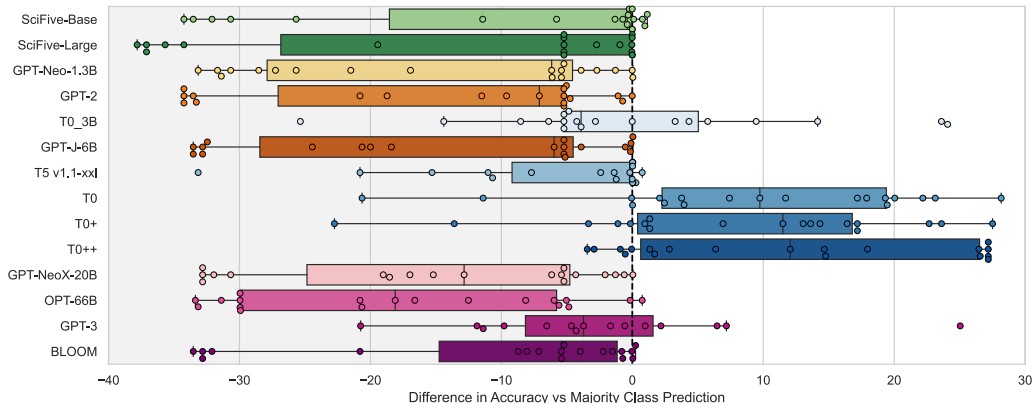

Figure 3: Zero-shot generalization to biomedical tasks. Box plots show pooled accuracy differences between a majority class baseline and zero-shot prediction for BioASQ, SciTail, MedNLI, and GAD. Points are per-prompt scores. T0 is the only family to consistently outperform the majority baseline.

templates already available in PromptSource when possible. NER tasks were formulated as text translation tasks, similar to TANL [31], where the output is a list of all entities found in the prompted text. We did not perform any iterative prompt tuning to improve performance of templates.

**Models and Baselines**    We evaluated 14 pretrained language models, ranging from 220 million to 176B billion parameters: SciFive-base/large [35], GPT Neo-1.3B [6], GPT-2 [36], GPT-J-6B [47], the T0 family [42], the base T5 model used to train T0 [37], GPT-NeoX-20B [5], OPT-66B [58], GPT-3 (text-davinci-002) [30], and BLOOM [52]. We focus on autoregressive and sequence-to-sequence language models due to their strong performance in text generation tasks. We included models that were exposed to PubMed text during pretraining (SciFive, GPT Neo-1.3B, GPT-J-6B) and models that were only pretrained on web crawled text (T5, T0 family, GPT-2, OPT, BLOOM). For classification tasks, we report a *majority class* baseline, defined as a rule-based classifier that always predicts the majority class. Where possible, to contextualize scores, we also report fine-tuned state-the-art (*fine-tuned SOTA*) performance as reported in the literature [44, 54, 35]. We did not evaluate OPT or BLOOM on NER tasks due to their high compute costs, e.g., 35 hours per dataset and model.

**Evaluation Protocol**    Models were evaluated using a BigScience prompted evaluation library. All evaluations used the canonical test split where possible, otherwise we used BLURB's test set definitions. Performance for BIOSSES is reported using Pearson's correlation after casting outputs to numbers. For NER tasks, since calculating entity-level F1 would require defining a heuristic string alignment between source text and output list, we report ROUGE-1[27], which measures matching unigrams between the predicted and true entity lists. All other tasks use accuracy.

**Results**    Fig. 3 shows that T5 and GPT model families often fail to generalize to biomedical text, regardless of parameter count or exposure to biomedical text during pretraining. However, the T0 family demonstrated somewhat surprising generalization capabilities, as these models were not exposed to any biomedical tasks during pretraining or prompted fine-tuning. Tables 2 and 3 includes performance statistics for all language models and datasets. For classification tasks, we replicate the finding in Sanh, et al. (2021) that models using more prompted pretraining tasks demonstrate better generalization, finding that T0++ performed best overall. Non-T0 models frequently performed worse than a simple majority class baseline and in many cases predicted completions were pathological, i.e., emitting the same answer for all prompts. For NER tasks, we find that GPT-3 performs best overall, however T0 consistently outperforms non-T0 language models of similar parameter counts. On BioASQ and SciTail using T0++, the best prompts performed very well, falling 0.5 and 8.2 points short of state of the state-of-the-art supervised models. The remaining datasets posed significant challenges for all models, although further manual prompt tuning or applying methods for automated prompt learning [12] could result in better performance.

Since both the T0 family and the latest version of GPT-3 were trained using instruction tuning, comparing these models' generalization to biomedical text opens up questions around the role prompt

Table 2: Zero-shot classification performance of prompted language models

| Model | Params | BIOSSES (Pearson) | | BioASQ (Accuracy) | | SciTail (Accuracy) | | MedNLI (Accuracy) | | GAD (Accuracy) | |
|---|---|---|---|---|---|---|---|---|---|---|---|
| | | Avg | Best | Avg | Best | Avg | Best | Avg | Best | Avg | Best |
| SciFive-Base† | 220M | 17.9 | 52.2 | 35.9 | 41.4 | 60.1 | 60.4 | 63.7 | 66.8 | 51.9 | 53.7 |
| SciFive-Large† | 770M | 15.3 | 44.7 | 30.7 | 32.9 | 59.8 | 60.4 | 61.6 | 66.7 | 47.4 | 47.4 |
| GPT-Neo-1.3B† | 1.3B | 36.4 | 36.4 | 43.9 | 67.1 | 50.7 | 60.4 | 43.0 | 64.0 | 47.9 | 51.3 |
| GPT-2 | 1.5B | 10.9 | 19.5 | 38.4 | 60.0 | 50.4 | 60.4 | 54.1 | 66.0 | 47.4 | 47.6 |
| GPT-J-6B† | 6B | 0.2 | 32.1 | 40.6 | 67.1 | 51.6 | 60.3 | 46.3 | 62.8 | 48.2 | 52.1 |
| T0_3B | 3B | 0.9 | 0.9 | 63.9 | 72.9 | 69.9 | 84.5 | 64.6 | 76.2 | 47.7 | 48.7 |
| T5 v1.1-xxl | 11B | -6.1 | 30.5 | 64.9 | 67.9 | 54.1 | 60.7 | 52.4 | 65.5 | 52.1 | 52.6 |
| T0 | 11B | 27.2 | 32.0 | 86.4 | 89.3 | **72.3** | **88.6** | 68.8 | 78.4 | **55.1** | **56.6** |
| T0+ | 11B | 35.2 | 40.7 | 84.3 | 90.7 | 71.0 | 87.9 | 68.9 | 79.7 | 52.4 | 53.9 |
| T0++ | 11B | 26.9 | 26.9 | **94.1** | **94.3** | 71.6 | 87.0 | **74.2** | **81.4** | 53.5 | 55.4 |
| GPT-NeoX-20B† | 20B | -14.8 | -6.5 | 41.3 | 67.1 | 50.5 | 59.8 | 48.6 | 62.4 | 47.9 | 51.3 |
| OPT-66B | 66B | - | - | 43.0 | 67.9 | 44.7 | 52.3 | 38.1 | 48.6 | 48.3 | 52.4 |
| GPT-3 | 175B | **47.3** | **64.5** | 73.0 | 92.1 | 52.0 | 61.4 | * | * | 48.4 | 50.9 |
| BLOOM | 176B | 0.5 | 17.7 | 40.9 | 67.1 | 52.4 | 59.6 | 64.0 | 66.9 | 48.8 | 51.9 |
| Majority Class | | - | | 67.1 | | 60.4 | | 66.7 | | 52.6 | |
| Fine-tuned SOTA | | 94.5 | | 94.8 | | 96.8 | | 86.6 | | 84.9 | |

† pretraining included PubMed/PubMed Central documents. * DUA prevents API usage.

Table 3: Zero-shot NER performance of prompted language models

| Model | Params | BC5-Disease | | BC5-Chemical | | NCBI Disease (ROUGE-1 F1) | | BC2GM | | JNLPBA | |
|---|---|---|---|---|---|---|---|---|---|---|---|
| | | Avg | Best | Avg | Best | Avg | Best | Avg | Best | Avg | Best |
| SciFive-Base† | 220M | 2.6 | 7.1 | 1.9 | 4.6 | 2.1 | 4.8 | 4.1 | 9.6 | 7.9 | 16.1 |
| SciFive-Large† | 770M | 5.4 | 8.7 | 5.1 | 7.5 | 7.0 | 9.7 | 9.4 | 13.1 | 14.7 | 21.0 |
| GPT-Neo-1.3B† | 1.3B | 5.6 | 7.2 | 4.0 | 6.1 | 6.0 | 8.6 | 6.5 | 7.7 | 11.4 | 16.9 |
| GPT-2 | 1.5B | 5.2 | 7.7 | 4.4 | 6.6 | 5.7 | 7.5 | 8.3 | 10.3 | 12.7 | 17.4 |
| GPT-J-6B† | 6B | 4.9 | 7.3 | 3.4 | 5.0 | 5.0 | 9.1 | 7.1 | 10.5 | 10.9 | 17.4 |
| T0_3B | 3B | 38.6 | 41.2 | 23.1 | 26.5 | 28.5 | 34.0 | 22.0 | 23.4 | 16.7 | 23.7 |
| T5 v1.1-xxl | 11B | 3.3 | 3.7 | 3.1 | 3.8 | 4.3 | 5.6 | 5.0 | 5.9 | 7.2 | 10.0 |
| T0 | 11B | **46.5** | 58.8 | 30.8 | 43.9 | 38.6 | 56.0 | 24.0 | 29.5 | 16.2 | 23.5 |
| T0+ | 11B | 44.4 | 54.0 | 29.4 | 40.7 | 36.3 | 50.1 | 25.0 | 27.1 | 11.1 | 25.2 |
| T0++ | 11B | 43.1 | 49.1 | 28.6 | 35.2 | 36.2 | 47.7 | 25.1 | 25.7 | 13.3 | 19.7 |
| GPT-NeoX-20B† | 20B | 5.7 | 10.3 | 3.5 | 5.5 | 5.5 | 8.9 | 7.0 | 9.9 | 8.9 | 13.0 |
| GPT-3 | 175B | 40.5 | **63.3** | **36.9** | **60.8** | **40.4** | **66.7** | **39.1** | **64.5** | **37.7** | **48.6** |

templates play in enabling transfer learning across domains. Our biomedical prompts are largely based on general domain templates used to tune T0, so these prompts are more "in-distribution" than completely novel prompts, even when populated with biomedical text. This likely gives an advantage to T0 when evaluating performance. GPT-3 instruction tuning likely takes a different form, potentially explaining why performance was generally lower for non-NER tasks when using GPT-3.

## 5.2 Large-Scale Multi-Task Learning

Recent work on multi-task learning has found that incorporating an additional *pre-finetuning* step (i.e, after pretraining but before finetuning) can lead to better learned representations and performance improvements that scale linearly with the number of MTL tasks [1]. However, pre-finetuning requires using a large number (15+) of heterogeneous tasks, a scale of MTL that is underexplored in biomedical NLP. We investigate the impact of massive MTL on biomedical pre-finetuning using 67 BIGBIO datasets to materialize 107 training tasks.

**Data Materialization**   We train and evaluate a multi-task learning model using the MaChAmp MTL framework [45]. We generated training and evaluation splits using 106 datasets that were available in the BIGBIO repository when we started the MTL experiment. From these 106 datasets, we filtered out datasets that: were non-English; had known implementation bugs; included silver-standard annotations; or were document-level or multilabel classification datasets. For the 67 remaining datasets, we extracted data for 8 task types: Named Entity Recognition, Text Classification, Question Answering, Coreference Resolution, Event Detection, Event Argument Extraction, Relation Extraction and Semantic Textual Similarity, yielding 107 tasks (dataset/task type combinations).

**Training Protocol**   We train a single encoder-only transformer model with a separate classification head for each of the 107 tasks. We initialize the encoder with BioLinkBERT-base [54]. We follow the strategy Aghajanyan et al. (2021) outlined in MUPPET and use task-heterogeneous batching. At each training step, we sample 32 different tasks and select 16 examples for each of them leading to a total batch size of 512. We train the model to convergence, which takes less than 50 epochs and then select the best performing checkpoint based on validation performance.

**Evaluation Protocol**   We evaluate our model on a subset of dataset from the BLURB benchmark. We select all four datasets that are contained in our MTL training data and have the same splits in the MTL data as in BLURB. For all datasets, we use the version in the MaChAmp format, which differ in tokenization, sentence splitting and label space from the official BLURB versions. After prediction, we postprocess the results to match the BLURB label space. While this introduces confounders that makes direct comparison complicated, e.g., different choices in sentence splitting and tokenization, we include prior state-of-the-art results for the same model size [54] as a point of orientation. We additionally compare with a version of our MTL model that we fine-tune on the training data of the evaluation dataset using the MaChAmp default hyperparameters.

**Results**   MTL results are reported in Table 4. MTL+Finetuning results are reported as the mean and standard deviation of 3 different random seeds. For contextualizing scores, we also include state-of-the-art LinkBERT-base results. The MTL model performs markedly worse than the state-of-the-art LinkBERT model, with differences between 1.5 and 11.2 percentage points (pp) F1. However, additional fine-tuning only on the evaluation dataset narrows the gap between LinkBERT and the MTL model significantly with a maximum difference of 3.2 pp F1. This confirms the results of [1] that models trained using large-scale MTL setting are a suitable basis for further fine-tuning. However, the failure of the fine-tuned model to perform better than state-of-the-art indicates that more research on the conditions in which large-scale MTL pre-finetuning may improves results is required.

Table 4: F1 scores of the MTL model evaluation

| Dataset | Task | MTL | MTL+Finetuning | LinkBERT-base |
|---|---|---|---|---|
| NCBI-Disease | NER | 80.2 | 87.5 ± 0.9 | *88.2 |
| BC5CDR-Disease | NER | 78.5 | 84.8 ± 0.3 | *86.1 |
| BC5CDR-Chemical | NER | 92.2 | 94.4 ± 0.3 | *93.8 |
| ChemProt | RE | 66.4 | 74.3 ± 0.1 | *77.6 |

* indicates that comparing results is complicated by different preprocessing choices across benchmarks.

We suspect several possible explanations for our lower performance. First, more exploration of task head configurations could lead to better performance. MUPPET found that some tasks benefited from using a separate classification head while others resulted in severe overfitting. We did not systematically explore different task head configurations due to computational cost. Second, our biomedical task mixture is different than MUPPET, which focused on general reasoning tasks (e.g., summarization, reading comprehension). Biomedical datasets, in contrast, are skewed towards information extraction-style tasks [14]. We hypothesize the paucity of similar reasoning-type datasets in biomedical NLP may impact the overall benefit of using MTL with task head approaches.

Finally, we took the common approach of creating a unified input format by adding non-linguistic markers to differentiate structural elements required by a task, e.g., denoting relation entities with prefix/suffix tokens. While this unifies all inputs for training, it is unclear how well this strategy facilitates transfer learning across tasks. Some evidence is suggested by In-BoXBART [32], which looked at MTL using 32 biomedical tasks reformulated as unified, text-to-text tasks using prompts. Their

task mixture is similar to ours and their text outputs contain similar structural markers. Their vanilla MTL method only trains on input/outputs without any additional prompt context and substantially underperforms in corresponding single-task models, similar to our results. However, when including prompt information, they observe consistent performance benefits over single-task models.

## 6  Discussion

The focus of BIGBIO on providing a unified view over a large number of diverse NLP datasets has a number of benefits. First, it could increase the robustness of data-centric machine learning because it allows end-to-end data generation workflows that trace data provenance and codify assumptions on data transformations, such as checking for duplicates. Second, the unified view allows to programatically assure quality of both the source data and the transformed datasets, as exemplified by our suite of unit tests. Finally, it drastically reduces the amount of work required for training or evaluating models on a large number of tasks, as can be seen in the MTL usecase, where we had to write only 8 data transformation scripts (one for each task type) as opposed to up to 67 (one for each dataset). Crucially, BIGBIO achieves this without making strong assumptions about the downstream use case or type of model, e.g. by unifying tasks directly into a conditional text generation/prompting setting.

We believe that our work provides useful suggestions on how to write data loaders for a large number of datasets in a collaborative setting. We found a uniform view of the datasets useful for quality assurance during implementation, because it allowed having a standard suite of unit tests. Furthermore, the categorization of datasets into schemas allowed code reviewers to specialize in a subset of schemas, which likely improved the quality of code reviews. Finally, we found developing our use cases immensely helpful, because this informed design decisions for the library and helped identify bugs in accepted data loaders.

Our work has several limitations. First, some data loaders likely contain implementation errors that were missed by our code review and unit tests. Second, our choice of schema makes assumptions on what structures are most useful for biomedical NLP research and thus will not represent all interesting tasks. Third, BIGBIO reflects biases that are present in the included data sets, for instance a very strong focus on English text as only 23 of the 126 currently implemented datasets are in a language other than English. Finally, our use case experiments could be expanded upon, e.g., including masked language models in our zero-shot analyses or conducting additional MTL experiments. We believe that these limitations will be mitigated over time as researchers continue to use and improve BIGBIO.

## 7  Conclusion and Future Work

We introduce BIGBIO a community library of 126+ biomedical NLP datasets currently covering 12 task categories and 10+ languages. BIGBIO enables reproducible, data-centric machine learning workflows by focusing on programmatic access to datasets and their metadata in a uniform format. We discussed our process for task schema harmonization, data auditing, contribution guidelines and describe two illustrative use cases of BIGBIO: zero-shot evaluation of large language models for biomedical prompting and large-scale MTL. We believe BIGBIO poses little-to-no negative societal impacts, as all datasets we support are public or governed by HIPAA protections as appropriate. A chief motivation of this work is the belief that codifying dataset curation choices in code, tracking provenance of meta-dataset curation, and other decisions around transparent training set generation are critical to the ethical application of machine learning. In the worst case, BIGBIO might amplify negative impacts already inherent to included datasets as it facilitates dataset access. For future work, we plan to curate a library of prompted representations of BIGBIO tasks, including prompts formulated like those used to train T0, as well as longer instruction sets for novel biomedical tasks. Constructing such a library requires a framework for reproducible data ingestion which is provided by BIGBIO.

## Acknowledgments and Disclosure of Funding

Jason Fries was supported in part by a Stanford AIMI-HAI Partnership Grant. Leon Weber acknowledges the support of the Helmholtz Einstein International Berlin Research School in Data Science

(HEIBRiDS). Samuele Garda is supported by the Deutsche Forschungsgemeinschaft as part of the research unit "Beyond the Exome". Wojciech Kusa is supported by the EU Horizon 2020 ITN/ETN on Domain Specific Systems for Information Extraction and Retrieval (H2020-EU.1.3.1., ID: 860721). Jonas Golde is supported by the German Federal Ministry of Economic Affairs and Climate Action (BMWK) as part of the project ENA (KK5148001LB0). We are grateful to CoreWeave and EleutherAI for providing the compute needed to evaluate the 6 and 11 billion parameter models on our benchmarks, and to Suzana Ilić, Clem Delangue, and others for helping to advertise our calls for participation in the biomedical hackathon. Special thanks to the entire BigScience team, including but not limited to Huu Nguyen, Vassilina Nikoulina, Aurélie Névéol, Yong Zheng-Xin, Victor Sanh, and many others, for their thoughtful discussions and contributions in support of the biomedical working group.

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
