# A Appendix Overview

This section summarizes the elements required by NeurIPS for inclusion in supplementary materials.

1. **Dataset documentation and intended uses. Recommended documentation frameworks include datasheets for datasets, dataset nutrition labels, data statements for NLP, and accountability frameworks.** We have provided datasheets for all datasets (see §M) in BIGBIO as well as a datasheet for the meta-dataset itself (see §N). The intended use of BIGBIO is to enable research on (biomedical) Natural Language Processing. Any usage for direct diagnostic use or medical decision making without review and supervision by medical professionals is out of scope.

2. **URL to website/platform where the dataset/benchmark can be viewed and downloaded by the reviewers.** All code required to download datasets and run machine learning experiments outlined in this manuscript is available on the BIGBIO GitHub code repository https://github.com/bigscience-workshop/biomedical. We are in the process of creating a website that summarizes the aims and contributions of BIGBIO.

3. **Author statement that they bear all responsibility in case of violation of rights, etc., and confirmation of the data license.** The authors of this manuscript bear all responsibility for any violation of rights caused by the development and release of BIGBIO. All code for BIGBIO is released under Apache License 2.0. All dataset licensing remains the same as the source.

4. **Hosting, licensing, and maintenance plan. The choice of hosting platform is yours, as long as you ensure access to the data (possibly through a curated interface) and will provide the necessary maintenance.** All code is hosted on GitHub at the repository linked above. We have released all dataset-related software under an Apache License 2.0. BIGBIO is an active open source project that is maintained by an international community of volunteers and 4+ code administrators associated with the BigScience biomedical working group. See §E and §B for protocols for new dataset contributions and unit testing to ensure ongoing quality checks. Datasets are hosted by their original owners. In cases where the original license permits redistribution, we will mirror dataset releases on our community hub https://huggingface.co/bigscience-biomedical.

5. **Links to access the dataset and its metadata.** See our project GitHub for all dataset code and metadata.

6. **The dataset itself should ideally use an open and widely used data format. Provide a detailed explanation on how the dataset can be read. For simulation environments, use existing frameworks or explain how they can be used.** BIGBIO is implemented using Hugging Face's datasets library to support easy integration into existing machine learning workflows. See §C for details on standardized schema to permit easier reuse.

7. **Long-term preservation** For the subset of public datasets that can be redistributed, we intend to create regular snapshots on BIGBIO on a data archiving website such as https://zenodo.org/.

8. **Explicit license** All code for BIGBIO is released under Apache License 2.0. All dataset licensing remains the same as the source. See §D and §N for complete licensing information for all datasets in BIGBIO.

9. **For benchmarks, the supplementary materials must ensure that all results are easily reproducible.** All machine learning experiments include instructions and code for reproducing results. See §J for zero-shot biomedical benchmarking and §K for multi-task learning experiments.

# B Author Contributions

The core idea behind this manuscript emerged from discussions in the BigScience biomedical working group. We formalized the following criteria for determining authorship. Joint first authorship required significant intellectual contribution shaping this project, including organization, contributing/reviewing code, writing documentation, and writing this manuscript. Co-authorship required 3+ submitted dataset implementations that passed all unit tests and other quality control measures. Co-second authorship required one or more significant contributions to the project beyond participation in the hackathon.

We also thank Giyaseddin Bayrak, Gully Burns, Antonio Miranda-Escalada, Abhinav Ramesh Kashyap and Tanmay Laud (tlaud@ucsd.edu) for their dataset contributions.

Specific contribution categories are listed below and visualized by author in Figure 4.

- **3 Datasets, 4-6 Datasets, 7+ Datasets**: Number of dataset loaders coded during the hackathon.
- **Challenging Dataset**: Implemented a difficult dataset loader (e.g., many label errors, poor documentation on structure).
- **PR Review**: Managed PR process during hackathon, including code review, debugging, and other quality control measures. This includes llive QA sessions during hackathon office hours on the team Discord server.
- **Documentation**: Wrote instructional material for participants on designing data loaders, coding tutorials, and logistics material for hackathon participation
- **Website**: Contributed to the creation of the BigBIO hackathon website.
- **Compute**: Provided computational resources for running machine learning experiments.
- **Dataset Dev**: Contributed to the design and implementation of task schema design, designing dataset loaders, data unit tests, and other dataset loader infrastructure.
- **API Dev**: Contributed to the design and development of the BIGBIO API, including querying of metadata, programmatic access across datasets, and other infrastructure.
- **Prompt Engineering**: Designed biomedical dataset prompts in PromptSource
- **Prompt Eval**: Contributed to the infrastructure of connecting BIGBIO data loaders with the language model evaluation harness and/or ran prompt evaluation experiments.
- **MTL**: Contributed to the multi-task learning experiments
- **Data Viz**: Designed data visualizations
- **Team Logistics**: Organizational tracking of team goals and action items.
- **Weekly Syncs**: Attended and contributed to weekly team meetings
- **Writing**: Contributed text or edited content within this manuscript

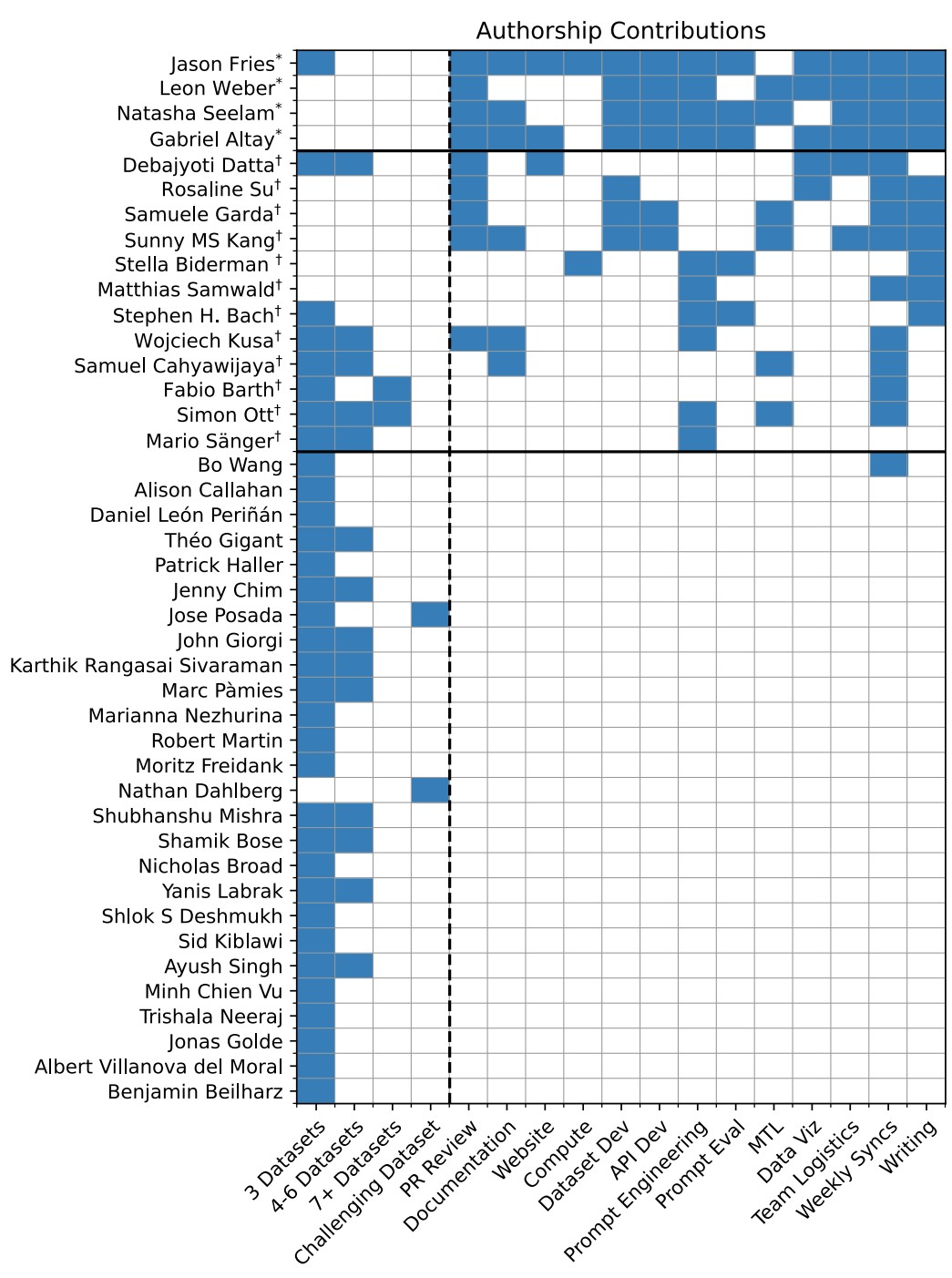

Figure 4: Authorship contribution matrix. Cells to the left of the dotted black vertical line are hackathon dataset contributions, while the right are other paper contributions as part of the BigScience biomedical working group. For each author, ∗ denotes co-first author and † denotes co-second author, with equal contributions within category.

Table 5: Author Affiliations

| Name | Affiliation |
| --- | --- |
| Jason Fries | Stanford University |
| Leon Weber | Humboldt-Universität zu Berlin & Max Delbrück Center for Molecular Medicine |
| Natasha Seelam | Sherlock Biosciences |
| Gabriel Altay | Tempus Labs Inc. |
| Debajyoti Datta | University of Virginia |
| Rosaline Su | Independent researcher |
| Samuele Garda | Humboldt-Universität zu Berlin |
| Sunny MS Kang | Immuneering |
| Stella Biderman | EleutherAI, Booz Allen Hamilton |
| Matthias Samwald | Institute of Artificial Intelligence, Medical University of Vienna |
| Stephen H. Bach | Brown University |
| Wojciech Kusa | TU Wien |
| Samuel Cahyawijaya | The Hong Kong University of Science and Technology |
| Fabio Barth | Humboldt-Universität zu Berlin |
| Simon Ott | Institute of Artificial Intelligence, Medical University of Vienna |
| Mario Sänger | Humboldt-Universität zu Berlin |
| Bo Wang | Massachusetts General Hospital |
| Alison Callahan | Stanford University |
| Daniel León Periñán | TU Dresden |
| Théo Gigant | Independent researcher |
| Patrick Haller | Humboldt-Universität zu Berlin |
| Jenny Chim | Queen Mary University of London |
| Jose Posada | Universidad del Norte |
| John Giorgi | University of Toronto |
| Karthik Rangasai Sivaraman | BITS Pilani |
| Marc Pàmies | Barcelona Supercomputing Center |
| Marianna Nezhurina | Kuban State University Of Technology |
| Robert Martin | Humboldt-Universität zu Berlin |
| Michael Cullan | Independent researcher |
| Moritz Freidank | Visium SA |
| Nathan Dahlberg | Independent researcher |
| Shubhanshu Mishra | shubhanshu.com |
| Shamik Bose | Independent researcher |
| Nicholas Broad | Hugging Face |
| Yanis Labrak | Avignon University |
| Shlok S Deshmukh | Elucidata, Inc. |
| Sid Kiblawi | Microsoft |
| Ayush Singh | Independent researcher |
| Minh Chien Vu | Detomo Inc. |
| Trishala Neeraj | Cornell University |
| Jonas Golde | Humboldt-Universität zu Berlin |
| Albert Villanova del Moral | Hugging Face |
| Benjamin Beilharz | TU Darmstadt |

## C   Task Schema and Harmonization

We have defined a set of lightweight, task-specific schema to help simplify programmatic access to common biomedical datasets.

Each dataset loader implemented in BIGBIO provides at least one `source` view of the dataset and at least one `bigbio` view of the dataset. The `source` view attempts to capture the original form of the dataset with as little change as possible. The `bigbio` view attempts to normalize the dataset into one of our BIGBIO task-specific schemas. All schemas are defined by creating an instance of the `datasets.Features` class from the Hugging Face datasets package.

Every element of the BIGBIO schemas has an `id` attribute that is unique across the dataset. In some datasets, entities are represented as discontiguous spans. For example, the string "estrogen and progesterone receptor positive" could be labeled with two entities and two lists of character offsets,

```
["estrogen", "receptor"]; [(0,8), (26,34)]
["progesterone receptor"]; [(13, 34)]
```

To support these types of annotations and maintain consistency, we represent all text-offset combinations this way.

### C.1   Schema Definitions

**Knowledge Base (KB)**   The knowledge base schema covers entity based tasks and includes named entity recognition (NER), named entity disambiguation/normalization (NED), event extraction (EE), relation extraction (RE), and coreference resolution (COREF). The schema is loosely based on the XML BioC format [10] and the brat annotation format [43]. The top level features are,

```
{
    "id": datasets.Value("string"),
    "document_id": datasets.Value("string"),
    "passages": [],
    "entities": [],
    "events": [],
    "coreferences": [],
    "relations": [],
}
```

The `id` attribute can be set to anything that makes it unique and the `document_id` attribute represents any identifying value included in the original dataset. Passages capture the text content of a sample. A single sample can have one passage (such as a single abstract) or multiple elements (such as abstract and title). The character offsets in the rest of the KB schema elements index into the string that would be created by joining all the passage texts.

```
"passages": [
    {
        "id": datasets.Value("string"),
        "type": datasets.Value("string"),
        "text": datasets.Sequence(datasets.Value("string")),
        "offsets": datasets.Sequence([datasets.Value("int32")]),
    }
]
```

Entities can be associated with a type as well as multiple database entries.

```
"entities": [
    {
        "id": datasets.Value("string"),
        "type": datasets.Value("string"),
        "text": datasets.Sequence(datasets.Value("string")),
        "offsets": datasets.Sequence([datasets.Value("int32")]),
```

```
        "normalized": [
            {
                "db_name": datasets.Value("string"),
                "db_id": datasets.Value("string"),
            }
        ],
    }
]
```

Events are modeled in BIGBIO as they are in the brat annotation tool.

```
"events": [
    {
        "id": datasets.Value("string"),
        "type": datasets.Value("string"),
        "trigger": {
            "text": datasets.Sequence(datasets.Value("string")),
            "offsets": datasets.Sequence([datasets.Value("int32")]),
        },
        "arguments": [
            {
                "role": datasets.Value("string"),
                "ref_id": datasets.Value("string"),
            }
        ],
    }
]
```

Coreference annotations can be specified using a sequence of entity IDs.

```
"coreferences": [
    {
        "id": datasets.Value("string"),
        "entity_ids": datasets.Sequence(datasets.Value("string")),
    }
]
```

Binary typed relations with multiple database normalizations are also supported.

```
"relations": [
    {
        "id": datasets.Value("string"),
        "type": datasets.Value("string"),
        "arg1_id": datasets.Value("string"),
        "arg2_id": datasets.Value("string"),
        "normalized": [
            {
                "db_name": datasets.Value("string"),
                "db_id": datasets.Value("string"),
            }
        ],
    }
]
```

**Question Answering (QA)**  The QA schema supports several question answering tasks. The `type` attribute is not constrained but takes the values "factoid", "how", "list", "multiple_choice", "summary", "why", and "yesno" in the current BIGBIO datasets. For "multiple_choice" and "yesno" questions, the `choices` attribute is populated with valid answers. The `context` attribute is used for closed-domain QA.

```
{
    "id": datasets.Value("string"),
    "question_id": datasets.Value("string"),
    "document_id": datasets.Value("string"),
    "question": datasets.Value("string"),
    "type": datasets.Value("string"),
    "choices": [datasets.Value("string")],
    "context": datasets.Value("string"),
    "answer": datasets.Sequence(datasets.Value("string")),
}
```

**Textual Entailment (TE)**   The TE schema supports tasks in which two text spans can be mapped onto the triplet of entailment labels ("entailment", "neutral", "contradict").

```
{
    "id": datasets.Value("string"),
    "premise": datasets.Value("string"),
    "hypothesis": datasets.Value("string"),
    "label": datasets.Value("string"),
}
```

**Text (TEXT)**   The TEXT schema supports tasks with a single text span and one or more associated labels (TXTCLASS).

```
{
    "id": datasets.Value("string"),
    "document_id": datasets.Value("string"),
    "text": datasets.Value("string"),
    "labels": [datasets.Value("string")],
}
```

**Text Pairs (PAIRS)**   The PAIRS schema supports tasks with two text spans and one label. In this initial release, the only task using this schema is semantic similarity (STS).

```
{
    "id": datasets.Value("string"),
    "document_id": datasets.Value("string"),
    "text_1": datasets.Value("string"),
    "text_2": datasets.Value("string"),
    "label": datasets.Value("string"),
}
```

**Text to Text (T2T)**   The T2T schema supports sequence to sequence tasks such as paraphasing (PARA), translation (TRANSL), and summarization (SUM).

```
{
    "id": datasets.Value("string"),
    "document_id": datasets.Value("string"),
    "text_1": datasets.Value("string"),
    "text_2": datasets.Value("string"),
    "text_1_name": datasets.Value("string"),
    "text_2_name": datasets.Value("string"),
}
```

## C.2   Harmonization

Harmonization efforts aimed for the simplest schema, per task, that was able to flexibly cover the majority of relevant features. We found in the majority of cases, the schema provided suited the task of the original dataset. Toward that end, we found that only 22% (29/129 datasets submitted) of

the datasets required major refactors (defined by significant changes or fixes to the dataloader post submission). While the schema satisfied most cases, we noted some areas of improvement below:

**Extension of question answering**  Question-answering supports multiple choice, binary choice, or span-based answers, but does not enable 'long-form' responses that may provide greater context to the question asked. This particular issue arose in PubMedQA, of which the source schema has a context key that provides framing for the answer.

**Extension of text pairs classification**  The text-pairs schema enables a relationship between two input texts and their corresponding labels. However, in at least one dataset (Scielo), a three-language translation was provided. This can be handled be implementing the dataset twice, one for each translation, or omitting this feature altogether.

**Multi-label entities**  Several datasets had multiple labels associated to a single entity. While we have adapted the schema to associate multiple labels to a single entity. To resolve this concern, we duplicate the feature but change the label and provide a new unique id. This concern was particularly noted in the `MedMentions` dataset.

**Diverse label representations**  For classification problems, the labels associated to a feature may be a string answer, or a numerical score. To maintain a consistent format across all datasets, label keys across schemas in the BIGBIO-view are always `str` types. This limitation affected at least 4 datasets (UMNSRS, MayoSRS, BioSimVerb), particulary in the context of semantic similarity scores across text. For the user to appropriately cast the score type, they would need familiarity of the dataset. We opted to enable the source view to represent label information for scores as floats when present.

**Unsupported task types**  In certain cases, tasks may extend beyond the descriptive capacity of the provided BIGBIO-schemas. For example, tasks that explicitly required contextualization were unable to fit into a pre-existing schema. For example, speech-based tasks, such as MedDialogue require a text, label, and potential context; the BIGBIO-text classification schema does not enable a context key. Additionally, Ask-a-Patient required a tuple-like structure to represent a text, a social media response, and a medical concept to be relevant to the task. In addition to tasks that require context, part-of-speech tagging or annotations on a per-token basis was not easily represented in our pre-existing schema.

During the initiative, common themes of recurring problems in biomedical NLP processing occurred. We denote them as follows:

**Issues with offsets**  One of the unit-tests specifically monitored whether reported features matched offsets provided from the original dataset. We found a several datasets with slight offset errors, or inconsistencies. In several cases, offset errors included off-by-one or whitespacing considerations, discontiguous spans, and one case, entirely omitted from the original dataset.

**Large datasets**  Several datasets possessed corpora that were large in size (upwards of 20 GB). In at least one instance, the initial implementation of the dataset yielded examples exceedingly slow. While we standardized information content, we did not explicitly optimize for efficiency.

# D Dataset Metadata

We collected the structured metadata outlined in Table 6 for all datasets in the BIGBIO catalog. Required elements are written as code in the data loader. Figures 5 and 6 show treemap visualizations of all datasets based on their license and language respectively.

Table 6: Metadata collected for all datasets.

| Field | Required | Description |
|---|---|---|
| Name | ✓ | Dataset name |
| Task Types | ✓ | NER, question answering, coreference resolution, etc. |
| Domain | ✓ | Corpora domain: biomedical or clinical/health-related |
| PubMed/PMC | ✓ | Corpora are from PubMed/PubMed Central (PMC) |
| Splits | ✓ | Canonical definitions for training/validation/testing splits |
| Publication | ✓ | Manuscript describing dataset |
| Year | | Publication year |
| Homepage | ✓ | Website describing dataset |
| Public URL | ✓ | Open URL (no authentication) |
| Private | ✓ | Requires authentication/credentialing |
| License | ✓ | Provided license type |
| Languages | ✓ | Included languages |
| Multilingual | | Parallel corpora |
| Annotation Source | | Expert label provenance (e.g., hand labeled, silver labels) |

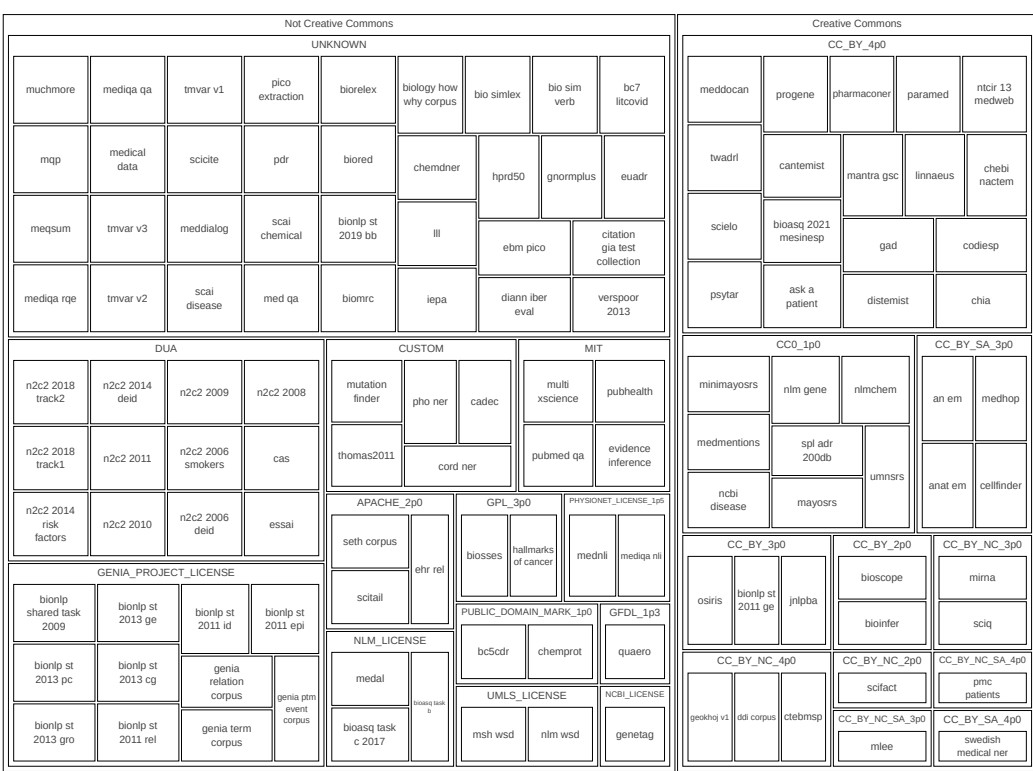

Figure 5: Treemap visualization of datasets by license.

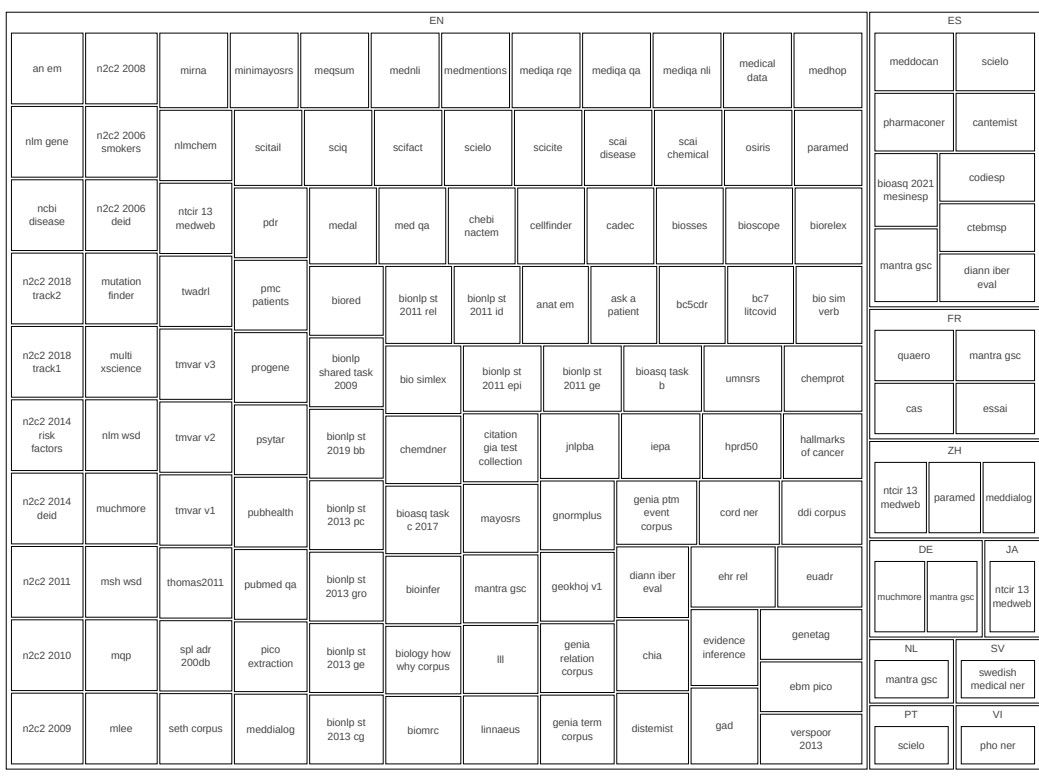

Figure 6: Treemap visualization of datasets by language.

# E   Unit Tests

We developed 11 unit tests to check the BIGBIOversions of all implemented data loaders. Unit tests run on all BIGBIO *configurations* (i.e., a schema view of the dataset) found within a dataset, whether they represent different dataset subsets or different tasks.

Among all implemented unit tests, we differentiate between **global** and **task-specific** tests. For datasets that support configurations with multiple schemas (each supporting different tasks), we run the task-specific tests using only the configuration supporting the task.

Below, we describe each unit test found in BIGBIO:

## E.1   Global Tests

1. **Metadata** Checks if the dataloader module provides relevant metadata attributes. Supported attributes include LANGUAGE (language of the dataset), LOCAL (whether the dataset is publicly accessible or requires local files), PUBMED (is part of Pubmed), and LICENSE (type of license). The LANGUAGE and LICENSE are standardized to common labels across datasets, whereas LOCAL and PUBMED are boolean.

2. **Unique Global IDs** Each element within a dataset is assigned a string ID that is unique across the dataset split (such as train, validation or test). For example, all passages, entities, relations, questions, labels, and other attributes will be assigned a unique string. This ID can be used to reference a given element if it is being used in a new context without considering explicit text overlap or other heuristics. This unit-test confirms that a every element has an ID that is unique across the full dataset split.

3. **Schema** This test checks whether the populated fields in the examples are consistent with the tasks supported by the dataset. For instance, if a dataset is annotated to support NER but there is not a single entity field populated across a full dataset split, the test will fail. Additionally, the test will provide a warning if fields are populated that would support a task missing from the annotated supported tasks. The loading procedure in Hugging Face's datasets fails if a dataloader does not adhere to its defined schema. Thus, we implicitly check for consistency between data and schema by loading the dataset.

4. **Feature Statistics** This test prints statistics of populated fields in the dataset to allow the user to manually check their plausibility. For each data split, it collects the number of elements (e.g. number of entities, relations, text pairs, etc.). We use these statistics for quality control by manually comparing to the dataset statistics reported in the publication describing the respective dataset.

## E.2   Task-specific Tests: Knowledge Base

1. **Referenced ids** Certain fields may be referenced by other elements (for example, a relation usually references two entities). References in the BIGBIO-schema will use the unique ID assigned to them. This unit test checks if all referenced IDs exist, and have an appropriate type. For instance, it makes sure that the arguments of a relation are indeed entities (and not relations or events).

2. **Passage Offsets** This test checks whether the start and end indices of all passages are correct. This is achieved by comparing the text span defined by the indices to the text field assigned to the passage. Additionally, the unit test will make sure that each passage is contiguous and does not overlap.

3. **Entity Offsets** This test makes sure that the start and end indices of entities are correct. Analogous to the *Passage Offsets* test, we compare the reported feature text for entities versus the extracted text from the start/ending index provided from the data. This test does not provide an explicit failure, but instead warns the user of all entities that do not explicitly match their offset-extracted text. We chose a warning over failure because some datasets contain faulty offsets in the original formats due to annotation errors.

4. **Event Offsets** Similar to the passage-offsets and entities-offset check, we compare the reported event text feature to the extracted text from provided offsets. We warn the user of any instances of discordance between the reported and extracted text.

5. **Multi-label Entities** The current BIGBIO schema does not support multiple types for entities. This test flags instances where an entity is assigned multiple types by concatenating the types with common connector symbols (such as '|' or ';').

6. **Multi-label Types** This unit-test performs the same check as `Multi-label Entities` for other features with the `type` attribute (passages, relations, events). This test is distinct from the multi-label entities test, because the envisioned BIGBIO schema revision to support multiple labels is different in this case.

### E.3   Task-specific Tests: Question Answering

1. **Multiple Choice** This test checks whether the answers of a question-answering schema are either multiple choice or binary (yes/no). It verifies that the answer provided exists in the choices available for each example.

All accepted data-loading scripts must pass code review, unit-tests, and implement explicit fixes for warnings that indicated destructive transformations of the original dataset (such as introducing faulty offsets).

In general, participants who implemented data-loading scripts were asked to refrain from resolving dataset issues in the dataloader for the original dataset but were free to fix the issues for the BIG-BIO versions. Any data quality changes were explicitly annotated within the review process, and the data loading script itself.

Certain datasets may require specific keys to be ignored. We implemented functions that allow a user to bypass a specific key (e.g., skip all events), a data split (e.g., skip the validation set), or a specific key within a dataset (e.g., skip relation labels in the test set). These functions were used to check the `BioNLP shared task` datasets, as the test splits of these datasets omitted annotations for some supported tasks. These bypass functions allow a user to test if all other aspects of the dataset implementation work as intended.

## F Dataset Submission Checklist

- ☐ Confirm that this PR is linked to the dataset issue.
- ☐ Create the dataloader script `biodatasets/my_dataset/my_dataset.py` (please use only lowercase and underscore for dataset naming).
- ☐ Provide values for
    - ☐ `_CITATION`
    - ☐ `_DATASETNAME`
    - ☐ `_DESCRIPTION`
    - ☐ `_HOMEPAGE`
    - ☐ `_LICENSE`
    - ☐ `_URLs`
    - ☐ `_SUPPORTED_TASKS`
    - ☐ `_SOURCE_VERSION`
    - ☐ `_BIGBIO_VERSION`
- ☐ Data loader implementations for
    - ☐ `_info()`
    - ☐ `_split_generators()`
    - ☐ `_generate_examples()`
- ☐ Make sure that the `BUILDER_CONFIGS` class attribute is a list with at least one 'BigBioConfig' for the source schema and one for a bigbio schema.
- ☐ Confirm dataloader script works with `datasets.load_dataset` function.
- ☐ Confirm that your dataloader script passes the test suite run with
  `python -m tests.test_bigbio biodatasets/my_dataset/my_dataset.py`.
- ☐ If my dataset is local, I have provided an output of the unit-tests in the PR (please copy paste). This is OPTIONAL for public datasets, as we can test these without access to the data files.

# G   BigScience Biomedical Hackathon

We catalogued an initial set of 174 datasets and prior to launching the hackathon, we provided users with a project board that tagged each dataset as a new issue within our GitHub repository. For all datasets, we provided meta-data tags such as language, license, and associated task (e.g., NER, question answering). Participants could assign themselves to a dataset via issues and status would be reflected in the project board (see Figure 7). Admins could change the status of the issue based on progress of the data loading script.

Figure 7: Participants volunteered to implement dataset loaders using GitHub project tracking tools.

Participants were asked to create a fork of the repository, and implement their data-loading script. We provided a template of a dataloading script, where explicit comments were left to indicate key functions and attributes the participant must complete. For datasets in common formats like BRAT or BioC, we provided utility functions to improve standardization across formats. At minimum, participants implemented an `_info_` function that instantiated the `source` and `bigbio` configs. A `_split_generators` function that identified how to access each data split in the dataset, and the `_generate_examples` that extracted relevant information from each data split according to the specifications of the configs.

Dataloader scripts were submitted through pull-requests (PRs) on GitHub. Prior to submitting code for review, we asked participants to check if the code passed unit-tests and style guidelines. Accepted PRs required at least 1 admin approval to merge to the library. To respect data governance, we did not accept any submissions that provided explicit dataset files. Dataloading scripts must access datasets via URLs, or expect a filepath to the local dataset.

If a dataset had multiple tasks, we asked the participant to implement tasks based on the number of unique schemas, if possible. Some datasets possess different views based on the different tasks that can be performed on them. Participants were told to handle multiple annotations/harmonization per the original dataset's recommendations. If none were given, participants were asked to choose what seemed reasonable, and iterate with an admin.

All contribution instructions may be found here.

Of the 174 datasets identified, 126 datasets satisfied the acceptance criteria, including the checklist in §F, code-review, and passing unit-tests. Exceptions were made on a case-by-case basis for datasets with unique challenges that extended beyond the scope of the schema provided.

## G.1   Frequently Asked Questions (FAQ)

During the hackthon, we developed the following list of frequently asked questions (FAQ).

**How can I find the appropriate license for my dataset?** The license for a dataset is not always obvious. Here are some strategies to try in your search:

1. Check the Experiment A: Annotated Datasets sheet of the we used while planning the hackathon

2. Check for files such as README or LICENSE that may be distributed with the dataset itself

3. Check the dataset webpage

4. Check publications that announce the release of the dataset

5. Check the website of the organization providing the dataset

If no official license is listed anywhere, but you find a webpage that describes general data usage policies for the dataset, you can fall back to providing that URL in the `_LICENSE` variable. If you can't find any license information, please make a note in your PR and put `_LICENSE = "Unknown"` in your dataset script.

**What if my dataset is not publicly available?** We understand that some biomedical datasets are not publicly available due to data usage agreements or licensing. For these datasets, we recommend implementing a dataloader script that references a local directory containing the dataset. You can find examples in the `n2c2_2011` and `bioasq` implementations. There are also local dataset specific instructions in template.

**What types of libraries can we import?** Eventually, your dataloader script will need to run using only the packages supplied by the datasets package. If you find a well supported package that makes your implementation easier (e.g. bioc), then feel free to use it.

We will address the specifics during review of your PR to the BigScience biomedical repo and find a way to make it usable in the final submission to huggingface bigscience-biomedical

**Can I upload my dataset anywhere?** No. Please don't upload the dataset you're working on to the huggingface hub or anywhere else. This is not the goal of the hackathon and some datasets have licensing agreements that prevent redistribution. If the dataset is public, include a downloading component in your dataset loader script. Otherwise, include only an "extraction from local files" component in your dataset loader script. If you have a custom dataset you would like to submit, please make an issue and an admin will get back to you.

**My dataset supports multiple tasks with different bigbio schemas. What should I do?** In some cases, a single dataset will support multiple tasks with different bigbio schemas. For example, the muchmore dataset can be used for a translation task (supported by the Text to Text (T2T) schema) and a named entity recognition task (supported by the Knowledge Base (KB) schema). In this case, please implement one config for each supported schema and name the config `<datasetname>_bigbio_<schema>`. In the muchmore example, this would mean one config called `muchmore_bigbio_t2t` and one config called `muchmore_bigbio_kb`.

**My dataset comes with multiple annotations per text and no/multiple harmonizations. How should I proceed?** Please implement all different annotations and harmonizations as source versions (see `examples/bioasq.py` for an example). If the authors suggest a preferred harmonization, use that for the bigbio version. Otherwise use the harmonization that you think is best.

**How should I handle offsets and text in the bigbio schema?** Full details on how to handle offsets and text in the bigbio kb schema can be found in the schema documentation.

**My dataset is complicated, can you help me?** Yes! Please feel free to leave a question in questions or ping the admins directly with @admins. We will be hosting office hours round the clock to be able to answer you in a timely manner!

**My dataset is too complicated, can I switch?** Yes! Some datasets are easier to write dataloader scripts for than others. If you find yourself working on a dataset that you can not make progress on, please make a comment in the associated issue, asked to be un-assigned from the issue, and start the search for a new unclaimed dataset. You are also welcome to ping the admins - we are happy to help you!

**Can I change the Big-Bio schema?** No, please do not modify the Big-Bio Schema. The goal of this hackathon is to enable simple, programmatic access to a large variety of biomedical datasets. Part of this requires having a dependable interface. We developed our schema to address the most salient types of questions to ask of the datasets. We would be more than happy to discuss your suggestions, and you are welcome to implement it as a new config.

**My dataset has multiple labels to a span of text - what do I do?** In many of our schemas, we have a 1:1 mapping between a key and its label (i.e. in KB, entity and label). In some datasets, we've noticed that there are multiple labels assigned to a text entity. Generally speaking, if a big-bio key has multiple labels associated with it, please populate the list with multiple instances of (key, label) according to each label that correspond to it.

So for instance if the dataset has an entity "copper" with the types "Pharmacologic Substance" and "Biologically Active", please create one entity with type "Pharmacologic Substance" and an associated unique id and another entity with type "Biologically Active" with a different unique id. The rest of the inputs (text, offsets, and normalization) of both entities will be identical.

**What happens after I claim a dataset?** In order to keep turnaround time reasonable, and ensure datasets are being completed, we propose a few notes on claiming a dataset:

1. Please claim a dataset only if you intend to work on it. We'll try to check in within 3 days to ensure you have the help you need. Don't hesitate to contact the admins! We are ready to help!

2. If you have already claimed a dataset prior to (2022/04/05), we will check in on Friday (2022/04/08). If we do not hear back via GitHub issues OR a message to the Discord admins on general, we will make the dataset open for other participants by Saturday (2022/04/09).

3. If things are taking longer than expected - that is totally ok! Please let us know via GitHub issues (preferred) or by pinging the @admins channel on Discord.

## H Assessing Dataset Overlap for De-Duplication

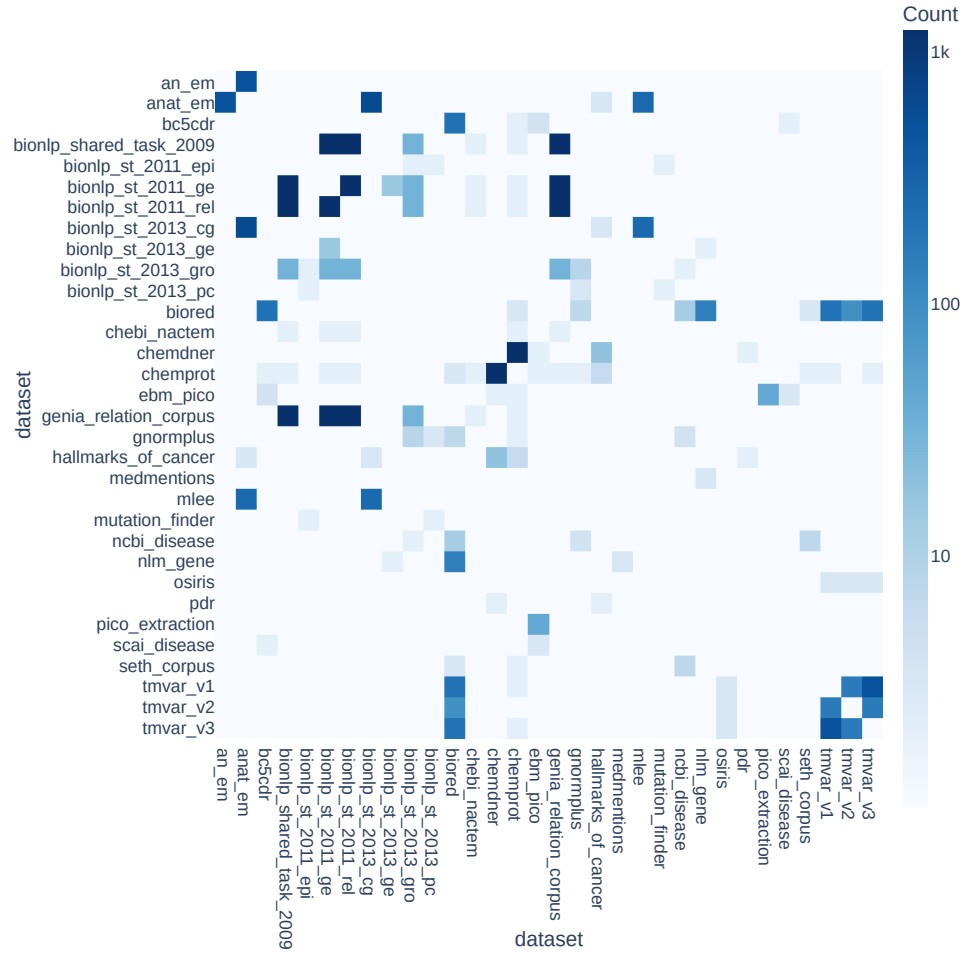

Figure 8: A heatmap representation of PubMed overlap between public datasets in BIGBIO. Each cell is shaded using the log count of PMIDs shared by the pair of datasets it represents.

Table 7: Example document IDs as they appear in the original source datasets and their corresponding BIGBIO normalization to PubMed PMIDs, Pubmed Central PMCIDs, and journal titles.

| Original Document ID | PMID | PMCID | Journal |
|---|---|---|---|
| PMID-12604762 | 12604762 | PMC1497507 | Public Health Rep |
| BB-kb+ner-F-25496341-000 | 25496341 | PMC4320590 | BMC Genomics |
| 17389645_04_discussion | 17389645 | PMC1885650 | Nucleic Acids Res |
| pmcA2538543 | 2538543 | PMC2189270 | J Exp Med |
| 10747015-3 | 10747015 | PMC310216 | EMBO J |
| 6421395:4 | 6421395 | PMC1444356 | Br Med J (Clin Res Ed) |
| PMC2885601-03-RESULTS-01 | 20556207 | PMC2885601 | Open Microbiol J |
| PMC-2626671-01-INTRODUCTION | 19139168 | PMC2626671 | J Exp Med |

As biomedical models are trained and evaluated on ever larger meta-datasets, it is important to characterize duplication within and between datasets. This can take the form of direct train/test leakage [13] or more subtle issues of near-duplicates and repeated substrings which can negatively impact performance and training time of language models [25]. In biomedical NLP, annotation efforts often build upon existing datasets meaning meta-dataset curation needs to take additional steps to mitigate possible train/test leakage. To assess the magnitude of this phenomena across the

Table 8: Dataset clusters of document (PMID) overlap.

| Dataset Names | Count | PMID Overlap |
|---|---|---|
| BioRED, NCBI Disease | 2 | 11 |
| MLEE, AnatEM | 2 | 12 |
| Hallmarks of Cancer, CHEMDNER | 2 | 12 |
| BioNLP ST 2013 GE, BioNLP ST 2011 GE | 2 | 14 |
| BioNLP ST 2011 REL, BioNLP ST 2013 GRO, GENIA Relation Corpus, BioNLP Shared Task 2009, BioNLP ST 2011 GE | 5 | 29 |
| PICO Extraction, EBM PICO | 2 | 41 |
| tmVar v1, tmVar v2, tmVar v3 | 3 | 69 |
| BioRED, tmVar v1, tmVar v2, tmVar v3 | 4 | 87 |
| BioRED, tmVar v1, tmVar v3 | 3 | 109 |
| NLM Gene, BioRED | 2 | 140 |
| BC5CDR, BioRED | 2 | 203 |
| tmVar v1, tmVar v3 | 2 | 232 |
| MLEE, BioNLP ST 2013 CG, AnatEM | 3 | 250 |
| BioNLP ST 2013 CG, AnatEM | 2 | 348 |
| AnatEM, AnEM | 2 | 492 |
| GENIA Relation Corpus, BioNLP Shared Task 2009, BioNLP ST 2011 REL, BioNLP ST 2011 GE | 4 | 1179 |
| ChemProt, CHEMDNER | 2 | 1199 |

BIGBIO corpus, we conducted a preliminary analysis counting the number of shared documents across all annotated datasets sourced from PubMed or PubMed Central (PMC).

**PubMed Document ID Normalization** PubMed/PMC provides uniform identifiers for documents: PubMed PMID and PubMed Central PMCID. However, many datasets encode this document information using inconsistent formats as shown in Table 7. We wrote a normalization function to standardize all document identifiers to facilitate joins with other PubMed/PMC datsets. We then joined this data with the `PMC-ids.csv.gz` file available from the National Library of Medicine[1].

**PubMed Dataset Overlap Analysis** Our normalizations of PMIDs allowed us to calculate which PubMed articles were used in multiple datasets. In Table 8 we show the largest PMID clusters, i.e., sets of datasets that contain the same documents. In Figure 8 we visualize this overlap as a heatmap. We observe several cases of clear dataset iteration (e.g., tmVar v1-v3, AnEM to AnatEM) and NLP challenges building on the same source datasets (BioNLP shared tasks 2009 and 2011 build on the GENIA Relation Corpus). BioRED illustrates another common pattern, where documents were sampled from 5 existing biomedical datasets before annotating [121].

---

[1] https://ftp.ncbi.nlm.nih.gov/pub/pmc accessed May 29, 2022

# I  Data Visualization and Exploration

To highlight the efficiency of using consistent schema across datasets, we created a Streamlit[2] web application to allow anyone to browse through any schema-specific details and visualization for all supported datasets. The web application enables task sorting at the level of task schema (e.g., NER), which supports downstream approaches to use groups of datasets with minimal effort. Such as prompt based methods or multi-task learning (MTL).

For each split, we provide basic dataset details (like number of training samples, character counts, word counts, number of unique labels, etc.). Further, we also present distributions of token lengths and labels (or sub-component types) within each dataset to compare across splits. We used periods and new lines to break the text block into sentences, and tokenized each sentence by white space to count the token lengths. For datasets of tasks that do not have labels, which is the case for most common knowledge base construction and information extraction tasks, we analyze the data distribution across the sub-component types. For instance, our task schema for the BioCreative V Chemical Disease Relation (CDR) dataset [112] provides an efficient way to compare the distribution of chemical and disease entities across splits (See Figure 9).

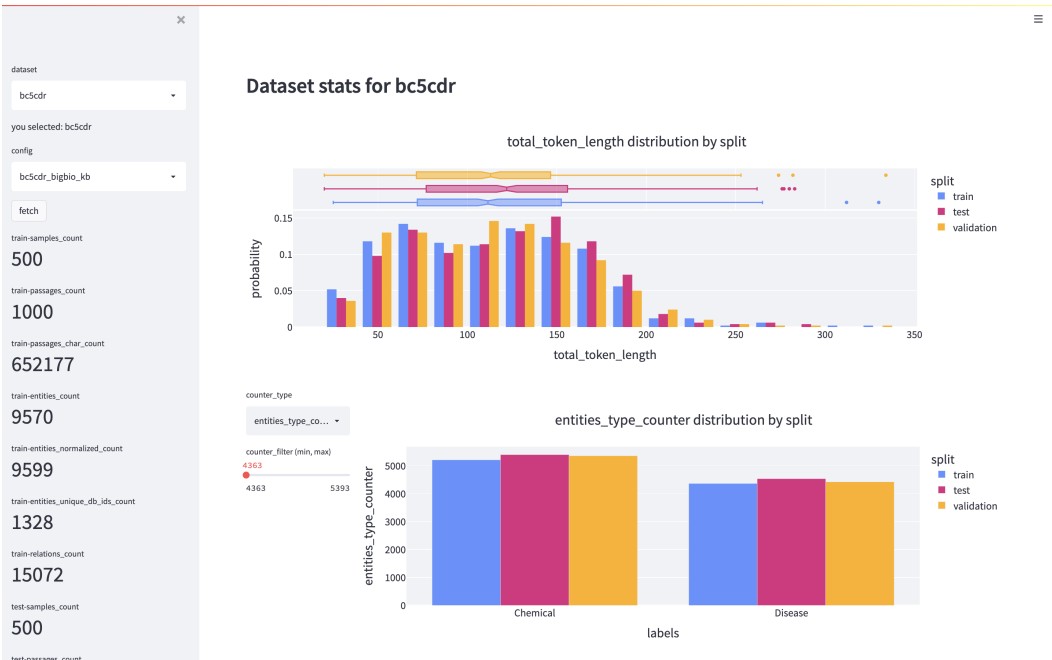

Figure 9: Streamlit web application for visualizing dataset-specific details and textual analysis at the span-level. Here we show plots for the BioCreative V Chemical Disease Relation (CDR) dataset.

---

[2]https://streamlit.io/

## J   Zero-shot Language Model Evaluation

### J.1   Expanded Results

Table 9 summarizes key properties of the language models used in our zero-shot experiments.

Table 9: Language model parameter counts and pretraining datasets.

| Language Model | Parameters | PubMed/PMC | Pretraining Dataset(s) |
|---|---|---|---|
| SciFive-Base | 220M | ✓ | C4, PubMed Abstracts, PubMed Central (PMC) |
| SciFive-Large | 770M | ✓ | C4, PubMed Abstracts, PubMed Central (PMC) |
| GPT-Neo-1.3B | 1.3B | ✓ | The Pile |
| GPT-2 | 1.5B | | WebText |
| GPT-J-6B | 6B | ✓ | The Pile |
| T0_3B | 3B | | C4, P3 (Public Pool of Prompts) |
| T5 v1.1-xxl | 11B | | C4 |
| T0 | 11B | | C4, P3 (Public Pool of Prompts) |
| T0+ | 11B | | C4, P3 (Public Pool of Prompts) |
| T0++ | 11B | | C4, P3 (Public Pool of Prompts) |
| GPT-NeoX-20B | 20B | ✓ | The Pile |
| OPT-66B | 66B | | RoBERTa (subsets: BookCorpus, Stories, CC-News v2), The Pile (subsets: Common-Crawl, DM Mathematics, Project Gutenberg, HackerNews, OpenSubtitles, OpenWebText2, USPTO and Wikipedia), PushShift.io Reddit |
| GPT-3 | 175B | | WebText, ??? |
| BLOOM | 176B | | Roots Corpus |

Tables 10 and 11 and contains complete zero-shot language model results pooled across all prompts by dataset.

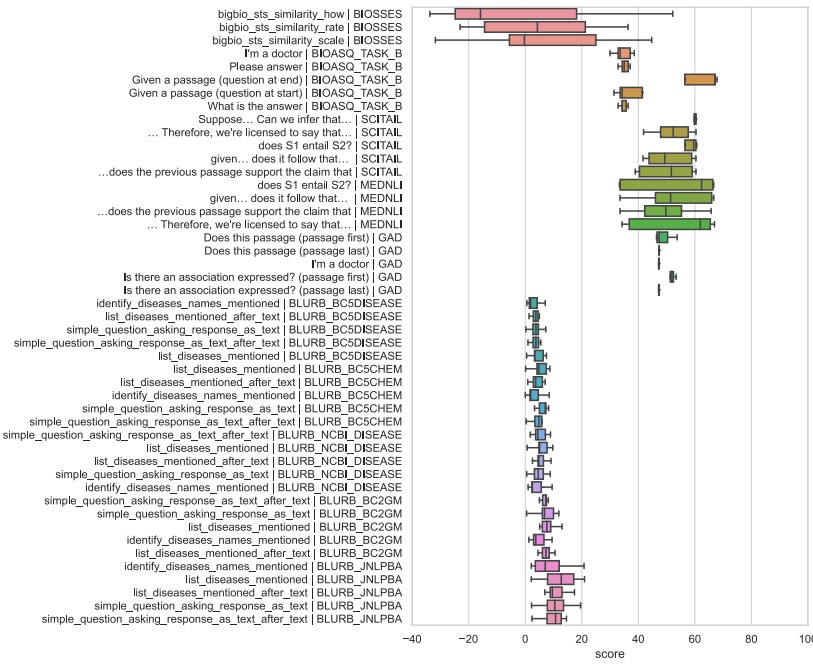

Figure 10: Per-prompt scores (x-axis) for all non-instruction tuned language models (SciFive, GPT-Neo-1.3B, GPT-2, GPT-J-6B, T5 v1.1-xxl, OPT-66B, BLOOM). Prompt template and dataset names are on the y-axis.

Table 10: Summary results across all non-NER tasks.

| Model | Dataset | Metric | Mean | SE | Min | Max |
|---|---|---|---|---|---|---|
| SciFive-Base | bigbio_biosses | pearson | 17.9 | 17.4 | -4.7 | 52.2 |
| SciFive-Base | bigbio_bioasq_task_b | accuracy | 35.9 | 1.5 | 32.9 | 41.4 |
| SciFive-Base | bigbio_scitail | accuracy | 60.1 | 0.3 | 59.1 | 60.4 |
| SciFive-Base | bigbio_mednli | accuracy | 63.7 | 2.8 | 55.3 | 66.8 |
| SciFive-Base | bigbio_gad | accuracy | 51.9 | 1.3 | 46.8 | 53.7 |
| SciFive-Large | bigbio_biosses | pearson | 15.3 | 29.5 | -14.2 | 44.7 |
| SciFive-Large | bigbio_bioasq_task_b | accuracy | 30.7 | 0.6 | 29.3 | 32.9 |
| SciFive-Large | bigbio_scitail | accuracy | 59.8 | 0.5 | 57.7 | 60.4 |
| SciFive-Large | bigbio_mednli | accuracy | 61.6 | 4.8 | 47.3 | 66.7 |
| SciFive-Large | bigbio_gad | accuracy | 47.4 | 0.0 | 47.4 | 47.4 |
| GPT-Neo-1.3B | bigbio_biosses | pearson | 36.4 | NaN | 36.4 | 36.4 |
| GPT-Neo-1.3B | bigbio_bioasq_task_b | accuracy | 43.9 | 5.9 | 35.7 | 67.1 |
| GPT-Neo-1.3B | bigbio_scitail | accuracy | 50.7 | 4.1 | 38.9 | 60.4 |
| GPT-Neo-1.3B | bigbio_mednli | accuracy | 43.0 | 7.1 | 33.5 | 64.0 |
| GPT-Neo-1.3B | bigbio_gad | accuracy | 47.9 | 0.9 | 46.4 | 51.3 |
| GPT-2 | bigbio_biosses | pearson | 10.9 | 8.6 | 2.4 | 19.5 |
| GPT-2 | bigbio_bioasq_task_b | accuracy | 38.4 | 5.4 | 32.9 | 60.0 |
| GPT-2 | bigbio_scitail | accuracy | 50.4 | 4.3 | 39.6 | 60.4 |
| GPT-2 | bigbio_mednli | accuracy | 54.1 | 7.3 | 33.4 | 66.0 |
| GPT-2 | bigbio_gad | accuracy | 47.4 | 0.0 | 47.4 | 47.6 |
| GPT-J-6B | bigbio_biosses | pearson | 0.2 | 31.9 | -31.8 | 32.1 |
| GPT-J-6B | bigbio_bioasq_task_b | accuracy | 40.6 | 6.6 | 33.6 | 67.1 |
| GPT-J-6B | bigbio_scitail | accuracy | 51.6 | 4.4 | 40.4 | 60.3 |
| GPT-J-6B | bigbio_mednli | accuracy | 46.3 | 6.0 | 34.2 | 62.8 |
| GPT-J-6B | bigbio_gad | accuracy | 48.2 | 1.0 | 46.6 | 52.1 |
| T0_3B | bigbio_biosses | pearson | 0.9 | NaN | 0.9 | 0.9 |
| T0_3B | bigbio_bioasq_task_b | accuracy | 63.9 | 2.4 | 58.6 | 72.9 |
| T0_3B | bigbio_scitail | accuracy | 69.9 | 7.4 | 46.0 | 84.5 |
| T0_3B | bigbio_mednli | accuracy | 64.6 | 7.9 | 41.4 | 76.2 |
| T0_3B | bigbio_gad | accuracy | 47.7 | 0.3 | 47.4 | 48.7 |
| T5 v1.1-xxl | bigbio_biosses | pearson | -6.1 | 19.1 | -33.7 | 30.5 |
| T5 v1.1-xxl | bigbio_bioasq_task_b | accuracy | 64.9 | 2.1 | 56.4 | 67.9 |
| T5 v1.1-xxl | bigbio_scitail | accuracy | 54.1 | 4.2 | 39.6 | 60.7 |
| T5 v1.1-xxl | bigbio_mednli | accuracy | 52.4 | 6.9 | 33.5 | 65.5 |
| T5 v1.1-xxl | bigbio_gad | accuracy | 52.1 | 0.5 | 50.2 | 52.6 |
| T0 | bigbio_biosses | pearson | 27.2 | 4.8 | 17.7 | 32.0 |
| T0 | bigbio_bioasq_task_b | accuracy | 86.4 | 0.9 | 84.3 | 89.3 |
| T0 | bigbio_scitail | accuracy | 72.3 | 7.5 | 49.0 | 88.6 |
| T0 | bigbio_mednli | accuracy | 68.8 | 7.6 | 46.1 | 78.4 |
| T0 | bigbio_gad | accuracy | 55.1 | 0.7 | 52.6 | 56.6 |
| T0+ | bigbio_biosses | pearson | 35.2 | 5.5 | 29.7 | 40.7 |
| T0+ | bigbio_bioasq_task_b | accuracy | 84.3 | 1.8 | 80.7 | 90.7 |
| T0+ | bigbio_scitail | accuracy | 71.0 | 7.6 | 46.8 | 87.9 |
| T0+ | bigbio_mednli | accuracy | 68.9 | 8.4 | 44.0 | 79.7 |
| T0+ | bigbio_gad | accuracy | 52.4 | 0.9 | 49.3 | 53.9 |
| T0++ | bigbio_biosses | pearson | 26.9 | NaN | 26.9 | 26.9 |
| T0++ | bigbio_bioasq_task_b | accuracy | 94.1 | 0.1 | 93.6 | 94.3 |
| T0++ | bigbio_scitail | accuracy | 71.6 | 5.6 | 57.0 | 87.0 |
| T0++ | bigbio_mednli | accuracy | 74.2 | 3.9 | 63.8 | 81.4 |
| T0++ | bigbio_gad | accuracy | 53.5 | 0.7 | 51.7 | 55.4 |
| GPT-NeoX-20B | bigbio_biosses | pearson | -14.8 | 8.3 | -23.0 | -6.5 |
| GPT-NeoX-20B | bigbio_bioasq_task_b | accuracy | 41.3 | 6.5 | 34.3 | 67.1 |
| GPT-NeoX-20B | bigbio_scitail | accuracy | 50.5 | 3.6 | 41.9 | 59.8 |
| GPT-NeoX-20B | bigbio_mednli | accuracy | 48.6 | 5.7 | 34.7 | 62.4 |
| GPT-NeoX-20B | bigbio_gad | accuracy | 47.9 | 0.9 | 46.4 | 51.3 |
| OPT-66B | bigbio_bioasq_task_b | accuracy | 43.0 | 6.2 | 35.7 | 67.9 |
| OPT-66B | bigbio_scitail | accuracy | 44.7 | 2.4 | 39.6 | 52.3 |
| OPT-66B | bigbio_mednli | accuracy | 38.1 | 3.6 | 33.3 | 48.6 |
| OPT-66B | bigbio_gad | accuracy | 48.3 | 1.1 | 46.6 | 52.4 |
| GPT-3 | bigbio_biosses | pearson | 47.3 | 9.4 | 32.0 | 64.5 |
| GPT-3 | bigbio_bioasq_task_b | accuracy | 73.0 | 5.8 | 55.7 | 92.1 |
| GPT-3 | bigbio_scitail | accuracy | 52.0 | 4.0 | 39.7 | 61.4 |
| GPT-3 | bigbio_gad | accuracy | 48.4 | 0.8 | 46.1 | 50.9 |
| BLOOM | bigbio_biosses | pearson | 0.5 | 9.7 | -15.8 | 17.7 |
| BLOOM | bigbio_bioasq_task_b | accuracy | 40.9 | 6.6 | 33.6 | 67.1 |
| BLOOM | bigbio_scitail | accuracy | 52.4 | 3.6 | 39.6 | 59.6 |
| BLOOM | bigbio_mednli | accuracy | 64.0 | 1.8 | 59.6 | 66.9 |
| BLOOM | bigbio_gad | accuracy | 48.8 | 1.0 | 47.2 | 51.9 |

Table 11: Summary results across all NER tasks.

| Model | Dataset | Metric | Mean | SE | Min | Max |
|---|---|---|---|---|---|---|
| SciFive-Base | bigbio_blurb_bc5chem | rouge1_fmeasure | 2.6 | 1.6 | 0.0 | 7.1 |
| SciFive-Base | bigbio_blurb_bc5disease | rouge1_fmeasure | 1.9 | 0.9 | 0.2 | 4.6 |
| SciFive-Base | bigbio_blurb_ncbi_disease | rouge1_fmeasure | 2.1 | 0.9 | 0.5 | 4.8 |
| SciFive-Base | bigbio_blurb_bc2gm | rouge1_fmeasure | 4.1 | 2.0 | 0.5 | 9.6 |
| SciFive-Base | bigbio_blurb_jnlpba | rouge1_fmeasure | 7.9 | 3.1 | 2.2 | 16.1 |
| SciFive-Large | bigbio_blurb_bc5chem | rouge1_fmeasure | 5.4 | 1.7 | 0.3 | 8.7 |
| SciFive-Large | bigbio_blurb_bc5disease | rouge1_fmeasure | 5.1 | 1.4 | 1.0 | 7.5 |
| SciFive-Large | bigbio_blurb_ncbi_disease | rouge1_fmeasure | 7.0 | 1.5 | 1.8 | 9.7 |
| SciFive-Large | bigbio_blurb_bc2gm | rouge1_fmeasure | 9.4 | 2.1 | 2.6 | 13.1 |
| SciFive-Large | bigbio_blurb_jnlpba | rouge1_fmeasure | 14.7 | 3.7 | 2.4 | 21.0 |
| GPT-Neo-1.3B | bigbio_blurb_bc5chem | rouge1_fmeasure | 5.6 | 1.0 | 1.9 | 7.2 |
| GPT-Neo-1.3B | bigbio_blurb_bc5disease | rouge1_fmeasure | 4.0 | 0.8 | 1.6 | 6.1 |
| GPT-Neo-1.3B | bigbio_blurb_ncbi_disease | rouge1_fmeasure | 6.0 | 1.3 | 2.3 | 8.6 |
| GPT-Neo-1.3B | bigbio_blurb_bc2gm | rouge1_fmeasure | 6.5 | 0.7 | 3.7 | 7.7 |
| GPT-Neo-1.3B | bigbio_blurb_jnlpba | rouge1_fmeasure | 11.4 | 1.6 | 7.6 | 16.9 |
| GPT-2 | bigbio_blurb_bc5chem | rouge1_fmeasure | 5.2 | 1.3 | 0.9 | 7.7 |
| GPT-2 | bigbio_blurb_bc5disease | rouge1_fmeasure | 4.4 | 1.1 | 1.4 | 6.6 |
| GPT-2 | bigbio_blurb_ncbi_disease | rouge1_fmeasure | 5.7 | 1.0 | 2.6 | 7.5 |
| GPT-2 | bigbio_blurb_bc2gm | rouge1_fmeasure | 8.3 | 1.1 | 4.5 | 10.3 |
| GPT-2 | bigbio_blurb_jnlpba | rouge1_fmeasure | 12.7 | 2.5 | 6.0 | 17.4 |
| GPT-J-6B | bigbio_blurb_bc5chem | rouge1_fmeasure | 4.9 | 0.9 | 1.9 | 7.3 |
| GPT-J-6B | bigbio_blurb_bc5disease | rouge1_fmeasure | 3.4 | 0.6 | 1.9 | 5.0 |
| GPT-J-6B | bigbio_blurb_ncbi_disease | rouge1_fmeasure | 5.0 | 1.2 | 2.5 | 9.1 |
| GPT-J-6B | bigbio_blurb_bc2gm | rouge1_fmeasure | 7.1 | 1.1 | 3.8 | 10.5 |
| GPT-J-6B | bigbio_blurb_jnlpba | rouge1_fmeasure | 10.9 | 2.0 | 6.7 | 17.4 |
| T0_3B | bigbio_blurb_bc5chem | rouge1_fmeasure | 38.6 | 1.0 | 36.1 | 41.2 |
| T0_3B | bigbio_blurb_bc5disease | rouge1_fmeasure | 23.1 | 1.4 | 19.3 | 26.5 |
| T0_3B | bigbio_blurb_ncbi_disease | rouge1_fmeasure | 28.5 | 1.7 | 25.4 | 34.0 |
| T0_3B | bigbio_blurb_bc2gm | rouge1_fmeasure | 22.0 | 0.4 | 21.3 | 23.4 |
| T0_3B | bigbio_blurb_jnlpba | rouge1_fmeasure | 16.7 | 3.3 | 7.7 | 23.7 |
| T5 v1.1-xxl | bigbio_blurb_bc5chem | rouge1_fmeasure | 3.3 | 0.3 | 2.0 | 3.7 |
| T5 v1.1-xxl | bigbio_blurb_bc5disease | rouge1_fmeasure | 3.1 | 0.5 | 1.3 | 3.8 |
| T5 v1.1-xxl | bigbio_blurb_ncbi_disease | rouge1_fmeasure | 4.3 | 0.5 | 2.4 | 5.6 |
| T5 v1.1-xxl | bigbio_blurb_bc2gm | rouge1_fmeasure | 5.0 | 0.5 | 3.3 | 5.9 |
| T5 v1.1-xxl | bigbio_blurb_jnlpba | rouge1_fmeasure | 7.2 | 1.5 | 2.1 | 10.0 |
| T0 | bigbio_blurb_bc5chem | rouge1_fmeasure | 46.5 | 3.1 | 42.6 | 58.8 |
| T0 | bigbio_blurb_bc5disease | rouge1_fmeasure | 30.8 | 4.5 | 22.5 | 43.9 |
| T0 | bigbio_blurb_ncbi_disease | rouge1_fmeasure | 38.6 | 4.9 | 29.9 | 56.0 |
| T0 | bigbio_blurb_bc2gm | rouge1_fmeasure | 24.0 | 1.5 | 21.1 | 29.5 |
| T0 | bigbio_blurb_jnlpba | rouge1_fmeasure | 16.2 | 3.4 | 7.7 | 23.5 |
| T0+ | bigbio_blurb_bc5chem | rouge1_fmeasure | 44.4 | 2.5 | 38.9 | 54.0 |
| T0+ | bigbio_blurb_bc5disease | rouge1_fmeasure | 29.4 | 3.2 | 24.3 | 40.7 |
| T0+ | bigbio_blurb_ncbi_disease | rouge1_fmeasure | 36.3 | 3.7 | 31.0 | 50.1 |
| T0+ | bigbio_blurb_bc2gm | rouge1_fmeasure | 25.0 | 0.6 | 23.9 | 27.1 |
| T0+ | bigbio_blurb_jnlpba | rouge1_fmeasure | 11.1 | 4.0 | 3.2 | 25.2 |
| T0++ | bigbio_blurb_bc5chem | rouge1_fmeasure | 43.1 | 1.7 | 39.0 | 49.1 |
| T0++ | bigbio_blurb_bc5disease | rouge1_fmeasure | 28.6 | 2.2 | 24.7 | 35.2 |
| T0++ | bigbio_blurb_ncbi_disease | rouge1_fmeasure | 36.2 | 3.0 | 31.4 | 47.7 |
| T0++ | bigbio_blurb_bc2gm | rouge1_fmeasure | 25.1 | 0.2 | 24.8 | 25.7 |
| T0++ | bigbio_blurb_jnlpba | rouge1_fmeasure | 13.3 | 2.3 | 7.5 | 19.7 |
| GPT-NeoX-20B | bigbio_blurb_bc5chem | rouge1_fmeasure | 5.7 | 1.6 | 1.5 | 10.3 |
| GPT-NeoX-20B | bigbio_blurb_bc5disease | rouge1_fmeasure | 3.5 | 0.6 | 2.1 | 5.5 |
| GPT-NeoX-20B | bigbio_blurb_ncbi_disease | rouge1_fmeasure | 5.5 | 0.9 | 4.2 | 8.9 |
| GPT-NeoX-20B | bigbio_blurb_bc2gm | rouge1_fmeasure | 7.0 | 1.2 | 2.6 | 9.9 |
| GPT-NeoX-20B | bigbio_blurb_jnlpba | rouge1_fmeasure | 8.9 | 1.7 | 2.7 | 13.0 |
| GPT-3 | bigbio_blurb_bc5chem | rouge1_fmeasure | 40.5 | 9.9 | 13.3 | 63.3 |
| GPT-3 | bigbio_blurb_bc5disease | rouge1_fmeasure | 36.9 | 9.8 | 12.8 | 60.8 |
| GPT-3 | bigbio_blurb_ncbi_disease | rouge1_fmeasure | 40.4 | 10.5 | 14.9 | 66.7 |
| GPT-3 | bigbio_blurb_bc2gm | rouge1_fmeasure | 39.1 | 10.7 | 14.7 | 64.5 |
| GPT-3 | bigbio_blurb_jnlpba | rouge1_fmeasure | 37.7 | 4.4 | 23.8 | 48.6 |

## J.2 Evaluation

All language models summary statistics are calculated using n=5 samples (1 score per prompt). Standard error is calculated using the sample standard deviation. Pearson's Correlation was calculated using SciPy v1.7.3. All other metrics are calculated calculated using Scikit-learn v1.0.2. All models less than 11B in size are evaluated using fp32 precision on a single 8x A40 compute node

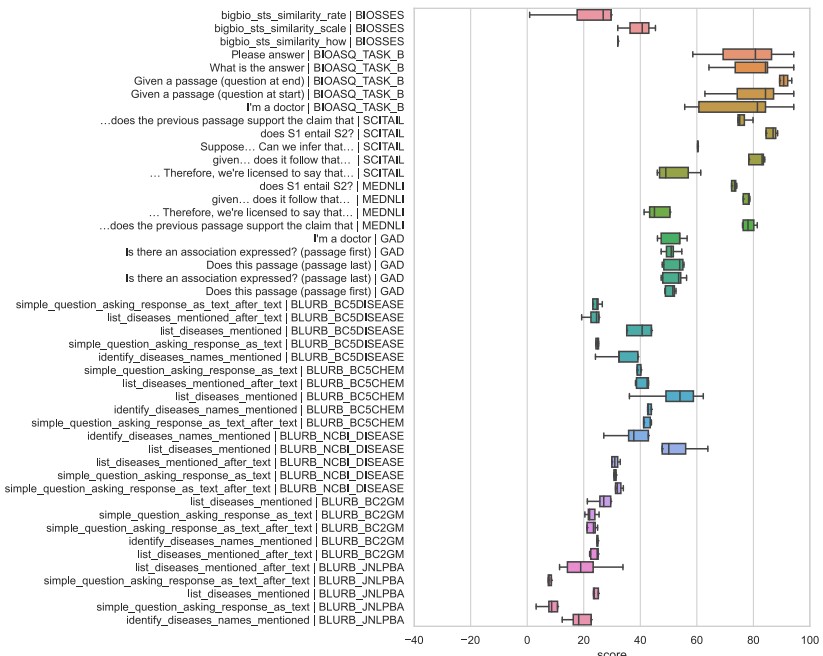

Figure 11: Per-prompt scores (x-axis) for all instruction-tuned language models (T0_3B, T0, T0+, T0++, GPT-3 (text-davinci-0002). Prompt template names are on the y-axis. Performance for non semantic similarity tasks is more varied and higher performing compared to current GPT-2 style pretrained models or T5 models with standard pretraining.

running CUDA 11.2. GPT-NeoX-20B, OPT-66B, and BLOOM were run on an 4xA100 80GB using LLM.int8() for inference. GPT-3 results were queried from OpenAI on 10/3/2022.

## J.3    Code

All experiment were run using the most up-to-date version of BIGBIO before paper submssion. https://github.com/bigscience-workshop/biomedical commit 0ff295b25bb1be813e64f13246090bff6168cb5a

Complete language model evaluation harness code and instructions for running BIGBIO experiments: https://github.com/bigscience-workshop/lm-evaluation-harness/tree/bigbio

For these experiments, we used a modified version of PromptSource: https://github.com/bigscience-workshop/promptsource/tree/eval-hackathon Prompt templates are available at https://github.com/OpenBioLink/promptsource and are outlined below.

All pretrained language models were downloaded from Hugging Face's datasets hub.

## J.4    Prompt Templates

The following prompt templates were developed using PromptSource. A prompt consists of a set of answer choices, an input template, and an output template.

### J.4.1    BIOSSES

**Prompt 1: "bigbio_sts_similarity_scale"**

**Answer Choices:**

```
0.0 ||| 0.1 ||| 0.2 ||| 0.3 ||| 0.4 ||| 0.5 ||| 0.6 ||| 0.7 ||| 0.8 ||| 0.9
||| 1.0 ||| 1.1 ||| 1.2 ||| 1.3 ||| 1.4 ||| 1.5 ||| 1.6 ||| 1.7 ||| 1.8
```

Table 12: BIOSSES example instance.

| Key | Value |
|-----|-------|
| id | 6 |
| document_id | 7 |
| text_1 | Recently, it was reported that expression of IDH1R132H su... |
| text_2 | the mechanism was clarified by yet another genomic survey... |
| label | 1.6 |

```
||| 1.9 ||| 2.0 ||| 2.1 ||| 2.2 ||| 2.3 ||| 2.4 ||| 2.5 ||| 2.6 ||| 2.7
||| 2.8 ||| 2.9 ||| 3.0 ||| 3.1 ||| 3.2 ||| 3.3 ||| 3.4 ||| 3.5 ||| 3.6
||| 3.7 ||| 3.8 ||| 3.9 ||| 4.0
```

**Input Template:**

```
from {{"0"}} to {{"4"}}, how similar are "{{text_1}}" and "{{text_2}}"?
```

**Output Template:**

```
{{label}}
```

**Prompt 2: "bigbio_sts_similarity_how"**

**Answer Choices:**

```
0.0 ||| 0.1 ||| 0.2 ||| 0.3 ||| 0.4 ||| 0.5 ||| 0.6 ||| 0.7 ||| 0.8 ||| 0.9
||| 1.0 ||| 1.1 ||| 1.2 ||| 1.3 ||| 1.4 ||| 1.5 ||| 1.6 ||| 1.7 ||| 1.8
||| 1.9 ||| 2.0 ||| 2.1 ||| 2.2 ||| 2.3 ||| 2.4 ||| 2.5 ||| 2.6 ||| 2.7
||| 2.8 ||| 2.9 ||| 3.0 ||| 3.1 ||| 3.2 ||| 3.3 ||| 3.4 ||| 3.5 ||| 3.6
||| 3.7 ||| 3.8 ||| 3.9 ||| 4.0
```

**Input Template:**

```
How similar are "{{text_1}}" and "{{text_2}}"? Give a score \
between {{"0"}} and {{"4"}}.
```

**Output Template:**

```
{{label}}
```

**Prompt 3: "bigbio_sts_similarity_rate"**

**Answer Choices:**

```
0.0 ||| 0.1 ||| 0.2 ||| 0.3 ||| 0.4 ||| 0.5 ||| 0.6 ||| 0.7 ||| 0.8 ||| 0.9
||| 1.0 ||| 1.1 ||| 1.2 ||| 1.3 ||| 1.4 ||| 1.5 ||| 1.6 ||| 1.7 ||| 1.8
||| 1.9 ||| 2.0 ||| 2.1 ||| 2.2 ||| 2.3 ||| 2.4 ||| 2.5 ||| 2.6 ||| 2.7
||| 2.8 ||| 2.9 ||| 3.0 ||| 3.1 ||| 3.2 ||| 3.3 ||| 3.4 ||| 3.5 ||| 3.6
||| 3.7 ||| 3.8 ||| 3.9 ||| 4.0
```

**Input Template:**

```
Rate the similarity of these two sentences ({{"0"}} being the lowest \
and {{"4"}} the highest):
"{{text_1}}" and "{{text_2}}"
```

**Output Template:**

```
{{label}}
```

---

**Prompt 4: "bigbio_sts_similarity_on_a_scale"**

**Answer Choices:**

```
0.0 ||| 0.1 ||| 0.2 ||| 0.3 ||| 0.4 ||| 0.5 ||| 0.6 ||| 0.7 ||| 0.8 ||| 0.9
||| 1.0 ||| 1.1 ||| 1.2 ||| 1.3 ||| 1.4 ||| 1.5 ||| 1.6 ||| 1.7 ||| 1.8
||| 1.9 ||| 2.0 ||| 2.1 ||| 2.2 ||| 2.3 ||| 2.4 ||| 2.5 ||| 2.6 ||| 2.7
||| 2.8 ||| 2.9 ||| 3.0 ||| 3.1 ||| 3.2 ||| 3.3 ||| 3.4 ||| 3.5 ||| 3.6
||| 3.7 ||| 3.8 ||| 3.9 ||| 4.0
```

**Input Template:**

```
On a scale of {{"0"}} (completely unrelated) to {{"4"}} (exactly same) \
score these sentences:
"{{text_1}}" and "{{text_2}}"
```

**Output Template:**

```
{{label}}
```

---

**Prompt 5: "bigbio_sts_similarity_what_is"**

**Answer Choices:**

```
0.0 ||| 0.1 ||| 0.2 ||| 0.3 ||| 0.4 ||| 0.5 ||| 0.6 ||| 0.7 ||| 0.8 ||| 0.9
||| 1.0 ||| 1.1 ||| 1.2 ||| 1.3 ||| 1.4 ||| 1.5 ||| 1.6 ||| 1.7 ||| 1.8
||| 1.9 ||| 2.0 ||| 2.1 ||| 2.2 ||| 2.3 ||| 2.4 ||| 2.5 ||| 2.6 ||| 2.7
||| 2.8 ||| 2.9 ||| 3.0 ||| 3.1 ||| 3.2 ||| 3.3 ||| 3.4 ||| 3.5 ||| 3.6
||| 3.7 ||| 3.8 ||| 3.9 ||| 4.0
```

**Input Template:**

```
What is the similarity of these two sentences on a scale of {{"0"}} (low) \
to {{"4"}} (high): "{{text_1}}" and "{{text_2}}"
```

**Output Template:**

```
{{label}}
```

Table 13: BioASQ example instance.

| Key | Value |
| --- | --- |
| id | 5c58a74e86df2b917400000d_0 |
| question_id | 5c58a74e86df2b917400000d |
| document_id | http://www.ncbi.nlm.nih.gov/pubmed/29623652 |
| question | Is Baloxavir effective for influenza? |
| type | yesno |
| choices | [] |
| context | Baloxavir marboxil (Xofluza™; baloxavir) is an oral cap-d... |
| answer | ['yes'] |

### J.4.2 BioASQ

**Prompt 1: "Given a passage (question at end)"**

**Answer Choices:**

```
no ||| yes
```

**Input Template:**

```
Given a passage: {{ context }}

Answer the question: "{{question}}"
```

**Output Template:**

```
{{answer[0]}}
```

---

**Prompt 2: "I'm a doctor"**

**Answer Choices:**

```
no ||| yes
```

**Input Template:**

```
I'm a doctor and I need to answer the question "{{ question }}" using \
the following passage:

{{ context }}
```

**Output Template:**

```
{{answer[0]}}
```

---

**Prompt 3: "What is the answer"**

**Answer Choices:**

```
no ||| yes
```

**Input Template:**

```
What is the answer to the question "{{ question }}" based on \
the following passage:

{{ context }}
```

**Output Template:**

```
{{answer[0]}}
```

---

**Prompt 4: "Please answer"**

**Answer Choices:**

```
no ||| yes
```

**Input Template:**

```
Please answer the question "{{ question }}" using \
the following passage:

{{ context }}
```

**Output Template:**

```
{{answer[0]}}
```

---

**Prompt 5: "Given a passage (question at start)"**

**Answer Choices:**

```
no ||| yes
```

**Input Template:**

```
Given the following passage, answer the question: "{{question}}"

Passage: {{ context }}
```

**Output Template:**

```
{{answer[0]}}
```

### J.4.3 SciTail

Table 14: SciTail example instance.

| Key | Value |
| --- | --- |
| id | 0 |
| premise | Based on the list provided of the uses of substances 1-7,... |
| hypothesis | If a substance has a ph value greater than 7,that indicat... |
| label | neutral |

**Prompt 1: "... Therefore, we're licensed to say that..."**

**Answer Choices:**

```
true ||| false
```

**Input Template:**

```
{{premise}} Therefore, we are licensed to say that {{hypothesis}}
{{ answer_choices | join(' or ') }}
```

**Output Template:**

```
{% if label == "entailment" %}
{{answer_choices[0]}}
{% else %}
{{answer_choices[1]}}
{% endif %}
```

---

**Prompt 2: "Suppose... Can we infer that..."**

**Answer Choices:**

```
neutral ||| entailment
```

**Input Template:**

```
Suppose {{premise}} Can we infer that {{hypothesis}}?
```

**Output Template:**

```
{{label}}
```

---

**Prompt 3: "...does the previous passage support the claim that"**

**Answer Choices:**

```
yes ||| no
```

**Input Template:**

```
{{premise}} Does the previous passage support the claim that {{hypothesis}}?
```

**Output Template:**

```
{% if label == "entailment" %}
{{answer_choices[0]}}
{% else %}
{{answer_choices[1]}}
{% endif %}
```

---

**Prompt 4: "given... does it follow that..."**

**Answer Choices:**

```
yes ||| no
```

**Input Template:**

```
Given that {{premise}} Does it follow that {{hypothesis}}
{{ answer_choices | join(' or ') }}
```

**Output Template:**

```
{% if label == "entailment" %}
{{answer_choices[0]}}
{% else %}
{{answer_choices[1]}}
{% endif %}
```

---

**Prompt 5: "does S1 entail S2?"**

**Answer Choices:**

```
yes ||| no
```

**Input Template:**

```
Sentence 1: {{premise}}

Sentence 2: {{hypothesis}}

Question: Does Sentence 1 entail Sentence 2? \
{{ answer_choices | join(' or ') }}
```

**Output Template:**

```
{% if label == "entailment" %}
{{answer_choices[0]}}
{% else %}
{{answer_choices[1]}}
{% endif %}
```

### J.4.4  MedNLI

Table 15: MedNLI example instance.

| Key | Value |
| --- | --- |
| id | 1f2a8146-66c7-11e7-b4f2-f45c89b91419 |
| premise | In the ED, initial VS revealed T 98.9, HR 73, BP 121/90, ... |
| hypothesis | The patient is hemodynamically stable |
| label | entailment |

**Prompt 1: "... Therefore, we're licensed to say that..."**

**Answer Choices:**

```
yes ||| no
```

**Input Template:**

```
{{premise}} Therefore, we are licensed to say that {{hypothesis}}
{{ answer_choices | join(' or ') }}
```

**Output Template:**

```
{% if label == "entailment" %}
{{answer_choices[0]}}
{% else %}
{{answer_choices[1]}}
{% endif %}
```

**Prompt 2: "Suppose... Can we infer that..."**

**Answer Choices:**

```
yes ||| no
```

**Input Template:**

```
Suppose {{premise}} Can we infer that {{hypothesis}}?
```

**Output Template:**

```
{% if label == "entailment" %}
{{answer_choices[0]}}
{% else %}
{{answer_choices[1]}}
{% endif %}
```

---

**Prompt 3: "...does the previous passage support the claim that"**

**Answer Choices:**

```
yes ||| no
```

**Input Template:**

```
{{premise}} Does the previous passage support the claim that {{hypothesis}}?
```

**Output Template:**

```
{% if label == "entailment" %}
{{answer_choices[0]}}
{% else %}
{{answer_choices[1]}}
{% endif %}
```

---

**Prompt 4: "given... does it follow that..."**

**Answer Choices:**

```
yes ||| no
```

**Input Template:**

```
Given that {{premise}} Does it follow that {{hypothesis}} \
{{ answer_choices | join(' or ') }}
```

**Output Template:**

```
{% if label == "entailment" %}
{{answer_choices[0]}}
{% else %}
{{answer_choices[1]}}
{% endif %}
```

---

**Prompt 5: "does S1 entail S2?"**

**Answer Choices:**

```
yes ||| no
```

**Input Template:**

```
Sentence 1: {{premise}}

Sentence 2: {{hypothesis}}

Question: Does Sentence 1 entail Sentence 2? \
{{ answer_choices | join(' or ') }}
```

**Output Template:**

```
{% if label == "entailment" %}
{{answer_choices[0]}}
{% else %}
{{answer_choices[1]}}
{% endif %}
```

### J.4.5  GAD

Table 16: GAD example instance.

| Key | Value |
|-----|-------|
| id | 0 |
| document_id | 0 |
| text | These results suggest that the C1772T polymorphism in @GE... |
| labels | ['1'] |

**Prompt 1: "Does this passage (passage last)"**

**Answer Choices:**

```
No ||| Yes
```

**Input Template:**

```
Does the following passage indicate that there is an association \
between the gene @GENE$ and the disease @DISEASE$ ?

{{ text }}
```

**Output Template:**

```
{{ answer_choices[labels[0] | int] }}
```

**Prompt 2: "Does this passage (passage first)"**

**Answer Choices:**

```
No ||| Yes
```

**Input Template:**

```
{{ text }}

Does this passage indicate that there is an association between the \
gene @GENE$ and the disease @DISEASE$ ?
```

**Output Template:**

```
{{ answer_choices[labels[0] | int] }}
```

---

**Prompt 3: "Is there an association expressed? (passage last)"**

**Answer Choices:**

```
No ||| Yes
```

**Input Template:**

```
Is there an association between the gene @GENE$ and the disease \
@DISEASE$ expressed in this passage?

{{ text }}
```

**Output Template:**

```
{{ answer_choices[labels[0] | int] }}
```

---

**Prompt 4: "I'm a doctor"**

**Answer Choices:**

```
No ||| Yes
```

**Input Template:**

```
I'm a doctor. Can you tell me, is there an association between the \
gene @GENE$ and the disease @DISEASE$ expressed in this passage?

{{ text }}
```

**Output Template:**

```
{{ answer_choices[labels[0] | int] }}
```

---

**Prompt 5: "Is there an association expressed? (passage first)"**

**Answer Choices:**

```
No ||| Yes
```

**Input Template:**

```
{{ text }}

Is there an association between the gene @GENE\$ and the disease \
@DISEASE$ expressed in this passage?
```

**Output Template:**

```
{{ answer_choices[labels[0] | int] }}
```

# K   Large-scale Multi-Task Learning

We make the MTL model available at `https://huggingface.co/bigscience-biomedical/bigbio-mtl`. Code and instructions to reproduce our results can be found in `https://github.com/leonweber/biomuppet`.

Table 17: MTL dataset statistics

| Task | Abbrev. | # Train Examples | # Valid Examples | # Datasets |
|---|---|---|---|---|
| Relation Extraction | RE | 656,171 | 106,519 | 14 |
| Coreference Resolution | COREF | 113,137 | 35,030 | 9 |
| Event Argument Extraction | EAE | 294,129 | 119,033 | 10 |
| Text Classification | CLASS | 30,743 | 3,416 | 2 |
| Semantic Textual Similarity | STS | 7,215 | 804 | 6 |
| Question Answering | QA | 6,490 | 561 | 2 |
| Named Entity Recognition | NER | 287,582 | 89,135 | 53 |
| Event Detection | ED | 28,388 | 9,883 | 10 |
| Total | | 1,423,855 | 364,381 | 106 |

## K.1   Conversion to MaChAmp

We generated training and evaluation data for 106 datasets that were available when we started to develop the MTL project source code (BIGBIO version found here). If a dataset within this collective set did not have a predefined validation split, we reserved 10% of its training data as the validation set. Each dataset also had one BIGBIO-to-MaChAmp transformation script per BIGBIO task. The purpose of this transformation script is to convert the data represented in the BIGBIO-schema in a MaChAmp-compatible input for simple extension to the ML library. For statistics of the resulting data set Table 17 and for examples of the transformed task data see Tables 18 and 19.

We model **Relation Extraction** (RE) as relation classification. Each sentence in an input passage is split; subsequently, we construct on example per entity-pair by introducing special marker tokens to mark the start and end of each head and tail entity. We consider each example as a text classification problem in MaChAmp, where the goal is to predict the type of relation between the marked head/ail entities, including a 'None' type relation. We follow the BLURB preprocessing strategy for RE and replace the strings of the marked head and tail entity with their respective entity type. For multi-label datasets where an entity or relation may possess multiple labels, we transform such cases to a multiclass dataset by concatenating all labels. We use this multilabel-to-multiclass transformation for all task types, if required.

We treat **Coreference Resolution** (COREF) in a similar fashion as RE, with the only difference that we have only two relation types: 'coref' denoting a coreference relation between two token spans and 'None'.

We transform the **Event Argument Extraction** (EAE) data in exactly the same way as RE, with the trigger span acting as the head entity and all possible event arguments (entities and triggers) acting as tail entities.

For **Text Classification** (CLASS), we adapt the BIGBIO version to the MaChAmp format without any further modification apart from the multilabel-to-multiclass transformation.

We transform the **Semantic Textual Similarity** (STS) task from a regression task to classification by replacing the STS score with the decantile into which it falls. We use the template 'Text1 [SEP] Text2' where 'Text1' and 'Text2' are either words, sentences or paragraphs depending on the dataset.

We model **Named Entity Recognition** (NER) and Event Detection (ED) as sequence labelling tasks using an IOB-tagging scheme after sentence splitting.

For **Question Answering** (QA), we experimented with two formulations. In the classification formulation, we construct one example per answer candidate by using the template 'Context [SEP] Question [SEP] AnswerCandidate' and the two labels 'True' (if 'AnswerCandidate' is the correct answer) and 'False' (if 'AnswerCandidate' is the wrong answer). In the sequence labelling setting, we

use the template 'Context. Question' and mark all tokens in occurrences of the answer in 'Context' with 'answer' and the rest with 'O'.

We use Flair's [59] 'SegtokSentenceSplitter' for sentence splitting and 'SpaceTokenizer' for tokenization.

Table 18: Examples for the classification task formulation

| Task Type | Input | Label |
|-----------|-------|-------|
| RE | Taken together, these results make it clear that @chemical$-bound forms of ORC and @protein$ are likely to be required for productive interactions and pre-RC formation. | bind |
| COREF | We investigated the potential of the @aryl hydrocarbon receptor$ (@AHR$) to suppress NF-kappaB regulated-gene expression, especially acute-phase genes, such as serum amyloid A (Saa). | coref |
| EAE | v-erbA @Gene_expression$ is required to @Negative_regulation$ c-erbA function in erythroid cell differentiation and regulation of the erbA target gene CAII. | cause |
| CLASS | These results are in contrast with the findings of Santos et al.(16), who reported a significant association between low sedentary time and healthy CVF among Portuguese | result&supportive |
| STS | Renal failure [SEP] Kidney failure | 8 |
| QA (class) | Cytokeratin 7/20 staining has been reported to be helpful [...] [SEP] Is cytokeratin immunoreactivity useful in the diagnosis of short-segment Barrett's oesophagus in Korea? | True |

## K.2 Hyperparameters

For hyperparameter choices, we use a mixture of the MaChAmp default hyperparameters and the suggestions from [1]. We use AdamW [119] with a polynomial decay learning rate schedule with 50,000 warmup steps with a maximum learning rate of 1e-4. We set weight decay to 0.01, dropout to 0.1 and the maximum length of the transformer to 512. We use an effective batch size of 32 tasks and 16 examples per task, train the model with Automated Mixed Precision set to fp16 using apex (https://github.com/NVIDIA/apex) and clip the gradient norm to 5. Finally, we downsample large datasets by using MaChAmp's multinomial sampling with alpha set to 0.5.

For model selection we evaluate the model after each epoch on all validation sets and select the model with the highest average accuracy.

## K.3 Results on Validation Sets

We evaluate our MTL model on all validation sets and deliberately refrain from evaluating on the test sets, because we did not rule out train/test overlap. The validation results can be found in Figure 12. Results vary strongly across task types, with the model performing well on COREF (mean 86.9% F1), CLASS (mean 85.4 acc), and NER (mean 72.2% F1). Performance on STS (mean 28.1 Pearson's r) and QA (mean 42.8 acc) is surprisingly low. We attribute the weak performance on both STS and QA

Table 19: Examples for the sequence labeling task formulation

| Task Type | Input | Label |
|---|---|---|
| NER | Tricuspid valve regurgitation and lithium carbonate toxicity in a newborn infant. | `B-Disease I-Disease I-Disease O B-Chemical I-Chemical B-Disease O O O O` |
| ED | Coexpression of NF-kappa B/Rel and Sp1 transcription factors in human immunodeficiency virus 1-induced, dendritic cell-T-cell syncytia. | `B-Gene_expression O O O O O O O O O O O O O O` |
| QA (seq) | a frameshift mutation is a deletion or insertion of one or more nucleotides [...] a frameshift mutation is a deletion or insertion of one or more of what that changes the reading frame of the base sequence ? | `O O O O O [...]  answer [...]  O` |

to the small amount of data per task (7,215 and 6,490 training examples respectively), which might prevent the model from allocating parameters for these tasks.

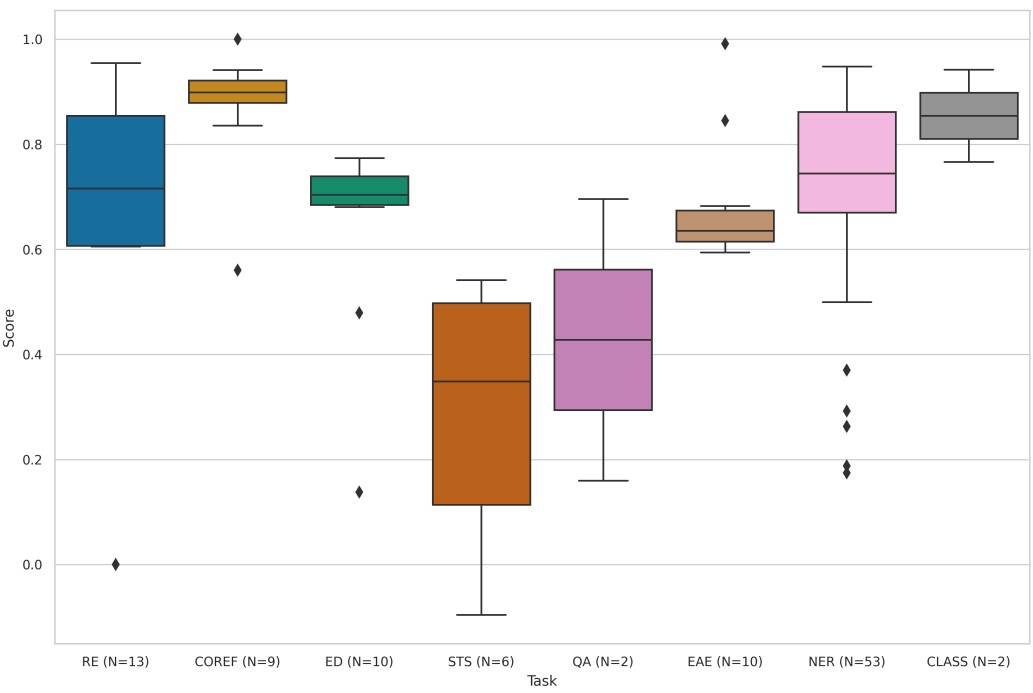

Figure 12: Validation set results of the MTL model by task. 'RE' denotes Relation Extraction, 'COREF' Coreference Resolution, 'ED' event detection, 'STS' Semantic Textual Similarity, 'QA' Question Answering, 'EAE' Event Argument Extraction, 'NER' Named Entity Recognition, and 'CLASS' Text Classification. Score is accuracy for QA and CLASS, Pearson's r for STS and F1 for the rest.

### K.4 Resources Used for Training

We trained the MTL model on a local machine on four RTX 3090 GPUs. Training for 50 epochs allowed the model to converge in all tested configurations and took roughly 33 hours.

## L  BIGBIO vs. Existing Benchmarks

**Biomedical Meta-dataset Benchmarks**    Table 20 compares BIGBIO against attributes of other popular English language biomedical meta-dataset benchmarks. To-date, our framework is the only one that supports API-based dataset access, providing access to 4x more datasets than the largest comparable meta-dataset. BLUE and BLURB do not provide dataset access via an API and require manual downloading and preprocessing. Depending on the dataset, these preprocessing choices may not be easily reproducible. For example, in 4/5 NER tasks BLURB uses the IOB transformed datasets generated by Crichton et al. [75]. These datasets rely on regular expression-based tokenization and sentence boundary detection methods developed by Crichton et al. and can vary by dataset, making it difficult to systemically the impact of different tokenization and sentence splitting choices.

End-to-end few and zero-shot evaluation of datasets, prompts, and pretrained language models is emerging as a standardized way to measure the performance of pretrained language models. BLUE and BLURB do not directly support prompt evaluation. BoX provides prompts for 32 biomedical datasets and Python tools for evaluating BART [116]-based language models, however BoX does not provide any access to the original datasets themselves. BIGBIO integrates with the prompt engineering framework PROMPTSOURCE to support users more easily designing prompts and running evaluations using the EleutherAI Language Model Evaluation Harness [17]. We currently support several seq2seq and causal language models (e.g., T5, T0, GPT families) available in Hugging Face's model hub. Currently BIGBIO implements 25 prompts (5 datasets, 5 prompts), with future work focusing on constructing a library of task and dataset-specific biomedical prompts.

Table 20: Attributes of existing English biomedical meta-dataset benchmarks

| Name | Datasets | Tasks | Langs | Data API | Reproducible Preprocessing | Prompts | Evaluation Harness |
|---|---|---|---|---|---|---|---|
| BIGBIO | 127 | 12 | 10 | ✓ | ✓ | *partial* | ✓ |
| BLUE [34] | 10 | 5 | 1 | | *partial* | | |
| BLURB [19] | 13 | 7 | 1 | | *partial* | | |
| BoX [32] | 32 | 9 | 1 | | | ✓ | ✓ |

**Dataset Coverage**    Table 21 enumerates the list of datasets currently used by BIGBIO , BLUE, BLURB, and BoX. Abbreviations are as follows: Named Entity Recognition (NER); Relation Extraction (RE); Question Answering (QA); Part-of-Speech Tagging (POS); Sentiment Analysis (SA) ; Natural Language Inference (NLI); and Systematic Review (SR). For the 32 public datasets BIGBIO provides data loaders for the majority (28/32), while the remaining 4 are still being implemented by volunteers as of 06/16/2022. Note that *private* indicates that datasets are not available publicly or via DUA and thus cannot currently be included in BIGBIO.

Table 21: BIGBIO support of datasets used in popular meta-dataset benchmarks.

| Task Type | Dataset | BIGBIO | BLUE | BLURB | BoX | DUA |
|---|---|---|---|---|---|---|
| NER | BC2GM | ✓ | | ✓ | ✓ | |
| NER | BC5-chem | ✓ | ✓ | ✓ | ✓ | |
| NER | BC5-disease | ✓ | ✓ | ✓ | ✓ | |
| NER | EBM PICO | ✓ | | ✓ | | |
| NER | JNLPBA | ✓ | | ✓ | ✓ | |
| NER | NCBI-disease | ✓ | | ✓ | ✓ | |
| RE | ChemProt | ✓ | ✓ | ✓ | ✓ | |
| RE | DDI | ✓ | ✓ | ✓ | ✓ | |
| RE | GAD | ✓ | | ✓ | | |
| QA | PubMedQA | ✓ | | ✓ | ✓ | |
| QA | BioASQ | ✓ | | ✓ | ✓ | ✓ |
| DC | HoC | ✓ | ✓ | ✓ | ✓ | |
| STS | BIOSSES | ✓ | ✓ | ✓ | | |
| STS | MedSTS | * | ✓ | | | ✓ |
| NER | n2c2 2010 | ✓ | ✓ | | ✓ | ✓ |
| NER | ShARe/CLEF 2013 | * | ✓ | | | ✓ |
| NLI | MedNLI | ✓ | ✓ | | | ✓ |
| NER | n2c2 deid 2006 | ✓ | | | ✓ | ✓ |
| DC | n2c2 RFHD 2014 | ✓ | | | ✓ | ✓ |
| NER | AnatEM | ✓ | | | ✓ | |
| NER | BC4CHEMD | ✓ | | | ✓ | |
| NER | BioNLP09 | ✓ | | | ✓ | |
| NER | BioNLP11EPI | ✓ | | | ✓ | |
| NER | BioNLP11ID | ✓ | | | ✓ | |
| NER | BioNLP13CG | ✓ | | | ✓ | |
| NER | BioNLP13GE | ✓ | | | ✓ | |
| NER | BioNLP13PC | ✓ | | | ✓ | |
| NER | CRAFT | * | | | ✓ | |
| NER | Ex-PTM | ✓ | | | ✓ | |
| NER | Linnaeus | ✓ | | | ✓ | |
| POS | GENIA | * | | | ✓ | |
| SA | Medical Drugs | ✓ | | | ✓ | |
| SR | COVID | | | | *private* | |
| SR | Cooking | | | | *private* | |
| SR | HRT | | | | *private* | |
| SR | Accelerometer | | | | *private* | |
| SR | Acromegaly | | | | *private* | |

* denotes dataset implementation in-progress

# M   Example Data Card

We generated data cards for all BIGBIO datasets. We include an example dataset from each schema type to illustrate data cards for different tasks. A PDF of all content is available on our project homepage.

## Cantemist Data Card

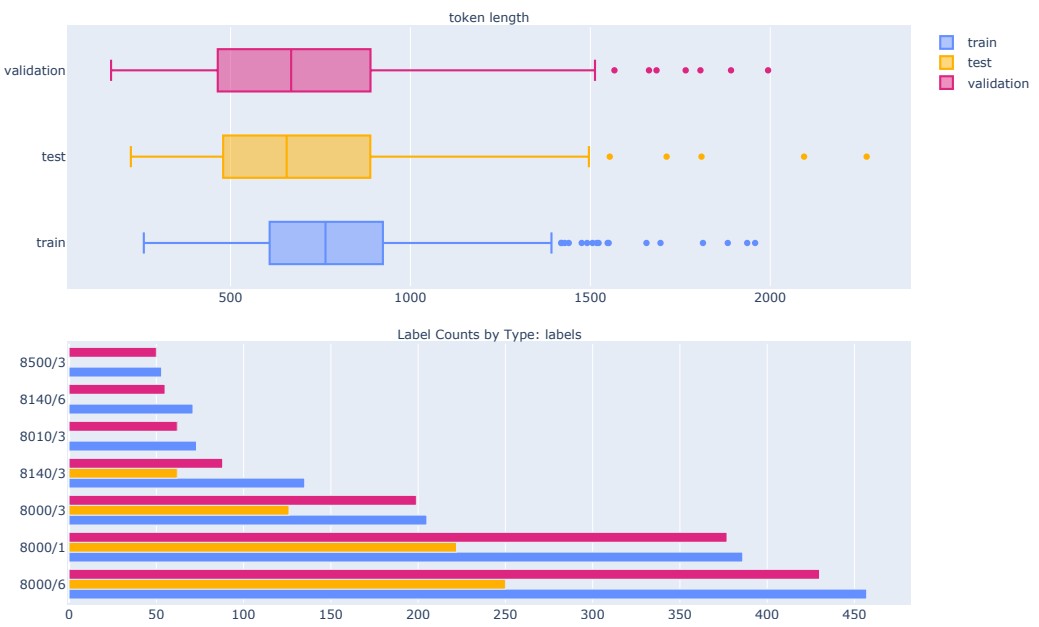

Figure 13: Token frequency distribution by split (top) and frequency of different kind of instances (bottom).

**Dataset Description:** Collection of 1301 oncological clinical case reports written in Spanish, with tumor morphology mentions manually annotated and mapped by clinical experts to a controlled terminology. Every tumor morphology mention is linked to an eCIE-O code (the Spanish equivalent of ICD-O). The original dataset is distributed in BRAT format, and was randomly sampled into 3 subsets. The training, development and test sets contain 501, 500 and 300 documents each, respectively. This dataset was designed for the CANcer TExt Mining Shared Task, sponsored by Plan-TL. The task is divided in 3 subtasks: CANTEMIST-NER, CANTEMIST-NORM and CANTEMIST-CODING.

CANTEMIST-NER track: requires finding automatically tumor morphology mentions. All tumor morphology mentions are defined by their corresponding character offsets in UTF-8 plain text medical documents.

CANTEMIST-NORM track: clinical concept normalization or named entity normalization task that requires to return all tumor morphology entity mentions together with their corresponding eCIE-O-3.1 codes i.e. finding and normalizing tumor morphology mentions.

CANTEMIST-CODING track: requires returning for each of document a ranked list of its corresponding ICD-O-3 codes. This it is essentially a sort of indexing or multi-label classification task or oncology clinical coding.

For further information, please visit `https://temu.bsc.es/cantemist` or send an email to encargo-pln-life@bsc.es

**Homepage:** `https://temu.bsc.es/cantemist/?p=4338`

**URL:** `https://zenodo.org/record/3978041/files/cantemist.zip?download=1`

**Licensing:** Creative Commons Attribution 4.0 International

**Languages:** Spanish

**Tasks:** NER, NED, Text Classification

**Schemas:** `TEXT`, `KB`, `source`

**Splits:** train, validation, test

## MEDIQA Data Card

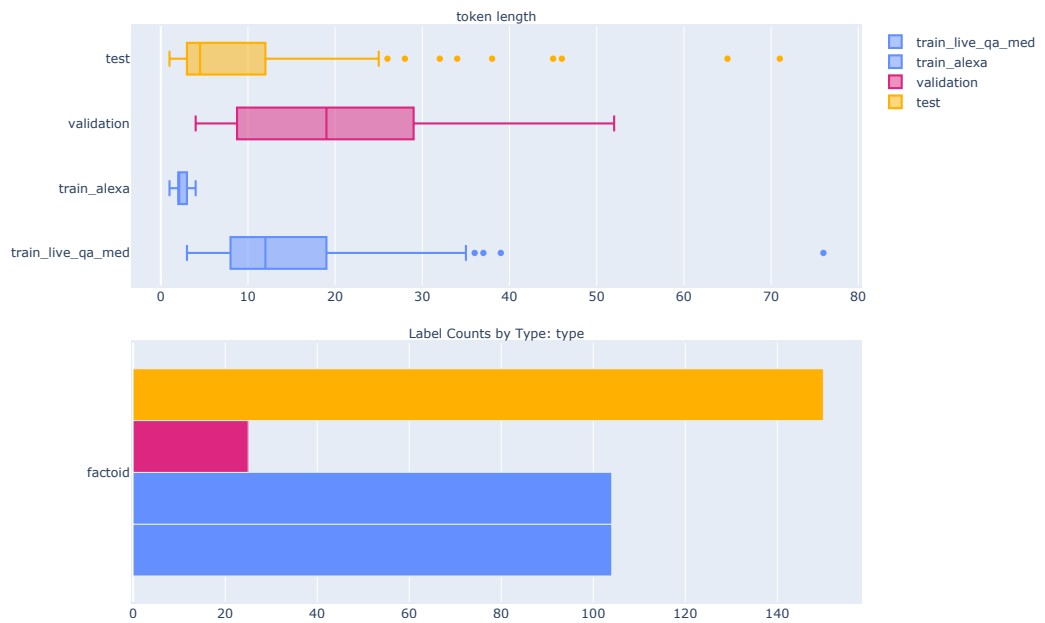

Figure 14: Token frequency distribution by split (top) and frequency of different kind of instances (bottom).

**Dataset Description:** The MEDIQA challenge is an ACL-BioNLP 2019 shared task aiming to attract further research efforts in Natural Language Inference (NLI), Recognizing Question Entailment (RQE), and their applications in medical Question Answering (QA). Mailing List: `https://groups.google.com/forum/#!forum/bionlp-mediqa`

In the QA task, participants are tasked to:- filter/classify the provided answers (1: correct, 0: incorrect).- re-rank the answers.

**Homepage:** `https://sites.google.com/view/mediqa2019`

**URL:** `https://github.com/abachaa/MEDIQA2019/archive/refs/heads/master.zip`

**Licensing:** License information unavailable

**Languages:** English

**Tasks:** Question Answering

**Schemas:** `QA`, `source`

**Splits:** train-1-liveQAMed, train-2-Alexa, validation, test

## AnEM Data Card

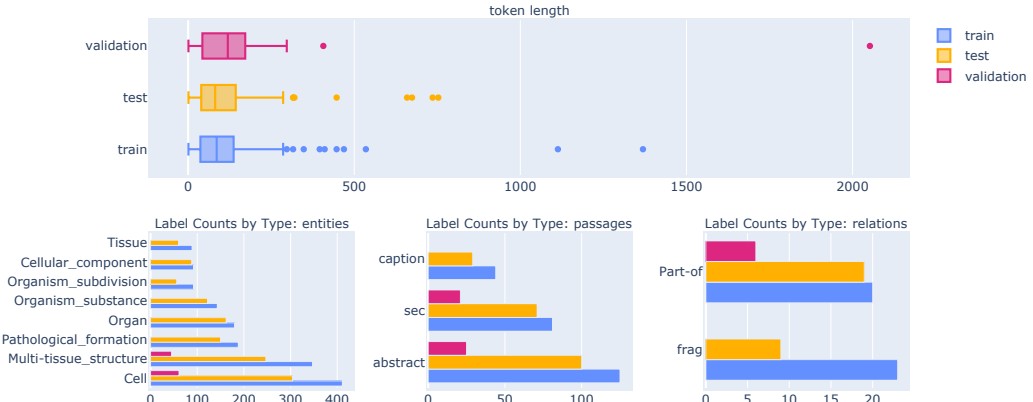

Figure 15: Token frequency distribution by split (top) and frequency of different kind of instances (bottom).

**Dataset Description** AnEM corpus is a domain- and species-independent resource manually annotated for anatomical entity mentions using a fine-grained classification system. The corpus consists of 500 documents (over 90,000 words) selected randomly from citation abstracts and full-text papers with the aim of making the corpus representative of the entire available biomedical scientific literature. The corpus annotation covers mentions of both healthy and pathological anatomical entities and contains over 3,000 annotated mentions.

**Homepage:** http://www.nactem.ac.uk/anatomy/

**URL:** http://www.nactem.ac.uk/anatomy/data/AnEM-1.0.4.tar.gz

**Licensing:** Creative Commons Attribution Share Alike 3.0 Unported

**Languages:** English

**Tasks:** NER, Coreference Resolution, Relation Extraction

**Schemas:** KB, source

**Splits:** train, validation, test

## ParaMed Data Card

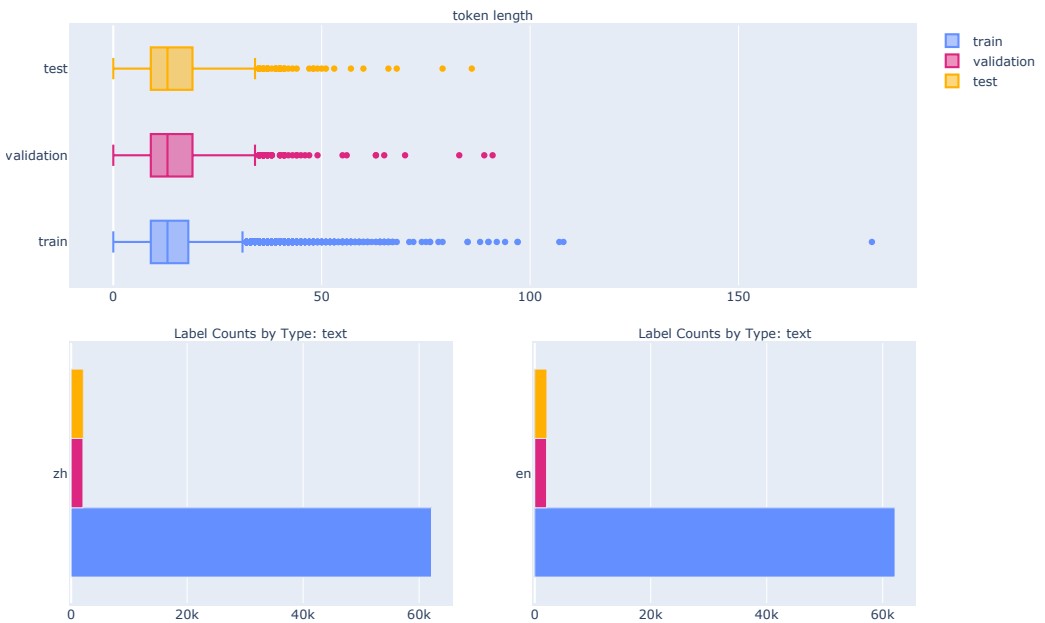

Figure 16: Token frequency distribution by split (top) and frequency of different kind of instances (bottom).

**Dataset Description:** NEJM is a Chinese-English parallel corpus crawled from the New England Journal of Medicine website. English articles are distributed through https://www.nejm.org/ and Chinese articles are distributed through http://nejmqianyan.cn/. The corpus contains all article pairs (around 2000 pairs) since 2011.

**Homepage:** `https://github.com/boxiangliu/ParaMed`

**URL:** `https://github.com/boxiangliu/ParaMed/blob/master/data/nejm-open-access.tar.gz?raw=true`

**Licensing:** Creative Commons Attribution 4.0 International

**Languages:** English, Chinese

**Tasks** Translation

**Schemas:** `t2t`, `source`

**Splits:** train, validation, test

## SciTail Data Card

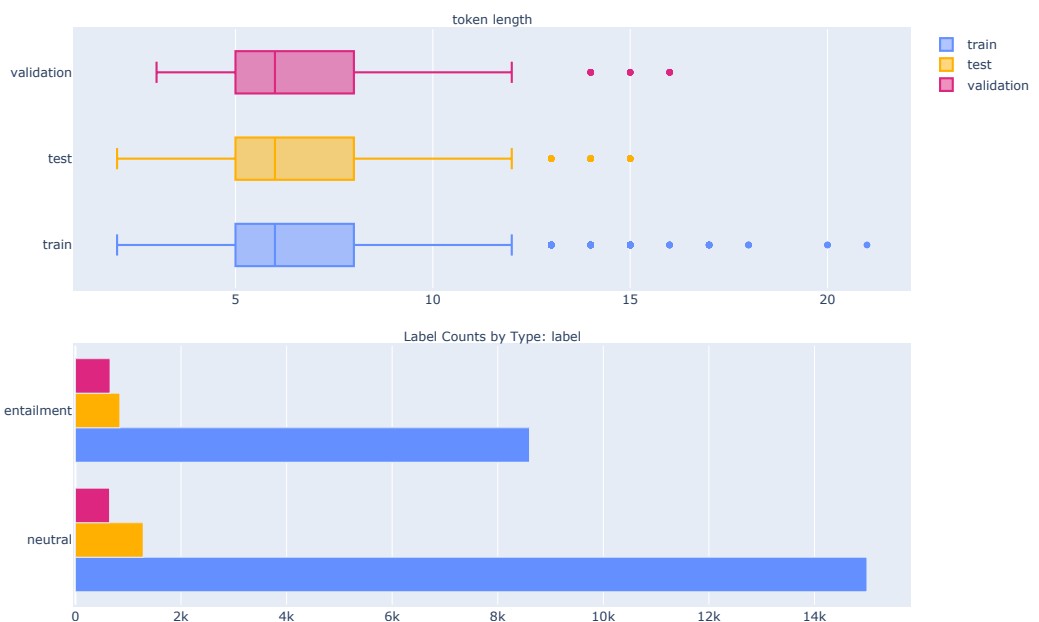

Figure 17: Token frequency distribution by split (top) and frequency of different kind of instances (bottom).

**Dataset Description** The SciTail dataset is an entailment dataset created from multiple-choice science exams and web sentences. Each question and the correct answer choice are converted into an assertive statement to form the hypothesis. We use information retrieval to obtain relevant text from a large text corpus of web sentences, and use these sentences as a premise P. We crowdsource the annotation of such premise-hypothesis pair as supports (entails) or not (neutral), in order to create the SciTail dataset. The dataset contains 27,026 examples with 10,101 examples with entails label and 16,925 examples with neutral label.

**Homepage:** `https://allenai.org/data/scitail`

**URL:** `https://ai2-public-datasets.s3.amazonaws.com/scitail/SciTailV1.1.zip`

**Licensing:** Apache License 2.0

**Languages:** English

**Tasks:** Textual Entailment

**Schemas:** `te`, `source`

**Splits:** train, validation, test

**MQP Data Card**

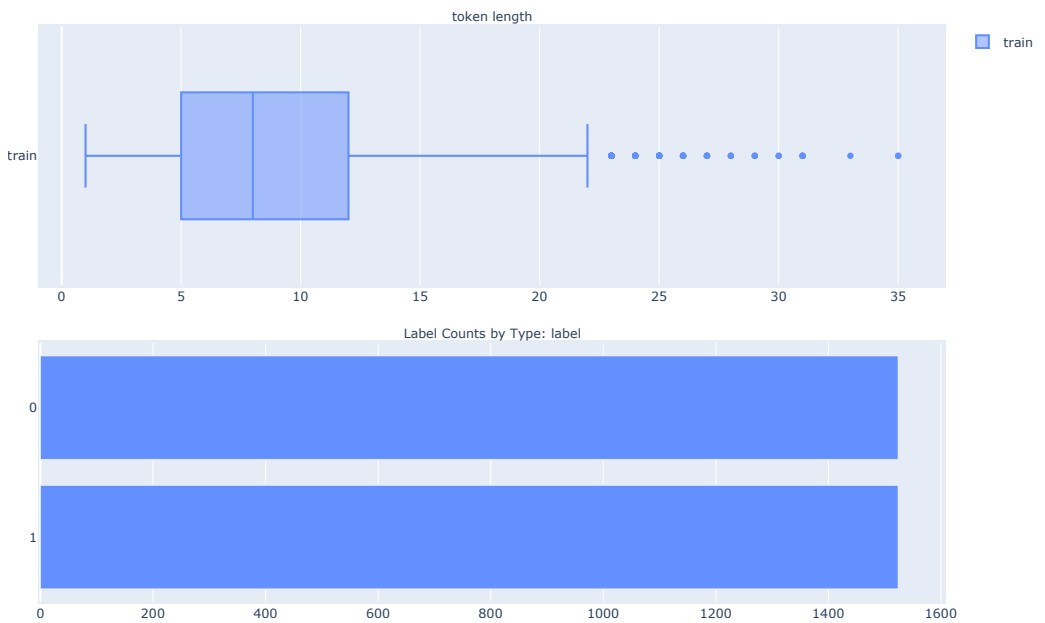

Figure 18: Token frequency distribution by split (top) and frequency of different kind of instances (bottom).

**Dataset Description:** Medical Question Pairs dataset by McCreery et al (2020) contains pairs of medical questions and paraphrased versions of the question prepared by medical professional. Paraphrased versions were labelled as similar (syntactically dissimilar but contextually similar ) or dissimilar (syntactically may look similar but contextually dissimilar). Labels 1: similar, 0: dissimilar

**Homepage:** https://github.com/curai/medical-question-pair-dataset

**URL:** https://raw.githubusercontent.com/curai/medical-question-pair-dataset/master/mqp.csv

**Licensing:** License information unavailable

**Languages:** English

**Tasks:** Semantic Similarity

**Schemas:** pairs, source

**Splits:** train

# N BIGBIO Data Card

**Dataset Description:** BIGBIO is a community project and meta-dataset consisting of 126+ dataset loader scripts providing programmatic access to expertly annotated biomedical natural language processing datasets. The constituent datasets support 12 tasks grouped into 6 schema types. 105 of these datasets are publicly available and can be automatically downloaded using the BIGBIO Python package. The remaining 21 require some level of manual action ranging from simple web forms to credentialed access and training on how to handle protected health information.

**Homepage:** `https://github.com/bigscience-workshop/biomedical`

**URL:** `https://github.com/bigscience-workshop/biomedical`

**Licensing:** `https://choosealicense.com/licenses/apache-2.0/`

**Languages:** English, Spanish, French, Chinese, German, Japanese, Dutch, Portuguese, Swedish, and Vietnamese

**Tasks:** named entity recognition (`NER`), named entity disambiguation/normalization (`NED`), event extraction (`EE`), relation extraction (`RE`), coreference resolution (`COREF`), question answering (`QA`), textual entailment (`TE`), text classification (`TXTCLASS`), semantic similarity (`STS`), paraphrasing (`PARA`), translation (`TRANSL`), summarization (`SUM`).

**Schemas:** Knowledge Base (`KB`), Question Answering (`QA`), Textual Entailment (`TE`), Text (`TEXT`), Text Pairs (`PAIRS`), Text to Text (`T2T`), source (`source`).

**Splits:** train, validation, test, sample

Table 22: Summary statistics for all datasets included in BIGBIO. Token counts (# Toks) assumes white space tokenziation and example instances (# N) correspond to the unit of text emitted by the dataloader iterable, usually a document, sentence, or text pair. Some datasets include k-folds or multiple training splits, which are noted by $k = *$. See each dataset's data card for more specific details, such as label counts by task.

| Dataset Name | BIGBIO Name | Split | # Chars | # Toks | # N | License | Tasks | Schema | Langs | Access |
|---|---|---|---|---|---|---|---|---|---|---|
| AnEM [134] | an_em | train | 300k | 44.3k | 250 | CC BY SA 3.0 | RE, NER, COREF | KB | EN | Public |
| | | valid | 85.6k | 11k | 50 | | | | | |
| | | test | 242k | 36.2k | 200 | | | | | |
| AnatEM [140] | anat_em | train | 840k | 122k | 606 | CC BY SA 3.0 | NER | KB | EN | Public |
| | | valid | 319k | 44.7k | 202 | | | | | |
| | | test | 547k | 79.2k | 404 | | | | | |
| AskAPatient [117] | ask_a_patient | train k=10 | 31.3k | 202k | 15665 | CC BY 4.0 | NER, NED | KB | EN | Public |
| | | validation k=10 | 1.74k | 10.6k | 792 | | | | | |
| | | test k=10 | 1.92k | 11.7k | 866 | | | | | |
| BC5CDR [112] | bc5cdr | train | 653k | 93k | 500 | Public Domain Mark 1.0 | RE, NER, NED | KB | EN | Public |
| | | valid | 647k | 92.3k | 500 | | | | | |
| | | test | 677k | 96.5k | 500 | | | | | |
| BC7-LitCovid [70] | bc7_litcovid | train | 34.4M | 4.97M | 24960 | *Unknown* | TXTCLASS | TEXT | EN | Public |
| | | valid | 3.69M | 532k | 2489 | | | | | |
| | | test | 8.68M | 1.26M | 6239 | | | | | |
| Bio-SimVerb [72] | bio_sim_verb | train | 14.9k | 2k | 1000 | *Unknown* | STS | PAIRS | EN | Public |
| Bio-SimLex [72] | bio_simlex | train | 16.1k | 1.98k | 988 | *Unknown* | STS | PAIRS | EN | Public |
| MESINESP 2021 [86] | bioasq_2021_mesinesp | valid | 256k | 38.6k | 109 | CC BY 4.0 | TXTCLASS | TEXT | ES | Public |
| | | test | 59.4k | 9.06k | 119 | | | | | |
| BioASQ Task B [161] | bioasq_task_b | train | 5.21M | 2.26M | 9955 | NLM | QA | QA | EN | DUA |
| | | valid | 573k | 249k | 1029 | | | | | |
| | | test | 581k | 253k | 1041 | | | | | |
| BioASQ Task C 2017 [127] | bioasq_task_c_2017 | train | 2.59B | 346M | 62952 | NLM | TXTCLASS | TEXT | EN | DUA |
| | | test | 895M | 120M | 22610 | | | | | |
| BioInfer [141] | bioinfer | train | 164k | 23.7k | 894 | CC BY 2.0 | RE, NER | KB | EN | Public |
| | | test | 40.8k | 5.93k | 206 | | | | | |
| BiologyHowWhy-Corpus [95] | biology_how_why_corpus | train | 2.33M | 985k | 1269 | *Unknown* | QA | QA | EN | Public |
| BIOMRC [137] | biomrc | train | 114k | 48.6k | 30 | *Unknown* | QA | QA | EN | Public |
| BioNLP 2009 [104] | bionlp_shared_task_2009 | train | 1.21M | 176k | 800 | GENIA Project | EE, NER, COREF | KB | EN | Public |
| | | valid | 234k | 33.8k | 150 | | | | | |
| | | test | 397k | 57.3k | 260 | | | | | |
| BioNLP 2011 EPI [133] | bionlp_st_2011_epi | train | 901k | 127k | 600 | GENIA Project | EE, NER, COREF | KB | EN | Public |
| | | valid | 310k | 43.5k | 200 | | | | | |
| | | test | 653k | 91.9k | 440 | | | | | |
| BioNLP 2011 GE [105] | bionlp_st_2011_ge | train | 1.41M | 206k | 908 | CC BY 3.0 | EE, NER, COREF | KB | EN | Public |
| | | valid | 435k | 64.1k | 259 | | | | | |
| | | test | 541k | 79k | 347 | | | | | |
| BioNLP 2011 ID [145] | bionlp_st_2011_id | train | 438k | 64.7k | 152 | GENIA Project | EE, NER, COREF | KB | EN | Public |
| | | valid | 119k | 18.4k | 46 | | | | | |
| | | test | 335k | 50k | 118 | | | | | |
| BioNLP 2011 REL [146] | bionlp_st_2011_rel | train | 1.21M | 176k | 800 | GENIA Project | RE, NER, COREF | KB | EN | Public |
| | | valid | 234k | 33.8k | 150 | | | | | |
| | | test | 397k | 57.3k | 260 | | | | | |
| BioNLP 2013 CG [142] | bionlp_st_2013_cg | train | 467k | 66.1k | 300 | GENIA Project | EE, NER, COREF | KB | EN | Public |
| | | valid | 153k | 21.7k | 100 | | | | | |
| | | test | 297k | 42.1k | 200 | | | | | |
| BioNLP 2013 GE [106] | bionlp_st_2013_ge | train | 371k | 54.9k | 222 | GENIA Project | RE, EE, NER, COREF | KB | EN | Public |
| | | valid | 391k | 57.9k | 249 | | | | | |
| | | test | 506k | 75.1k | 305 | | | | | |
| BioNLP 2013 GRO [107] | bionlp_st_2013_gro | train | 200k | 29.4k | 150 | GENIA Project | RE, EE, NER | KB | EN | Public |
| | | valid | 59.8k | 8.7k | 50 | | | | | |
| | | test | 132k | 19k | 100 | | | | | |
| BioNLP 2013 PC [132] | bionlp_st_2013_pc | train | 378k | 53.8k | 260 | GENIA Project | EE, NER, COREF | KB | EN | Public |

| Name | Config | Split | | | | License | Tasks | Schema | Lang | Access |
|---|---|---|---|---|---|---|---|---|---|---|
| | | valid | 131k | 18.6k | 90 | | | | | |
| | | test | 253k | 36k | 175 | | | | | |
| BioNLP 2019 BB [65] | bionlp_st_2019_bb | train | 129k | 19k | 133 | Unknown | RE, NER, NED | KB | EN | Public |
| | | valid | 66.5k | 9.71k | 66 | | | | | |
| | | test | 110k | 16.2k | 96 | | | | | |
| BioRED [122] | biored | train | 660k | 94.2k | 400 | Unknown | RE, NER | KB | EN | Public |
| | | valid | 173k | 24.9k | 100 | | | | | |
| | | test | 168k | 24.1k | 100 | | | | | |
| BioRelEx [100] | biorelex | train | 237k | 37.8k | 1405 | Unknown | RE, NER, NED, COREF | KB | EN | Public |
| | | valid | 33.1k | 5.29k | 201 | | | | | |
| BioScope [170] | bioscope | train | 171k | 42k | 6383 | CC BY 2.0 | NER | KB | EN | Public |
| BIOSSES [154] | biosses | train | 20.1k | 2.94k | 64 | GPL 3.0 | STS | PAIRS | EN | Public |
| | | valid | 5.09k | 733 | 16 | | | | | |
| | | test | 6.44k | 925 | 20 | | | | | |
| CADEC [99] | cadec | train | 575k | 104k | 1250 | Custom | NER, NED | KB | EN | Public |
| CANTEMIST [124] | cantemist | train | 2.6M | 382k | 501 | CC BY 4.0 | NER, NED, TXTCLASS | KB, TEXT | ES | Public |
| | | valid | 2.33M | 341k | 500 | | | | | |
| | | test | 1.41M | 206k | 300 | | | | | |
| CAS [89] | cas | train | 972k | 175k | 7580 | DUA | TXTCLASS | TEXT, KB | FR | DUA |
| CellFinder [129] | cellfinder | train | 171k | 25.2k | 26 | CC BY SA 3.0 | NER | KB | EN | Public |
| | | test | 205k | 30.3k | 27 | | | | | |
| CHEBI Corpus [149] | chebi_nactem | train | 1.95M | 306k | 100 | CC BY 4.0 | RE, NER | KB | EN | Public |
| CHEMDNER [111] | chemdner | train | 4.88M | 687k | 3500 | Unknown | NER, TXTCLASS | KB, TEXT | EN | Public |
| | | valid | 4.86M | 683k | 3500 | | | | | |
| | | test | 4.19M | 591k | 3000 | | | | | |
| ChemProt [112] | chemprot | train | 1.64M | 230k | 1020 | Public Domain Mark 1.0 | RE, NER | KB | EN | Public |
| | | valid | 990k | 139k | 612 | | | | | |
| | | test | 1.3M | 182k | 800 | | | | | |
| CHIA [114] | chia | train | 1.04M | 151k | 2000 | CC BY 4.0 | RE, NER | KB | EN | Public |
| Citation GIA Test Collection [177] | citation_gia_ test_collection | test | 230k | 33.4k | 151 | Unknown | NER, NED | KB | EN | Public |
| CodiEsp [125] | codiesp | train | 193M | 29.1M | 176294 | CC BY 4.0 | TXTCLASS | TEXT | ES | Public |
| | | train | 0 | 0 | 500 | | | | | |
| | | valid | 0 | 0 | 250 | | | | | |
| | | test | 0 | 0 | 250 | | | | | |
| CORD-NER [172] | cord_ner | train | 407M | 62.5M | 29500 | Custom | NER | KB | EN | Public |
| CT-EBM-SP [68] | ctebmsp | train | 625k | 90.3k | 420 | CC BY NC 4.0 | NER | KB | ES | Public |
| | | valid | 212k | 30.7k | 140 | | | | | |
| | | test | 206k | 29.9k | 140 | | | | | |
| DDI Corpus [93] | ddi_corpus | train | 928k | 128k | 714 | CC BY NC 4.0 | RE, NER | KB | EN | Public |
| | | test | 281k | 38.7k | 303 | | | | | |
| DIANN [147] | diann_iber_eval | train | 548k | 81.8k | 400 | Unknown | NER | KB | EN, ES | Public |
| | | test | 144k | 21.5k | 100 | | | | | |
| | | train | 1.06M | 156k | 400 | | | | | |
| | | test | 275k | 40.9k | 100 | | | | | |
| DisTEMIST [85] | distemist | train | 1.76M | 264k | 750 | CC BY 4.0 | NER | KB | EN | Public |
| EBM NLP [130] | ebm_pico | train | 7.68M | 1.29M | 4746 | Unknown | NER | KB | EN | Public |
| | | test | 306k | 50.9k | 187 | | | | | |
| EHR-Rel [148] | ehr_rel | train | 174k | 23.4k | 3741 | Apache 2.0 | STS | PAIRS | EN | Public |
| ESSAI [76] | essai | train | 1.83M | 314k | 13848 | DUA | TXTCLASS | TEXT, KB | FR | DUA |
| EU-ADR [168] | euadr | train | 452k | 64.3k | 300 | Unknown | RE, NER | KB | EN | Public |
| Evidence Inference 2.0 [78] | evidence_inference | train | 2.91M | 446k | 10150 | MIT | TE | TE | EN | Public |
| | | valid | 352k | 53.6k | 1238 | | | | | |
| | | test | 358k | 54.9k | 1228 | | | | | |
| GAD [66] | gad | train | 740k | 113k | 4261 | CC BY 4.0 | TXTCLASS | TEXT | EN | Public |
| | | valid | 91k | 13.9k | 535 | | | | | |
| | | test | 98.6k | 14.9k | 534 | | | | | |
| GENETAG [157] | genetag | train | 1.16M | 197k | 7500 | NCBI | NER | KB | EN | Public |
| | | valid | 783k | 133k | 5000 | | | | | |
| | | test | 387k | 65.5k | 2500 | | | | | |

| Name | Code | Split | Col1 | Col2 | Col3 | License | Tasks | Type | Lang | Access |
|------|------|-------|------|------|------|---------|-------|------|------|--------|
| PTM Events [131] | `genia_ptm_event_corpus` | train | 145k | 20.8k | 112 | GENIA Project | EE, NER, COREF | KB | EN | Public |
| GENIA Relation Corpus [143] | `genia_relation_corpus` | train | 1.21M | 176k | 800 | GENIA Project | RE | KB | EN | Public |
| | | valid | 234k | 33.8k | 150 | | | | | |
| | | test | 397k | 57.3k | 260 | | | | | |
| GENIA Term Corpus [135] | `genia_term_corpus` | train | 2.99M | 435k | 2000 | GENIA Project | NER | KB | EN | Public |
| GEOkhoj v1 [81] | `geokhoj_v1` | train | 4.25M | 554k | 25000 | CC BY NC 4.0 | TXTCLASS | TEXT | EN | Public |
| | | test | 848k | 111k | 5000 | | | | | |
| GNormPlus [177] | `gnormplus` | train | 379k | 55.7k | 281 | Unknown | NER, NED | KB | EN | Public |
| | | test | 359k | 52.5k | 262 | | | | | |
| Hallmarks of Cancer [62] | `hallmarks_of_cancer` | train | 1.96M | 312k | 12119 | GPL 3.0 | TXTCLASS | TEXT | EN | Public |
| | | valid | 296k | 47.1k | 1798 | | | | | |
| | | test | 573k | 91.8k | 3547 | | | | | |
| HPRD50 [83] | `hprd50` | train | 18.1k | 2.67k | 34 | Unknown | RE, NER | KB | EN | Public |
| | | test | 4.94k | 710 | 9 | | | | | |
| IEPA [79] | `iepa` | train | 75.1k | 10.9k | 160 | Unknown | RE | KB | EN | Public |
| | | test | 18.6k | 2.68k | 40 | | | | | |
| JNLPBA [74] | `jnlpba` | train | 0 | 0 | 37094 | CC BY 3.0 | NER | KB | EN | Public |
| | | valid | 0 | 0 | 7714 | | | | | |
| LINNAEUS [87] | `linnaeus` | train | 2.46M | 373k | 84 | CC BY 4.0 | NER, NED | KB | EN | Public |
| LLL05 [72] | `lll` | train | 13.2k | 1.99k | 77 | Unknown | RE | KB | EN | Public |
| | | test | 13.1k | 2.07k | 87 | | | | | |
| Mantra GSC [109] | `mantra_gsc` | train | 16k | 2.12k | 50 | CC BY 4.0 | NER, NED | KB | EN, FR, DE, NL, ES | Public |
| MayoSRS [139] | `mayosrs` | train | 2.69k | 314 | 101 | CC0 1.0 | STS | PAIRS | EN | Public |
| MedQA [97] | `med_qa` | train | 1.84M | 890k | 11298 | Unknown | QA | QA | EN | Public |
| | | valid | 229k | 111k | 1412 | | | | | |
| | | test | 234k | 114k | 1413 | | | | | |
| MedDialog [71] | `meddialog` | train | 290k | 51k | 981 | Unknown | TXTCLASS | TEXT | EN, ZH | Public |
| | | valid | 41.8k | 7.35k | 126 | | | | | |
| | | test | 35.5k | 6.31k | 122 | | | | | |
| MEDDOCAN [123] | `meddocan` | train | 1.42M | 208k | 500 | CC BY 4.0 | NER | KB | ES | Public |
| | | valid | 755k | 111k | 250 | | | | | |
| | | test | 711k | 105k | 250 | | | | | |
| MedHop [180] | `medhop` | train | 187M | 78.7M | 1620 | CC BY SA 3.0 | QA | QA | EN | Public |
| | | valid | 32.8M | 13.8M | 342 | | | | | |
| Medical Data [101] | `medical_data` | train | 11M | 1.81M | 5279 | Unknown | TE | TE | EN | DUA |
| | | test | 7.12M | 1.16M | 2924 | | | | | |
| MEDIQA NLI [151] | `mediqa_nli` | test | 49.6k | 8.37k | 405 | PhysioNet 1.5 | TE | TE | EN | DUA |
| MEDIQA QA [64] | `mediqa_qa` | train k=2 | 1.91M | 4.56M | 104 | Unknown | QA | QA | EN | Public |
| | | valid | 1.24M | 519k | 25 | | | | | |
| | | test | 5.78M | 2.42M | 150 | | | | | |
| MEDIQA RQE [64] | `mediqa_rqe` | train | 1.69M | 262k | 8588 | Unknown | TE | TE | EN | Public |
| | | valid | 86.5k | 15.6k | 302 | | | | | |
| | | test | 68.1k | 12.1k | 230 | | | | | |
| MedMentions [126] | `medmentions` | train | 4.16M | 606k | 2635 | CC0 1.0 | NER, NED | KB | EN | Public |
| | | valid | 1.4M | 204k | 878 | | | | | |
| | | test | 1.39M | 203k | 879 | | | | | |
| MedNLI [150] | `mednli` | train | 1.51M | 240k | 11232 | PhysioNet 1.5 | TE | TE | EN | DUA |
| | | valid | 196k | 31.1k | 1395 | | | | | |
| | | test | 187k | 29.6k | 1422 | | | | | |
| MeQSum [63] | `meqsum` | train | 405k | 70.8k | 1000 | Unknown | SUM | T2T | EN | Public |
| MiniMayoSRS [139] | `minimayosrs` | train | 803 | 92 | 29 | CC0 1.0 | STS | PAIRS | EN | Public |
| miRNA [61] | `mirna` | train | 272k | 38.2k | 201 | CC BY NC 3.0 | NER, NED | KB | EN | Public |
| | | test | 115k | 16k | 100 | | | | | |
| MLEE [144] | `mlee` | train | 199k | 27.9k | 131 | CC BY NC SA 3.0 | RE, EE, NER, COREF | KB | EN | Public |
| | | valid | 68.1k | 9.61k | 44 | | | | | |

| Dataset | config | split | | | | License | Task | Schema | Lang | Availability |
|---|---|---|---|---|---|---|---|---|---|---|
| | | test | 135k | 19.1k | 87 | | | | | |
| MQP [112] | mqp | train | 644k | 120k | 3048 | *Unknown* | STS | PAIRS | EN | Public |
| MSH WSD [96] | msh_wsd | train | 52.8M | 7.59M | 37888 | UMLS | NED | KB | EN | DUA |
| MuchMore [67] | muchmore | train
train | 8.43M
12.7M | 1.11M
1.69M | 7808
6374 | *Unknown* | NER | KB | EN,
DE | Public |
| Multi-XScience [120] | multi_xscience | train
valid
test | 143M
23.9M
23.6M | 21.3M
3.54M
3.51M | 30369
5066
5093 | MIT | SUM, PARA | T2T | EN | Public |
| MutationFinder [69] | mutation_finder | valid
test | 416k
726k | 61.4k
107k | 305
508 | Custom | NER | KB | EN | Public |
| n2c2 2006 De-identification [165] | n2c2_2006_deid | train
test | 2.25M
952k | 340k
146k | 669
220 | DUA | NER | KB | EN | DUA |
| n2c2 2006 Smoking Status [164] | n2c2_2006_smokers | train
test | 1.72M
479k | 304k
85.1k | 398
104 | DUA | TXTCLASS | TEXT | EN | DUA |
| n2c2 2008 Obesity [162] | n2c2_2008 | train
test | 5M
3.5M | 852k
595k | 730
507 | DUA | TXTCLASS | TEXT | EN | DUA |
| n2c2 2009 Medication [166] | n2c2_2009 | train
test | 4.86M
3.75M | 824k
637k | 696
553 | DUA | NER | KB | EN | DUA |
| n2c2 2010 Relations [167] | n2c2_2010 | train
test | 827k
1.48M | 150k
267k | 170
256 | DUA | RE, NER | KB | EN | DUA |
| n2c2 2011 Coreference [163] | n2c2_2011 | train
test | 1.37M
916k | 247k
167k | 251
173 | DUA | COREF | KB | EN | DUA |
| n2c2 2014 De-identification [156] | n2c2_2014_deid | train
test | 3.4M
2.19M | 489k
316k | 790
514 | DUA | NER | KB | EN | DUA |
| n2c2 2014 Cardiac Risk Factors [113] | n2c2_2014_risk_factors | train
test | 3.4M
2.19M | 489k
316k | 790
514 | DUA | TXTCLASS | TEXT | EN | DUA |
| n2c2 2018 Selection Criteria [155] | n2c2_2018_track1 | train
test | 3.91M
1.64M | 550k
231k | 202
86 | DUA | TXTCLASS | TEXT | EN | DUA |
| n2c2 2018 ADE [92] | n2c2_2018_track2 | train
test | 3.84M
2.54M | 574k
377k | 303
202 | DUA | RE, NER | KB | EN | DUA |
| NCBI Disease [80] | ncbi_disease | train
valid
test | 747k
133k
135k | 113k
20.1k
20.4k | 592
100
100 | CC0 1.0 | NER, NED | KB | EN | Public |
| NLM-Gene [94] | nlm_gene | train
test | 812k
180k | 114k
25.2k | 450
100 | CC0 1.0 | NER, NED | KB | EN | Public |
| NLM WSD [173] | nlm_wsd | train | 8.37M | 1.22M | 5000 | UMLS | NED | KB | EN | DUA |
| NLM-Chem [94] | nlmchem | train
valid
test | 2.69M
663k
1.52M | 408k
100k
229k | 80
20
50 | CC0 1.0 | NER, NED,
TXTCLASS | KB,
TEXT | EN | Public |
| NTCIR-13 MedWeb [152] | ntcir_13_medweb | train
test
train
test | 79.4M
8.38M
163k
50.7k | 3.71M
412k
27.2k
8.47k | 1920
640
1920
640 | CC BY 4.0 | TXTCLASS | TEXT | EN,
ZH,
JA | DUA |
| OSIRIS [84] | osiris | train | 172k | 25.7k | 105 | CC BY 3.0 | NER, NED | KB | EN | Public |
| ParaMed [118] | paramed | train
valid
test | 16.2M
552k
564k | 3.74M
128k
130k | 62127
2036
2102 | CC BY 4.0 | TRANSL | T2T | EN,
ZH | Public |
| PDR [103] | pdr | train | 274k | 40.5k | 179 | *Unknown* | EE, NER,
COREF | KB | EN | Public |
| PharmaCoNER [88] | pharmaconer | train
valid
test | 1.18M
567k
587k | 177k
85.1k
88.2k | 500
250
250 | CC BY 4.0 | NER,
TXTCLASS | KB,
TEXT | ES | Public |
| PhoNER_COVID19 [160] | pho_ner | train
valid
test | 671k
286k
433k | 168k
71.3k
108k | 5027
2000
3000 | Custom | NER | KB | VI | Public |
| PICO Annotation [182] | pico_extraction | train | 60.4k | 10.2k | 421 | *Unknown* | NER | KB | EN | Public |
| PMC-Patients [181] | pmc_patients | train
valid
test | 1.22B
6.72M
7.67M | 184M
1.02M
1.17M | 257366
2144
2366 | CC BY
NC SA 4.0 | STS | PAIRS | EN | Public |
| ProGene [82] | progene | split
k=10 | 821k | 4.76M | 30926 | CC BY 4.0 | NER | KB | EN | Public |

| | | | | | | | | | | |
|---|---|---|---|---|---|---|---|---|---|---|
| | | split k=10 | 43.3k | 251k | 1676 | | | | | |
| | | split k=10 | 96.1k | 557k | 3623 | | | | | |
| PsyTAR [183] | psytar | train | 319k | 56.4k | 3398 | CC BY 4.0 | NER | KB | EN | DUA |
| | | train | 57k | 7.56k | 6003 | | | | | |
| PUBHEALTH [110] | pubhealth | train | 5.61M | 899k | 9804 | MIT | TXTCLASS | PAIRS | EN | Public |
| | | valid | 683k | 110k | 1223 | | | | | |
| | | test | 692k | 111k | 1231 | | | | | |
| PubMedQA [98] | pubmed_qa | train | 1.28M | 549k | 450 | MIT | QA | QA | EN | Public |
| | | valid | 141k | 60.3k | 50 | | | | | |
| | | test | 1.45M | 618k | 500 | | | | | |
| PubTator Central [174] | pubtator_central | train | 19.5k | 2.91k | 4 | NCBI | NER, NED | KB | EN | Public |
| QUAERO [128] | quaero | train | 67.7k | 10.6k | 833 | GFDL 1.3 | NER | KB | FR | Public |
| | | valid | 68.2k | 10.5k | 832 | | | | | |
| | | test | 70k | 10.9k | 832 | | | | | |
| SCAI Chemical [108] | scai_chemical | train | 155k | 20.9k | 100 | Unknown | NER | KB | EN | Public |
| SCAI Disease [91] | scai_disease | train | 630k | 90.4k | 400 | Unknown | NER | KB | EN | Public |
| SciCite [73] | scicite | train | 1.82M | 280k | 8243 | Unknown | TXTCLASS | TEXT | EN | Public |
| | | valid | 203k | 31.3k | 916 | | | | | |
| | | test | 413k | 63.4k | 1861 | | | | | |
| SciELO [153] | scielo | train | 995M | 153M | 2828917 | CC BY 4.0 | TRANSL | T2T | EN, ES, PT | Public |
| SciFact [171] | scifact | train | 787k | 112k | 919 | CC BY NC 2.0 | TE | TE | EN | Public |
| | | valid | 280k | 39.6k | 339 | | | | | |
| | | test | 26.4k | 3.62k | 300 | | | | | |
| SciQ [179] | sciq | train | 11.8M | 4.96M | 11679 | CC BY NC 3.0 | QA | QA | EN | Public |
| | | valid | 993k | 418k | 1000 | | | | | |
| | | test | 1.02M | 428k | 1000 | | | | | |
| SciTail [102] | scitail | train | 4.19M | 681k | 23596 | Apache 2.0 | TE | TE | EN | Public |
| | | valid | 237k | 38.8k | 1304 | | | | | |
| | | test | 372k | 62.3k | 2126 | | | | | |
| SETH Corpus [159] | seth_corpus | train | 760k | 111k | 630 | Apache 2.0 | RE, NER | KB | EN | Public |
| SPL ADR [77] | spl_adr_200db | train | 29M | 3.46M | 2208 | CC0 1.0 | RE, NER, NED | KB | EN | Public |
| Swedish Medical NER [60] | swedish_medical_ner | train | 85k | 14.1k | 926 | CC BY SA 4.0 | NER | KB | SV | Public |
| SNP Corpus [158] | thomas2011 | test | 0 | 0 | 296 | Custom | NER, NED | KB | EN | Public |
| tmVar v1 [176] | tmvar_v1 | train | 547k | 80.2k | 334 | Unknown | NER | KB | EN | Public |
| | | test | 265k | 38.8k | 166 | | | | | |
| tmVar v2 [178] | tmvar_v2 | train | 259k | 38k | 158 | Unknown | NER, NED | KB | EN | Public |
| tmVar v3 [175] | tmvar_v3 | test | 812k | 119k | 500 | Unknown | NER, NED | KB | EN | Public |
| TwADR-L [117] | twadrl | train k=10 | 10k | 76.1k | 4805 | CC BY 4.0 | NER, NED | KB | EN | Public |
| | | validation k=10 | 327 | 1.84k | 125 | | | | | |
| | | test k=10 | 361 | 2.03k | 142 | | | | | |
| UMNSRS [136] | umnsrs | train | 11.3k | 1.2k | 587 | CC0 1.0 | STS | PAIRS | EN | Public |
| Verspoor 2013 [169] | verspoor_2013 | train | 279k | 42.9k | 120 | Unknown | RE, NER | KB | EN | Public |

## Supplementary References

[59] Alan Akbik, Tanja Bergmann, Duncan Blythe, Kashif Rasul, Stefan Schweter, and Roland Vollgraf. FLAIR: An easy-to-use framework for state-of-the-art NLP. In *Proceedings of the 2019 Conference of the North American Chapter of the Association for Computational Linguistics (Demonstrations)*, pages 54–59, Minneapolis, Minnesota, June 2019. Association for Computational Linguistics.

[60] Simon Almgren, Sean Pavlov, and Olof Mogren. Named entity recognition in swedish medical journals with deep bidirectional character-based lstms. In *Proceedings of the Fifth Workshop on Building and Evaluating Resources for Biomedical Text Mining (BioTxtM 2016)*, pages 30–39. The COLING 2016 Organizing Committee, 12 2016.

[61] Shweta Bagewadi, Tamara Bobi'c, Martin Hofmann-Apitius, Juliane Fluck, and Roman Klinger. Detecting mirna mentions and relations in biomedical literature. *F1000Research*, 3:205–205, Aug 2014. 26535109[pmid].

[62] Simon Baker, Ilona Silins, Yufan Guo, Imran Ali, Johan H"ogberg, Ulla Stenius, and Anna Korhonen. Automatic semantic classification of scientific literature according to the hallmarks of cancer. *Bioinform.*, 32(3):432–440, 2016.

[63] Asma Ben Abacha and Dina Demner-Fushman. On the summarization of consumer health questions. In *Proceedings of the 57th Annual Meeting of the Association for Computational Linguistics*, pages 2228–2234, Florence, Italy, July 2019. Association for Computational Linguistics.

[64] Asma Ben Abacha, Chaitanya Shivade, and Dina Demner-Fushman. Overview of the mediqa 2019 shared task on textual inference, question entailment and question answering. In *ACL-BioNLP 2019*, 2019.

[65] Robert Bossy, Louise Del'eger, Estelle Chaix, Mouhamadou Ba, and Claire N'edellec. Bacteria biotope at BioNLP open shared tasks 2019. In *Proceedings of The 5th Workshop on BioNLP Open Shared Tasks*, pages 121–131, Hong Kong, China, November 2019. Association for Computational Linguistics.

[66] Àlex Bravo, Janet Piñero, N'uria Queralt-Rosinach, Michael Rautschka, and Laura I Furlong. Extraction of relations between genes and diseases from text and large-scale data analysis: implications for translational research. *BMC Bioinformatics*, 16(1), February 2015.

[67] Paul Buitelaar, Thierry Declerck, Bogdan Sacaleanu, Špela Vintar, Diana Raileanu, and Claudia Crispi. A multi-layered, xml-based approach to the integration of linguistic and semantic annotations. In *Proceedings of EACL 2003 Workshop on Language Technology and the Semantic Web (NLPXML'03), Budapest, Hungary*, 2003.

[68] Leonardo Campillos-Llanos, Ana Valverde-Mateos, Adri'an Capllonch-Carri'on, and Antonio Moreno-Sandoval. A clinical trials corpus annotated with UMLS entities to enhance the access to evidence-based medicine. *BMC Medical Informatics and Decision Making*, 21, 2021.

[69] J. Gregory Caporaso, William A Baumgartner, David A Randolph, K. Bretonnel Cohen, and Lawrence Hunter. Mutationfinder: a high-performance system for extracting point mutation mentions from text. *Bioinformatics*, 23(14):1862–1865, Jul 2007.

[70] Qingyu Chen, Alexis Allot, Robert Leaman, Rezarta Islamaj Doğan, and Zhiyong Lu. Overview of the biocreative vii litcovid track: multi-label topic classification for covid-19 literature annotation. In *Proceedings of the seventh BioCreative challenge evaluation workshop*, 2021.

[71] Shu Chen, Zeqian Ju, Xiangyu Dong, Hongchao Fang, Sicheng Wang, Yue Yang, Jiaqi Zeng, Ruisi Zhang, Ruoyu Zhang, Meng Zhou, Penghui Zhu, and Pengtao Xie. Meddialog: A large-scale medical dialogue dataset. *CoRR*, abs/2004.03329, 2020.

[72] Billy Chiu, Sampo Pyysalo, Ivan Vulić, and Anna Korhonen. Bio-simverb and bio-simlex: Wide-coverage evaluation sets of word similarity in biomedicine. *BMC Bioinformatics*, 19, 02 2018.

[73] Arman Cohan, Waleed Ammar, Madeleine van Zuylen, and Field Cady. Structural scaffolds for citation intent classification in scientific publications. In *Conference of the North American Chapter of the Association for Computational Linguistics*, 2019.

[74] Nigel Collier and Jin-Dong Kim. Introduction to the bio-entity recognition task at JNLPBA. In *Proceedings of the International Joint Workshop on Natural Language Processing in Biomedicine and its Applications (NLPBA/BioNLP)*, pages 73–78, Geneva, Switzerland, August 28th and 29th 2004. COLING.

[75] Gamal Crichton, Sampo Pyysalo, Billy Chiu, and Anna Korhonen. A neural network multi-task learning approach to biomedical named entity recognition. *BMC bioinformatics*, 18(1):1–14, 2017.

[76] Clément Dalloux. Datasets – clément dalloux, 2020.

[77] Dina Demner-Fushman, Sonya Shooshan, Laritza Rodriguez, Alan Aronson, Francois Lang, Willie Rogers, Kirk Roberts, and Joseph Tonning. A dataset of 200 structured product labels annotated for adverse drug reactions. *Scientific Data*, 5:180001, 01 2018.

[78] Jay DeYoung, Eric Lehman, Benjamin Nye, Iain Marshall, and Byron C. Wallace. Evidence inference 2.0: More data, better models. In *Proceedings of the 19th SIGBioMed Workshop on Biomedical Language Processing*, pages 123–132, Online, July 2020. Association for Computational Linguistics.

[79] J Ding, D Berleant, D Nettleton, and E Wurtele. Mining MEDLINE: abstracts, sentences, or phrases? *Pac Symp Biocomput*, pages 326–337, 2002.

[80] Rezarta Islamaj Dogan, Robert Leaman, and Zhiyong Lu. Ncbi disease corpus: A resource for disease name recognition and concept normalization. *Journal of biomedical informatics*, 47:1–10, 2014.

[81] Inc. Elucidata. Geokhoj v1. https://github.com/ElucidataInc/GEOKhoj-datasets/tree/main/geokhoj_v1, 2020.

[82] Erik Faessler, Luise Modersohn, Christina Lohr, and Udo Hahn. ProGene - a large-scale, high-quality protein-gene annotated benchmark corpus. In *Proceedings of the 12th Language Resources and Evaluation Conference*, pages 4585–4596, Marseille, France, May 2020. European Language Resources Association.

[83] Katrin Fundel, Robert Küffner, and Ralf Zimmer. Relex–relation extraction using dependency parse trees. *Bioinformatics*, 23(3):365–371, 2007.

[84] Laura I Furlong, Holger Dach, Martin Hofmann-Apitius, and Ferran Sanz. Osirisv1.2: a named entity recognition system for sequence variants of genes in biomedical literature. *BMC Bioinformatics*, 9:84, 2008.

[85] Luis Gasco, Eulàlia Farré, Antonio Miranda-Escalada, Salvador Lima, and Martin Krallinger. DisTEMIST corpus: detection and normalization of disease mentions in spanish clinical cases, April 2022. Funded by the Plan de Impulso de las Tecnologías del Lenguaje (Plan TL).

[86] Luis Gasco, Anastasios Nentidis, Anastasia Krithara, Darryl Estrada-Zavala, Renato Toshiyuki Murasaki, Elena Primo-Peña, Cristina Bojo Canales, Georgios Paliouras, Martin Krallinger, et al. Overview of bioasq 2021-mesinesp track. evaluation of advance hierarchical classification techniques for scientific literature, patents and clinical trials. CEUR Workshop Proceedings, 2021.

[87] Martin Gerner, Goran Nenadic, and Casey M Bergman. Linnaeus: a species name identification system for biomedical literature. *BMC bioinformatics*, 11(1):1–17, 2010.

[88] Aitor Gonzalez-Agirre, Montserrat Marimon, Ander Intxaurrondo, Obdulia Rabal, Marta Villegas, and Martin Krallinger. Pharmaconer: Pharmacological substances, compounds and proteins named entity recognition track. In *Proceedings of The 5th Workshop on BioNLP Open Shared Tasks*, pages 1–10, Hong Kong, China, November 2019. Association for Computational Linguistics.

[89] Natalia Grabar, Vincent Claveau, and Cl'ement Dalloux. CAS: French corpus with clinical cases. In *Proceedings of the Ninth International Workshop on Health Text Mining and Information Analysis*, pages 122–128, Brussels, Belgium, October 2018. Association for Computational Linguistics.

[90] Yu Gu, Robert Tinn, Hao Cheng, Michael Lucas, Naoto Usuyama, Xiaodong Liu, Tristan Naumann, Jianfeng Gao, and Hoifung Poon. Domain-specific language model pretraining for biomedical natural language processing. *ACM Trans. Comput. Heal.*, 3(1):2:1–2:23, 2022.

[91] Harsha Gurulingappa, Roman Klinger, Martin Hofmann-Apitius, and Juliane Fluck. An empirical evaluation of resources for the identification of diseases and adverse effects in biomedical literature. In *LREC Workshop on Building and Evaluating Resources for Biomedical Text Mining*, 2010.

[92] Sam Henry, Kevin Buchan, Michele Filannino, Amber Stubbs, and Ozlem Uzuner. 2018 n2c2 shared task on adverse drug events and medication extraction in electronic health records. *J. Am. Medical Informatics Assoc.*, 27(1):3–12, 2020.

[93] María Herrero-Zazo, Isabel Segura-Bedmar, Paloma Martínez, and Thierry Declerck. The ddi corpus: An annotated corpus with pharmacological substances and drug–drug interactions. *Journal of Biomedical Informatics*, 46(5):914–920, 2013.

[94] Rezarta Islamaj, Robert Leaman, Sun Kim, Dongseop Kwon, Chih-Hsuan Wei, Donald C Comeau, Yifan Peng, David Cissel, Cathleen Coss, Carol Fisher, et al. Nlm-chem, a new resource for chemical entity recognition in pubmed full text literature. *Scientific Data*, 8(1):1–12, 2021.

[95] Peter Jansen, Mihai Surdeanu, and Peter Clark. Discourse complements lexical semantics for non-factoid answer reranking. In *Proceedings of the 52nd Annual Meeting of the Association for Computational Linguistics (Volume 1: Long Papers)*, pages 977–986, Baltimore, Maryland, June 2014. Association for Computational Linguistics.

[96] Antonio J Jimeno-Yepes, Bridget T McInnes, and Alan R Aronson. Exploiting mesh indexing in medline to generate a data set for word sense disambiguation. *BMC bioinformatics*, 12(1):1–14, 2011.

[97] Di Jin, Eileen Pan, Nassim Oufattole, Wei-Hung Weng, Hanyi Fang, and Peter Szolovits. What disease does this patient have? a large-scale open domain question answering dataset from medical exams. *Applied Sciences*, 11(14):6421, 2021.

[98] Qiao Jin, Bhuwan Dhingra, Zhengping Liu, William Cohen, and Xinghua Lu. Pubmedqa: A dataset for biomedical research question answering. In *Proceedings of the 2019 Conference on Empirical Methods in Natural Language Processing and the 9th International Joint Conference on Natural Language Processing (EMNLP-IJCNLP)*, pages 2567–2577, 2019.

[99] Sarvnaz Karimi, Alejandro Metke-Jimenez, Madonna Kemp, and Chen Wang. Cadec: A corpus of adverse drug event annotations. *Journal of biomedical informatics*, 55:73–81, 2015.

[100] Hrant Khachatrian, Lilit Nersisyan, Karen Hambardzumyan, Tigran Galstyan, Anna Hakobyan, Arsen Arakelyan, Andrey Rzhetsky, and Aram Galstyan. BioRelEx 1.0: Biological relation extraction benchmark. In *Proceedings of the 18th BioNLP Workshop and Shared Task*, pages 176–190, Florence, Italy, August 2019. Association for Computational Linguistics.

[101] Arbaaz Khan. Sentiment analysis for medical drugs, 2019.

[102] Tushar Khot, Ashish Sabharwal, and Peter Clark. Scitail: A textual entailment dataset from science question answering. In *AAAI*, 2018.

[103] Baeksoo Kim, Wonjun Choi, and Hyunju Lee. A corpus of plant–disease relations in the biomedical domain. *PLoS One*, 14(8):e0221582, 2019.

[104] Jin-Dong Kim, Tomoko Ohta, Sampo Pyysalo, Yoshinobu Kano, and Jun'ichi Tsujii. Overview of BioNLP'09 shared task on event extraction. In *Proceedings of the BioNLP 2009 Workshop Companion Volume for Shared Task*, pages 1–9, Boulder, Colorado, June 2009. Association for Computational Linguistics.

[105] Jin-Dong Kim, Yue Wang, Toshihisa Takagi, and Akinori Yonezawa. Overview of genia event task in bionlp shared task 2011. In *Proceedings of the BioNLP Shared Task 2011 Workshop*, BioNLP Shared Task '11, page 7–15, USA, 2011. Association for Computational Linguistics.

[106] Jin-Dong Kim, Yue Wang, and Yamamoto Yasunori. The Genia event extraction shared task, 2013 edition - overview. In *Proceedings of the BioNLP Shared Task 2013 Workshop*, pages 8–15, Sofia, Bulgaria, August 2013. Association for Computational Linguistics.

[107] Jung-jae Kim, Xu Han, Vivian Lee, and Dietrich Rebholz-Schuhmann. GRO task: Populating the gene regulation ontology with events and relations. In *Proceedings of the BioNLP Shared Task 2013 Workshop*, pages 50–57, Sofia, Bulgaria, August 2013. Association for Computational Linguistics.

[108] Corinna Kol'arik, Roman Klinger, Christoph M Friedrich, Martin Hofmann-Apitius, and Juliane Fluck. Chemical names: Terminological resources and corpora annotation. In *LREC Workshop on Building and Evaluating Resources for Biomedical Text Mining*, 2008.

[109] Jan A Kors, Simon Clematide, Saber A Akhondi, Erik M van Mulligen, and Dietrich Rebholz-Schuhmann. A multilingual gold-standard corpus for biomedical concept recognition: the Mantra GSC. *Journal of the American Medical Informatics Association*, 22(5):948–956, 05 2015.

[110] Neema Kotonya and Francesca Toni. Explainable automated fact-checking for public health claims. *arXiv preprint arXiv:2010.09926*, 2020.

[111] Martin Krallinger, Obdulia Rabal, Florian Leitner, Miguel Vazquez, David Salgado, Zhiyong Lu, Robert Leaman, Yanan Lu, Donghong Ji, Daniel M. Lowe, Roger A. Sayle, Riza Theresa Batista-Navarro, Rafal Rak, Torsten Huber, Tim Rockt"aschel, S'ergio Matos, David Campos, Buzhou Tang, Hua Xu, Tsendsuren Munkhdalai, Keun Ho Ryu, S. V. Ramanan, Senthil Nathan, Slavko Zitnik, Marko Bajec, Lutz Weber, Matthias Irmer, Saber A. Akhondi, Jan A. Kors, Shuo Xu, Xin An, Utpal Kumar Sikdar, Asif Ekbal, Masaharu Yoshioka, Thaer M. Dieb, Miji Choi, Karin Verspoor, Madian Khabsa, C. Lee Giles, Hongfang Liu, Komandur Elayavilli Ravikumar, Andre Lamurias, Francisco M. Couto, Hong-Jie

Dai, Richard Tzong-Han Tsai, Caglar Ata, Tolga Can, Anabel Usi'e, Rui Alves, Isabel Segura-Bedmar, Paloma Mart'inez, Julen Oyarzabal, and Alfonso Valencia. The chemdner corpus of chemicals and drugs and its annotation principles. *Journal of Cheminformatics*, 7(1):S2, Jan 2015.

[112] Rabal-O. Lourenço A. Krallinger, M. Effective transfer learning for identifying similar questions: Matching user questions to covid-19 faqs. *KDD '20: Proceedings of the 26th ACM SIGKDD International Conference on Knowledge Discovery & Data Mining*, 3458–3465, 2020.

[113] Vishesh Kumar, Amber Stubbs, Stanley Shaw, and Özlem Uzuner. Creation of a new longitudinal corpus of clinical narratives. *Journal of Biomedical Informatics*, 58:S6–S10, 2015. Supplement: Proceedings of the 2014 i2b2/UTHealth Shared-Tasks and Workshop on Challenges in Natural Language Processing for Clinical Data.

[114] Fabr'ıcio Kury, Alex Butler, Chi Yuan, Li-heng Fu, Yingcheng Sun, Hao Liu, Ida Sim, Simona Carini, and Chunhua Weng. Chia, a large annotated corpus of clinical trial eligibility criteria. *Scientific data*, 7(1):1–11, 2020.

[115] Katherine Lee, Daphne Ippolito, Andrew Nystrom, Chiyuan Zhang, Douglas Eck, Chris Callison-Burch, and Nicholas Carlini. Deduplicating training data makes language models better. *arXiv preprint arXiv:2107.06499*, 2021.

[116] Mike Lewis, Yinhan Liu, Naman Goyal, Marjan Ghazvininejad, Abdelrahman Mohamed, Omer Levy, Ves Stoyanov, and Luke Zettlemoyer. Bart: Denoising sequence-to-sequence pre-training for natural language generation, translation, and comprehension. *arXiv preprint arXiv:1910.13461*, 2019.

[117] Nut Limsopatham and Nigel Collier. Normalising medical concepts in social media texts by learning semantic representation. In *Proceedings of the 54th Annual Meeting of the Association for Computational Linguistics (Volume 1: Long Papers)*, pages 1014–1023, Berlin, Germany, August 2016. Association for Computational Linguistics.

[118] Boxiang Liu and Liang Huang. Paramed: a parallel corpus for english–chinese translation in the biomedical domain. *BMC Medical Informatics and Decision Making*, 21, 2021.

[119] Ilya Loshchilov and Frank Hutter. Decoupled weight decay regularization. In *International Conference on Learning Representations*, 2018.

[120] Yao Lu, Yue Dong, and Laurent Charlin. Multi-xscience: A large-scale dataset for extreme multi-document summarization of scientific articles, 2020.

[121] Ling Luo, Po-Ting Lai, Chih-Hsuan Wei, Cecilia N Arighi, and Zhiyong Lu. Biored: A comprehensive biomedical relation extraction dataset. *arXiv preprint arXiv:2204.04263*, 2022.

[122] Ling Luo, Po-Ting Lai, Chih-Hsuan Wei, Cecilia N. Arighi, and Zhiyong Lu. Biored: A comprehensive biomedical relation extraction dataset. *CoRR*, abs/2204.04263, 2022.

[123] Montserrat Marimon, Aitor Gonzalez-Agirre, Ander Intxaurrondo, Heidy Rodriguez, Jose Lopez Martin, Marta Villegas, and Martin Krallinger. Automatic de-identification of medical texts in spanish: the meddocan track, corpus, guidelines, methods and evaluation of results. In *IberLEF SEPLN*, pages 618–638, 2019.

[124] Antonio Miranda-Escalada, Eulàlia Farré, and Martin Krallinger. Named entity recognition, concept normalization and clinical coding: Overview of the cantemist track for cancer text mining in spanish, corpus, guidelines, methods and results. *IberLEF SEPLN*, pages 303–323, 2020.

[125] Antonio Miranda-Escalada, Aitor Gonzalez-Agirre, Jordi Armengol-Estapé, and Martin Krallinger. Overview of automatic clinical coding: Annotations, guidelines, and solutions for non-english clinical cases at codiesp track of clef ehealth 2020. *CLEF (Working Notes)*, 2020, 2020.

[126] Sunil Mohan and Donghui Li. Medmentions: A large biomedical corpus annotated with umls concepts, 2019.

[127] Anastasios Nentidis, Konstantinos Bougiatiotis, Anastasia Krithara, Georgios Paliouras, and Ioannis Kakadiaris. Results of the fifth edition of the BioASQ challenge. BioNLP 2017, 2007.

[128] Aurélie Névéol, Cyril Grouin, Jeremy Leixa, Sophie Rosset, and Pierre Zweigenbaum. The QUAERO French medical corpus: A ressource for medical entity recognition and normalization. In *Proc of BioTextMining Work*, pages 24–30, 2014.

[129] Mariana Neves, Alexander Damaschun, Andreas Kurtz, and Ulf Leser. Annotating and evaluating text for stem cell research. In *Proceedings of the Third Workshop on Building and Evaluation Resources for Biomedical Text Mining (BioTxtM 2012) at Language Resources and Evaluation (LREC). Istanbul, Turkey*, pages 16–23. Citeseer, 2012.

[130] Benjamin Nye, Junyi Jessy Li, Roma Patel, Yinfei Yang, Iain Marshall, Ani Nenkova, and Byron Wallace. A corpus with multi-level annotations of patients, interventions and outcomes to support language processing for medical literature. In *Proceedings of the 56th Annual Meeting of the Association for Computational Linguistics (Volume 1: Long Papers)*, pages 197–207, Melbourne, Australia, July 2018. Association for Computational Linguistics.

[131] Tomoko Ohta, Sampo Pyysalo, Makoto Miwa, Jin-Dong Kim, and Jun'ichi Tsujii. Event extraction for post-translational modifications. In *Proceedings of the 2010 Workshop on Biomedical Natural Language Processing*, pages 19–27, Uppsala, Sweden, July 2010. Association for Computational Linguistics.

[132] Tomoko Ohta, Sampo Pyysalo, Rafal Rak, Andrew Rowley, Hong-Woo Chun, Sung-Jae Jung, Sung-Pil Choi, Sophia Ananiadou, and Jun'ichi Tsujii. Overview of the pathway curation (PC) task of BioNLP shared task 2013. In *Proceedings of the BioNLP Shared Task 2013 Workshop*, pages 67–75, Sofia, Bulgaria, August 2013. Association for Computational Linguistics.

[133] Tomoko Ohta, Sampo Pyysalo, and Jun'ichi Tsujii. Overview of the epigenetics and post-translational modifications (EPI) task of BioNLP shared task 2011. In *Proceedings of BioNLP Shared Task 2011 Workshop*, pages 16–25, Portland, Oregon, USA, June 2011. Association for Computational Linguistics.

[134] Tomoko Ohta, Sampo Pyysalo, Jun'ichi Tsujii, and Sophia Ananiadou. Open-domain anatomical entity mention detection. volume W12-43. Association for Computational Linguistics, 2012.

[135] Tomoko Ohta, Yuka Tateisi, and Jin-Dong Kim. The genia corpus: An annotated research abstract corpus in molecular biology domain. In *Proceedings of the Second International Conference on Human Language Technology Research*, HLT '02, page 82–86, San Francisco, CA, USA, 2002. Morgan Kaufmann Publishers Inc.

[136] Serguei Pakhomov, Bridget McInnes, Terrence Adam, Ying Liu, Ted Pedersen, and Genevieve B Melton. Semantic similarity and relatedness between clinical terms: an experimental study. In *AMIA annual symposium proceedings*, volume 2010, page 572. American Medical Informatics Association, 2010.

[137] Dimitris Pappas, Petros Stavropoulos, Ion Androutsopoulos, and Ryan McDonald. BioMRC: A dataset for biomedical machine reading comprehension. In *Proceedings of the 19th SIGBioMed Workshop on Biomedical Language Processing*, pages 140–149, Online, July 2020. Association for Computational Linguistics.

[138] Mihir Parmar, Swaroop Mishra, Mirali Purohit, Man Luo, M Hassan Murad, and Chitta Baral. In-boxbart: Get instructions into biomedical multi-task learning. *arXiv preprint arXiv:2204.07600*, 2022.

[139] Ted Pedersen, Serguei VS Pakhomov, Siddharth Patwardhan, and Christopher G Chute. Measures of semantic similarity and relatedness in the biomedical domain. *Journal of biomedical informatics*, 40(3):288–299, 2007.

[140] Sampo Pyysalo and Sophia Ananiadou. Anatomical entity mention recognition at literature scale. *Bioinformatics*, 30(6):868–875, 2014.

[141] Sampo Pyysalo, Filip Ginter, Juho Heimonen, Jari Bj"orne, Jorma Boberg, Jouni J"arvinen, and Tapio Salakoski. Bioinfer: a corpus for information extraction in the biomedical domain. *BMC bioinformatics*, 8(1):1–24, 2007.

[142] Sampo Pyysalo, Tomoko Ohta, and Sophia Ananiadou. Overview of the cancer genetics (CG) task of BioNLP shared task 2013. In *Proceedings of the BioNLP Shared Task 2013 Workshop*, pages 58–66, Sofia, Bulgaria, August 2013. Association for Computational Linguistics.

[143] Sampo Pyysalo, Tomoko Ohta, Jin-Dong Kim, and Jun'ichi Tsujii. Static relations: a piece in the biomedical information extraction puzzle. In *Proceedings of the BioNLP 2009 Workshop*, pages 1–9, Boulder, Colorado, June 2009. Association for Computational Linguistics.

[144] Sampo Pyysalo, Tomoko Ohta, Makoto Miwa, Han-Cheol Cho, Jun'ichi Tsujii, and Sophia Ananiadou. Event extraction across multiple levels of biological organization. *Bioinformatics*, 28(18):i575–i581, 2012.

[145] Sampo Pyysalo, Tomoko Ohta, Rafal Rak, Dan Sullivan, Chunhong Mao, Chunxia Wang, Bruno Sobral, Jun'ichi Tsujii, and Sophia Ananiadou. Overview of the infectious diseases (ID) task of BioNLP shared task 2011. In *Proceedings of BioNLP Shared Task 2011 Workshop*, pages 26–35, Portland, Oregon, USA, June 2011. Association for Computational Linguistics.

[146] Sampo Pyysalo, Tomoko Ohta, and Jun'ichi Tsujii. Overview of the entity relations (rel) supporting task of bionlp shared task 2011. In *Proceedings of the BioNLP Shared Task 2011 Workshop*, BioNLP Shared Task '11, page 83–88, USA, 2011. Association for Computational Linguistics.

[147] Paolo Rosso, Julio Gonzalo, Raquel Martinez, Soto Montalvo, and Jorge Carrillo de Albornoz. Proceedings of the third workshop on evaluation of human language technologies for iberian languages (ibereval 2018). volume 2150 of *CEUR Workshop Proceedings*. CEUR-WS.org, 2018.

[148] Claudia Schulz, Josh Levy-Kramer, Camille Van Assel, Miklos Kepes, and Nils Hammerla. Biomedical concept relatedness – a large EHR-based benchmark. In *Proceedings of the 28th International Conference on Computational Linguistics*, pages 6565–6575, Barcelona, Spain (Online), dec 2020. International Committee on Computational Linguistics.

[149] M J Shardlow, N Nguyen, G Owen, C O'Donovan, A Leach, J McNaught, S Turner, and S Ananiadou. A new corpus to support text mining for the curation of metabolites in the ChEBI database. In *Proceedings of the Eleventh International Conference on Language Resources and Evaluation (LREC 2018)*, pages 280–285, May 2018.

[150] Chaitanya Shivade. Mednli – a natural language inference dataset for the clinical domain, 2017.

[151] Chaitanya Shivade. Mednli for shared task at acl bionlp 2019, 2019.

[152] Yoshinobu Kano Tomoko Ohkuma Shoko Wakamiya, Mizuki Morita and Eiji Aramaki. Overview of the ntcir-13 medweb task. *Proceedings of the 13th NTCIR Conference on Evaluation of Information Access Technologies (NTCIR-13)*, 2017.

[153] Felipe Soares, Viviane Moreira, and Karin Becker. A large parallel corpus of full-text scientific articles. In *Proceedings of the Eleventh International Conference on Language Resources and Evaluation (LREC-2018)*, 2018.

[154] Hakime Öztürk Soğancıoğlu, Gizem and Arzucan Özgür. Biosses: a semantic sentence similarity estimation system for the biomedical domain. *Bioinformatics*, 33(14):i49–i58, 2017.

[155] Amber Stubbs, Michele Filannino, Ergin Soysal, Samuel Henry, and Ozlem Uzuner. Cohort selection for clinical trials: n2c2 2018 shared task track 1. *J. Am. Medical Informatics Assoc.*, 26(11):1163–1171, 2019.

[156] Amber Stubbs, Christopher Kotfila, and Özlem Uzuner. Automated systems for the de-identification of longitudinal clinical narratives: Overview of 2014 i2b2/uthealth shared task track 1. *Journal of Biomedical Informatics*, 58:S11–S19, 2015.

[157] Lorraine Tanabe, Natalie Xie, Lynne H Thom, Wayne Matten, and W John Wilbur. GENETAG: a tagged corpus for gene/protein named entity recognition. *BMC Bioinformatics*, 6, 2005.

[158] Philippe Thomas, Roman Klinger, Laura Furlong, Martin Hofmann-Apitius, and Christoph Friedrich. Challenges in the association of human single nucleotide polymorphism mentions with unique database identifiers. *BMC Bioinformatics*, 12, 2011.

[159] Philippe Thomas, Tim Rockt"aschel, J"org Hakenberg, Yvonne Lichtblau, and Ulf Leser. Seth detects and normalizes genetic variants in text. *Bioinformatics*, Jun 2016.

[160] Thinh Hung Truong, Mai Hoang Dao, and Dat Quoc Nguyen. Covid-19 named entity recognition for vietnamese. In *Proceedings of the 2021 Conference of the North American Chapter of the Association for Computational Linguistics: Human Language Technologies*, page 2146–2153, 2021.

[161] George Tsatsaronis, Georgios Balikas, Prodromos Malakasiotis, Ioannis Partalas, Matthias Zschunke, Michael R Alvers, Dirk Weissenborn, Anastasia Krithara, Sergios Petridis, Dimitris Polychronopoulos, et al. An overview of the bioasq large-scale biomedical semantic indexing and question answering competition. *BMC bioinformatics*, 16(1):138, 2015.

[162] Ozlem Uzuner. Recognizing obesity and comorbidities in sparse data. *Journal of the American Medical Informatics Association*, 16(4):561–570, 07 2009.

[163] Ozlem Uzuner, Andreea Bodnari, Shuying Shen, Tyler Forbush, John Pestian, and Brett R South. Evaluating the state of the art in coreference resolution for electronic medical records. *Journal of the American Medical Informatics Association*, 19(5):786–791, 02 2012.

[164] Ozlem Uzuner, Ira Goldstein, Yuan Luo, and Isaac Kohane. Identifying patient smoking status from medical discharge records. *Journal of the American Medical Informatics Association*, 15(1):14–24, 01 2008.

[165] Özlem Uzuner, Yuan Luo, and Peter Szolovits. Evaluating the state-of-the-art in automatic de-identification. *Journal of the American Medical Informatics Association*, 14(5):550–563, 09 2007.

[166] Ozlem Uzuner, Imre Solti, and Eithon Cadag. Extracting medication information from clinical text. *J. Am. Medical Informatics Assoc.*, 17(5):514–518, 2010.

[167] Ozlem Uzuner, Brett R. South, Shuying Shen, and Scott L. DuVall. 2010 i2b2/va challenge on concepts, assertions, and relations in clinical text. *J. Am. Medical Informatics Assoc.*, 18(5):552–556, 2011.

[168] Erik M. van Mulligen, Annie Fourrier-Reglat, David Gurwitz, Mariam Molokhia, Ainhoa Nieto, Gianluca Trifiro, Jan A. Kors, and Laura I. Furlong. The eu-adr corpus: Annotated drugs, diseases, targets, and their relationships. *Journal of Biomedical Informatics*, 45(5):879–884, 2012. Text Mining and Natural Language Processing in Pharmacogenomics.

[169] Karin Verspoor, Antonio Jimeno Yepes, Lawrence Cavedon, Tara McIntosh, Asha Herten-Crabb, Zo"e Thomas, and John-Paul Plazzer. Annotating the biomedical literature for the human variome. *Database*, 2013, 2013.

[170] Veronika Vincze, Gy"orgy Szarvas, Rich'ard Farkas, Gy"orgy M'ora, and J'anos Csirik. The bioscope corpus: biomedical texts annotated for uncertainty, negation and their scopes. *BMC bioinformatics*, 9(11):1–9, 2008.

[171] David Wadden, Shanchuan Lin, Kyle Lo, Lucy Lu Wang, Madeleine van Zuylen, Arman Cohan, and Hannaneh Hajishirzi. Fact or fiction: Verifying scientific claims. pages 7534–7550, 2020.

[172] Xuan Wang, Xiangchen Song, Yingjun Guan, Bangzheng Li, and Jiawei Han. Comprehensive named entity recognition on CORD-19 with distant or weak supervision. *CoRR*, abs/2003.12218, 2020.

[173] M Weeber, J G Mork, and A R Aronson. Developing a test collection for biomedical word sense disambiguation. *Proc AMIA Symp*, pages 746–750, 2001.

[174] Chih-Hsuan Wei, Alexis Allot, Robert Leaman, and Zhiyong Lu. PubTator central: automated concept annotation for biomedical full text articles. *Nucleic Acids Research*, 47(W1):W587–W593, 05 2019.

[175] Chih-Hsuan Wei, Alexis Allot, Kevin Riehle, Aleksandar Milosavljevic, and Zhiyong Lu. tmvar 3.0: an improved variant concept recognition and normalization tool, 2022.

[176] Chih-Hsuan Wei, Bethany R Harris, Hung-Yu Kao, and Zhiyong Lu. tmvar: a text mining approach for extracting sequence variants in biomedical literature. *Bioinformatics*, 29(11):1433–1439, 2013.

[177] Chih-Hsuan Wei, Hung-Yu Kao, and Zhiyong Lu. GNormPlus: An integrative approach for tagging genes, gene families, and protein domains. *BioMed Research International*, 2015:1–7, Aug 2015.

[178] Chih-Hsuan Wei, Lon Phan, Juliana Feltz, Rama Maiti, Tim Hefferon, and Zhiyong Lu. tmvar 2.0: integrating genomic variant information from literature with dbsnp and clinvar for precision medicine. *Bioinformatics*, 34(1):80–87, 2018.

[179] Johannes Welbl, Nelson F. Liu, and Matt Gardner. Crowdsourcing multiple choice science questions. In *Proceedings of the 3rd Workshop on Noisy User-generated Text*, pages 94–106, Copenhagen, Denmark, September 2017. Association for Computational Linguistics.

[180] Johannes Welbl, Pontus Stenetorp, and Sebastian Riedel. Constructing datasets for multi-hop reading comprehension across documents. *Transactions of the Association for Computational Linguistics*, 6:287–302, 2018.

[181] Zhengyun Zhao, Qiao Jin, and Sheng Yu. Pmc-patients: A large-scale dataset of patient notes and relations extracted from case reports in pubmed central, 2022.

[182] Markus Zlabinger, Marta Sabou, Sebastian Hofst"atter, and Allan Hanbury. Effective crowd-annotation of participants, interventions, and outcomes in the text of clinical trial reports. In *Findings of the Association for Computational Linguistics: EMNLP 2020*, pages 3064–3074, Online, November 2020. Association for Computational Linguistics.

[183] Maryam Zolnoori, Kin Wah Fung, Timothy B. Patrick, Paul Fontelo, Hadi Kharrazi, Anthony Faiola, Yi Shuan Shirley Wu, Christina E. Eldredge, Jake Luo, Mike Conway, Jiaxi Zhu, Soo Kyung Park, Kelly Xu, Hamideh Moayyed, and Somaieh Goudarzvand. A systematic approach for developing a corpus of patient reported adverse drug events: A case study for SSRI and SNRI medications. *Journal of Biomedical Informatics*, 90, 2019.