# OpenReview forum: "BigBio: A Framework for Data-Centric Biomedical Natural Language Processing"
_NeurIPS.cc/2022/Track/Datasets_and_Benchmarks — NeurIPS 2022 Datasets and Benchmarks _

### Official Review · Reviewer_vUee · 2022-07-27
**A very good paper with a focused contribution.**

**Rating:** 7
**Confidence:** 3
**Clarity:** This paper is well-written.

**Strengths:**

1. This paper proposes  BIGBIO, a community resource for programmatically accessing biomedical NLP 33 datasets at scale and encouraging reproducibility when generating meta-datasets, which are well-written and well-motivated.

2. This paper contains the comprehensive process of task schema harmonization, data auditing, and contribution guidelines.

3. This paper conducts the zero-shot evaluation of biomedical prompts and large-scale, multi-task learning.

**Weaknesses:**

1. I think the writing can be further improved to highlight the motivation in the introduction section.

**Additional Feedback:**

I recommend revising the introduction section to improve the writing and highlight the motivation.

**Correctness:**

The claims in this paper are correct, and the datasets are constructed in a sound way. The evaluation methods and experiment design are correct.

**Documentation:**

Yes.

**Ethics:**

This paper has no ethics concerns.

**Relation To Prior Work:**

Yes.

**Summary And Contributions:**

This paper proposes BIGBIO, a community library of 126+ biomedical NLP datasets, currently covering 12 task categories and 10+ languages. BIGBIO facilitates reproducible meta-dataset curation via programmatic access to datasets and their metadata. Moreover, it is compatible with current platforms for prompt engineering and few/zero-shot language model evaluations. This paper discusses the process for task schema harmonization, data auditing, and contribution guidelines and outlines two illustrative use cases. Overall, this paper is well-written and well-motivated and releases a very useful and important resource for the community.

---

> ### Author Response · Authors · 2022-08-24
> **We thank the reviewer for their positive feedback**
>
> 1. *I think the writing can be further improved to highlight the motivation in the introduction section.*
>
> We agree and have revised the introduction to highlight our motivations more clearly.

---

### Official Review · Reviewer_Ewpa · 2022-07-27
**This work collects and organises a large number of different biological natural language processing tasks, and provides a complete set of benchmark tools to support them.**

**Rating:** 8
**Confidence:** 3
**Correctness:** The dataset and evaluation methods ar…
**Clarity:** The paper is generally clear, althoug…

**Strengths:**

1. The paper provides a professional and complete system solution for large-scale multi-task learning in biology
2. The article provides a useful exemplar for designing unified dataloaders and schema harmonisation.
3. This work systematically collects and organises a large number of existing datasets, which is a tedious but important task that should contribute significantly to the standardisation and advancement of the field.

**Weaknesses:**

1. The paper writing can be improved. Some descriptions are hard to follow (e.g., line 24).
2. In experiments such as Figure 3 and Table 2 only generative pre-trained language models are included, but there are also many important masked pre-trained language models, such as RoBERTa and PubmedBert, which should not be completely ignored.
3. There may be data-overlap between the included task and the training data for pre-trained language models, which may introduce unfair comparisons: a model may perform well simply because it has seen more repeated data. How to avoid such bias?


**Additional Feedback:**

NA

**Documentation:**

The documentation is satisfactory.

**Ethics:**

No specific converns.

**Relation To Prior Work:**

Related works are clearly discussed.

**Summary And Contributions:**

This paper introduces a biological natural language processing benchmark with a large number of different tasks and provides complete community tools and third-party library support.

---

> ### Author Response · Authors · 2022-08-24
> **We thank the reviewer for their positive comments and support of our work**
>
> >*The paper writing can be improved. Some descriptions are hard to follow (e.g., line 24).*
>
> We agree and have revised the introduction and Section 2 to improve clarity.
>
> > *In experiments such as Figure 3 and Table 2 only generative pre-trained language models are included, but there are also many important masked pre-trained language models, such as RoBERTa and PubmedBert, which should not be completely ignored.*
>
> This is a fair point and current limitation of our evaluation pipeline. Our initial goal was to assess state-of-the-art zero-shot performance with prompting and in this setting, large, generative language models (autoregressive and sequence-to-sequence) consistently outperform zero-shot formulations of masked language models (MLM). We agree that MLMs are an important family of language models and a key future research direction for BigBio, particularly in the context of prompt-based fine-tuning for improving few and zero-shot performance in smaller language models [1]. We now discuss these details in the manuscript.
>
> - [1] Tianyu Gao, Adam Fisch, and Danqi Chen. 2021. Making Pre-trained Language Models Better Few-shot Learners. In Proceedings of the 59th Annual Meeting of the Association for Computational Linguistics and the 11th International Joint Conference on Natural Language Processing (Volume 1: Long Papers), pages 3816–3830, Online. Association for Computational Linguistics.
>
> > *There may be data-overlap between the included task and the training data for pre-trained language models, which may introduce unfair comparisons: a model may perform well simply because it has seen more repeated data. How to avoid such bias?*
>
> This may be true for the models that were pretrained with PubMed/PMC data (SciFive, GPT-Neo-1.3B, GPT-J-6B). However, since all of these models performed very poorly compared to the T0 family, which was not trained on any PubMed/PMC data, we did not conduct a more systematic analysis of leakage in pretraining corpora. We agree that the general problem of data-overlap is a significant concern for fair evaluations of language models. One of the key motivations of BioBio is to provide tools that enable more systematic generation of biomedical pretraining corpora to mitigate potential data leakage. For example, you can now easily enumerate the PMIDs (unique PubMed/PMC IDs) of all splits for all PubMed/PMC datasets in BigBIO and exclude them from future language model corpora. We also describe a preliminary analysis of dataset overlap within BigBio in Appendix H.

---

### Official Review · Reviewer_xpav · 2022-07-28

**Rating:** 7
**Confidence:** 4
**Correctness:** Yes
**Clarity:** The paper is clear and well-written.

**Strengths:**

The BIGBIO differs from all previous works by providing programmatic access to datasets that are unit-tested, supporting various languages. They provide guides for new datasets. BIGBIO can be used in zero-shot biomedical language evaluation as well as multi-task learning. They have provided a Python package that supports 126 biomedical tasks that can be used for multiple tasks.  BIGBIO increases robustness for data-centric machine learning, and it also reduces the amount of time and work needed for training and evaluating models. Most importantly, it is a useful interface for loading large number of biomedical datasets. Furthermore, the paper is easy to follow except in few instances (which are discussed below).

**Weaknesses:**

Figure 1 and Figure 2 lack any explanation in the paper. They have not been cited in the text. The paper does not elaborate much on how it supports the datasets that need credentialed access from the users as such datasets have restricted usage. Also, there is no information about the values represented in Table 2. Is it F1 scores? Furthermore, line 222 mentions "trained or finetuned". Is it supposed to be "pre-trained" instead of "trained"? That part of the text is unclear and should be properly explained. Some concepts like "majority class" have not been explained.

Minor weaknesses:
Missing citation in line 219: "We replicate the finding in Sanh et al. that models".
The text in Figure 2 is too small.

**Additional Feedback:**

The library introduced in the paper has the potential of being a good resource for the biomedical community.

**Documentation:**

Proper documentation is provided. The dataset and the scripts are available in their GitHub repository. Details about hosting, licensing and maintenance plans are provided. It is reproducible.

**Ethics:**

No ethical concerns.

**Relation To Prior Work:**

Yes, the authors have described the lack of such a community library for biomedical datasets which is the motivation behind the work. They have clearly explained that the BIGBIO differs from all previous works by focusing on the infrastructure and curation needed for generation of meta-datasets. Unlike the previous works, BIGBIO provides programmatic access to datasets.

**Summary And Contributions:**

The paper has introduced a community library- BIGBIO which has more than 126 biomedical NLP datasets, covering multiple tasks and supporting various languages. It allows programmatic access to datasets and even supports platforms for prompt engineering along with evaluation of few/zero shot language model.

---

> ### Author Response · Authors · 2022-08-24
> **We thank the reviewer for their positive feedback and comments**
>
> > *Figure 1 and Figure 2 lack any explanation in the paper. They have not been cited in the text.*
>
> Thank you for catching this. We now explain and cite these figures in the paper.
>
> > *The paper does not elaborate much on how it supports the datasets that need credentialed access from the users as such datasets have restricted usage.*
>
> This is a good point. We have expanded Figure 1 and corresponding text to better illustrate how users can use private (local) datasets. Currently this process is manual (i.e., users must login to the source website and download files) however we are exploring tools to better streamline access in the future, e.g., authentication tools.
>
> > *Also, there is no information about the values represented in Table 2. Is it F1 scores?*
>
> We apologize for this ambiguity. The metrics (Pearson’s correlation for BIOSSES, ROUGE-1 for NER, and accuracy for the remaining datasets) are mentioned in text (lines 211-212) however to improve clarity, we now mention them directly in Table 2.
>
> > *Furthermore, line 222 mentions "trained or finetuned". Is it supposed to be "pre-trained" instead of "trained"? That part of the text is unclear and should be properly explained.*
>
> Thank you for catching this typo. The text has been updated to indicate we mean pretrained. This is in the context of language models that have been pretrained on biomedical language data from PubMed/PMC. We have updated the text to make this more clear.
>
> > *Some concepts like "majority class" have not been explained.*
>
> Thank you -- we now define the majority class baseline in text, i.e., a simple rule-based model that always predicts the majority class label.
>
> > *Minor weaknesses: Missing citation in line 219: "We replicate the finding in Sanh et al. that models". The text in Figure 2 is too small.*
>
> Thank you -- we have fixed the missing reference and redesigned Figure 2 to make the text larger.

---

### Official Review · Reviewer_2fRy · 2022-07-30
**Important community library for BioNLP tasks, but more research needed to show the utility of the datasets.**

**Rating:** 7
**Confidence:** 3
**Clarity:** Yes.

**Strengths:**

* Direct contribution to dataset quality, data governance, task schema harmonization, ML workflow integration, etc.
* Highly-credibility crowd-sourced effort in the dataset curation through a hackathon
* Identifying the performance limitation of existing LMs towards solving BioNLP tasks. The results from the general LM were somewhat expected and aligned with the previous research on domain-specific LM performance (e.g., PubMedBERT performance comparison).
* Results from T0 class of models were promising for shaping future research direction on specialized domains like biology or chemistry; The performance may suggest that the models that were capable to learn the task or functionality directly may generalize to new scientific domains.


**Weaknesses:**

* Lack of discussion towards few-shot NER implementation. NER finetuning performance is reported in the Section 5.2 with MTL model. However, there seems no performance advantage over the SOTA baseline. Bit skeptical on the poor performance of MTL over the LinkBERT though MTL encoder was initialized with the BioLinkBERT-base.
* How does the prompt designing on BioNLP tasks differ from general NLP tasks? The difficulty on MedNLI, GAD, and BIOSSES remained due to the lack of selecting best prompts?
* Since the authors proposing a framework design, there might be additional elements that they need to consider to adopt this library in the scientific ML domain. For example, complex wide data policy/strategy, complex security consideration, E.g., classified data, propriety data, scientific value, scale data infrastructure and frameworks for HPC facilities.


**Additional Feedback:**

N/A

**Correctness:**

The claims made in the submission is correct for the most parts. However, the reasoning on poor performance of MTL over the SOTA baseline can be improved.

**Documentation:**

Yes

**Ethics:**

No. The authors explicitly mention that datasets are compliant with the United States Health Insurance Portability and Accountability Act (HIPAA).

**Relation To Prior Work:**

Yes

**Summary And Contributions:**

This paper introduced a community library, BIGBIO, that provides curated set of meta-datasets and data governance capabilities for biomedical NLP tasks evaluation. This work differs from existing BioNLP benchmark datasets as it provides tooling to automate ML workflows through programmatically accessing and ingesting data.

---

> ### Author Response · Authors · 2022-08-24
> **We thank the reviewer for their positive feedback and answer their questions below**
>
> > *Lack of discussion towards few-shot NER implementation.*
>
> Thank you for this feedback. We now include results and discussion for zero-shot evaluation of 5 NER tasks from the BLURB benchmark in Section 5.1
>
> > *NER finetuning performance is reported in the Section 5.2 with MTL model. However, there seems no performance advantage over the SOTA baseline. Bit skeptical on the poor performance of MTL over the LinkBERT though MTL encoder was initialized with the BioLinkBERT-base.*
>
> The original inspiration for our MTL experiments was Muppet [1], which explored using massive, multi-task learning for additional pretraining in general domain NLP datasets. This scale of MTL (50+ datasets) is underexplored in biomedical NLP, where prior work analyzes smaller-scale  (<15 datasets), often more homogenous task mixtures using classification task heads. Our original hypothesis was that we would observe performance increases that scaled linearly with the addition of more datasets, as reported by Muppet. However this was not the case. We suspect several possible explanations
>   - **Task Head Configurations**: Muppet examined different configurations of shared task heads and found that for some tasks, using a per-dataset classification head performed well (sentence prediction tasks) while in other tasks, using a per-dataset head resulted in severe overfitting (commonsense and machine reading comprehension). We did not systematically explore different task head configurations here due to computational cost.
>   - **Available Task Mixtures**: Muppet focuses on general reasoning tasks (e.g., summarization, reading comprehension). Biomedical datasets, in contrast, are skewed towards information extraction-style tasks [2]. We hypothesize the relative paucity of similar reasoning-type datasets in biomedical NLP may impact the overall benefit of using MTL with task head approaches.
>   - **Task Formulation**: In our MTL setup, we take the common approach of augmenting task inputs by adding non-linguistic marker tokens to capture structural elements of inputs, e.g., denoting relational entities with prefix/suffix tokens.  While this defines a unified input format, it is unclear how well this strategy facilitates transfer learning across tasks. Some evidence is suggested by In-BoXBART [3], which looked at MTL using 32 biomedical tasks reformulated as unified text-to-text tasks using prompts. Their task mixture is similar to ours and their text outputs focus on similar structural outputs. Their baseline MTL method ("V-BB" in their Table 2) only trains on input/outputs without prompt context and substantially underperforms corresponding single-task models, similar to our results. However, when In-BoXBART includes prompt information ("I-BB"), they observe consistent performance benefits over single-task models using MTL.
>
> We believe these results are further motivation for BigBio and the need for easier prompt development.
> - [1] Aghajanyan et al. 2021. Muppet: Massive Multi-task Representations with Pre-Finetuning.
> - [2] Fries et al. 2022. Dataset Debt in Biomedical Language Modeling.
> - [3] Parmar et al. 2022. In-BoXBART: Get Instructions into Biomedical Multi-Task Learning.
>
> > *How does the prompt designing on BioNLP tasks differ from general NLP tasks? The difficulty on MedNLI, GAD, and BIOSSES remained due to the lack of selecting best prompts?*
>
> This is a good point. We have updated the manuscript to provide more detail on the prompt engineering process. Our prompt templates are largely modifications of similar tasks provided in the PromptSource repository. To avoid biasing results to any specific language model, we didn't do iterative prompt tuning. Methods for identifying good prompts is an active research area so we leave this for future work.
>
> > *Since the authors proposing a framework design, there might be additional elements that they need to consider to adopt this library in the scientific ML domain. For example, complex wide data policy/strategy, complex security consideration, E.g., classified data, propriety data, scientific value, scale data infrastructure and frameworks for HPC facilities.*
>
> We hope to work with the open source community to continue developing tools to facilitate adoption in complex scientific compute settings. BigBio currently supports local loading of private datasets (see updated Figure 1), which enables using our framework on secure data. HIPAA compliant clusters typically prohibit external downloading which prevents running BigBio out-of-box. To better support these environments, we will release a pre-configured Docker image that contains the BigBio workflow and the subset of redistributable BigBio datasets and models.
>
> > *The claims made in the submission is correct for the most parts. However, the reasoning on poor performance of MTL over the SOTA baseline can be improved.*
>
> We have expanded our discussion on MTL performance in the manuscript, per our answers above.

---

### Author Response · Authors · 2022-08-24
**Manuscript has been updated**

We thank all of the reviewers for their supportive comments and constructive feedback. We have now updated the manuscript to reflect reviewer suggestions.

---

### Meta-Review · Area_Chair_qxc5 · 2022-09-11

**Recommendation:** Accept
**Confidence:** 4

**Metareview:**

This submission introduces a "meta dataset", consolidating and harmonizing multiple bio-medical NLP datasets across several languages.

The submission generated good discussions between the reviewers and the authors as it shows potential utility of the corresponding dataset. There is hope that this dataset will spur forward research in biomedical NLP.

---

### Decision · Program_Chairs · 2022-09-16

Accept